# Convergence Analysis of Decentralized Hessian-/Jacobian-Free Algorithm for Nonconvex Stochastic Bi-Level Optimization

**Yihan Zhang** [1] **Xinwen Zhang** [1] **My T. Thai** [2] **Jie Wu** [1] **Hongchang Gao** [1]

## Abstract

Decentralized stochastic bi-level optimization has been actively studied in recent years. However, existing studies assume that the lower-level loss function is strongly convex, which limits their applicability to many machine learning models. To address this limitation, in this paper, we propose a novel decentralized stochastic first-order optimization algorithm, which does not require second-order Hessian or Jacobian matrices, for the setting where the lower-level loss function is nonconvex but satisfies the Polyak–Łojasiewicz (PL) condition. Additionally, unlike existing single-agent methods that introduce a regularization term to the lower-level loss function to artificially enforce strong convexity, our algorithm does not require such modification. Moreover, our algorithm employs a constant single-timescale learning rate for updating variables, which is different from the time-dependent and two-timescale learning rate schedules used in prior work. To establish the convergence rate, we develop a new convergence analysis framework for the pure PL condition, rather than relying on the artificial strong convexity introduced through regularization in existing single-agent methods. To the best of our knowledge, this is the first algorithm for nonconvex decentralized bi-level optimization that offers theoretical convergence guarantees under mild conditions. Finally, our extensive experimental results on hyperparameter optimization and model pruning applications validate the efficacy of the proposed algorithm.

[1]Department of Computer and Information Sciences, Temple University, Philadelphia, USA [2]Department of Computer & Information Science & Engineering, University of Florida, Gainesville, USA. Correspondence to: Hongchang Gao <hongchang.gao@temple.edu>.

*Proceedings of the 43rd International Conference on Machine Learning*, Seoul, South Korea. PMLR 306, 2026. Copyright 2026 by the author(s).

## 1. Introduction

This paper focuses on the decentralized stochastic bi-level optimization problem, which is defined as follows:

$$\min_{x\in\mathbb{R}^{d_x}, y\in Y^*(x)} f(x,y) = \frac{1}{K}\sum_{k=1}^{K} f^{(k)}(x,y)$$

$$s.t. \quad Y^*(x) = \arg\min_{y\in\mathbb{R}^{d_y}} g(x,y) = \frac{1}{K}\sum_{k=1}^{K} g^{(k)}(x,y) ,$$

$$(1)$$

where $f^{(k)}(x,y) = \mathbb{E}[f^{(k)}(x,y;\xi^{(k)})]$ is the upper-level loss function on the $k$-th worker, $g^{(k)}(x,y) = \mathbb{E}[g^{(k)}(x,y;\zeta^{(k)})]$ is the lower-level loss function on the $k$-th worker, $\xi^{(k)}$ and $\zeta^{(k)}$ denote data distributions on the $k$-th worker, and all workers $\{1,2,\cdots,K\}$ conduct peer-to-peer communication within a communication graph. Unlike existing decentralized stochastic bi-level optimization (Chen et al., 2025b; Yang et al., 2022; Gao et al., 2023; Chen et al., 2023), where the lower-level loss function is strongly convex in $y$, this paper assumes that it is nonconvex but satisfies the PL condition in $y$.

Decentralized stochastic bi-level optimization has recently gained tremendous attention because it covers a wide range of machine learning models, such as meta learning (Finn et al., 2017), hyperparameter optimization (Franceschi et al., 2018), neural architecture search (Liu et al., 2019). To address the challenges in estimating the hypergradient in the decentralized setting, a series of algorithms (Chen et al., 2025b; Yang et al., 2022; Gao et al., 2023; Chen et al., 2023; Dong et al., 2025; Zhang et al., 2026b; Kong et al., 2025; Zhu et al., 2024) have been developed. In terms of the strategy for estimating the Hessian-inverse-vector product, these methods can be categorized into the double-loop algorithm (Chen et al., 2025b; Yang et al., 2022; Gao et al., 2023; Chen et al., 2023) and the single-loop algorithm (Zhang et al., 2026b; Dong et al., 2025; Kong et al., 2025; Zhu et al., 2024). However, these algorithms focus on the setting in which the lower-level loss function is strongly convex in $y$. Recently, (Qin et al., 2025) takes a step further by studying decentralized bi-level optimization with a convex lower-level loss function. Since the strong

convexity and the regular convexity are typically not satisfied in practical applications, these algorithms cannot be applied to practical machine learning models.

To handle the case that goes beyond the strong convexity, some efforts (Shen & Chen, 2023; Chen et al., 2024; Xiao et al., 2023; Liu et al., 2024; Kwon et al., 2024b) have been made to solve the bi-level optimization problem with a non-convex lower-level loss function in the single-agent setting. In particular, since nonconvexity could result in the noncontinuity of $Y^*(x)$, which makes it intractable to compute the hypergradient, a couple of studies investigate further assumptions to make it tractable. For example, (Shen & Chen, 2023) studied the PL condition and introduced the penalty method to handle the nonconvex lower-level loss function, where the convergence rate was established for the full gradient descent algorithm. (Kwon et al., 2024b) also studied the PL condition but focused on the stochastic and constrained settings, whose convergence rate is then refined by (Chen et al., 2024) for stochastic unconstrained bi-level problems.

It can be seen that the PL condition of the lower-level loss function plays an important role in guaranteeing the tractability of nonconvex bi-level optimization problems. Moreover, the PL condition is easy to be satisfied by many machine learning models, such as the overparameterized neural networks (Liu et al., 2022a). Therefore, the goal of this paper is to develop a decentralized algorithm for bi-level optimization problems where the lower-level loss function satisfies only the PL condition, rather than strong convexity, and to investigate its theoretical convergence rate. Specifically, we aim to develop an efficient decentralized stochastic bi-level gradient descent algorithm based on the penalty framework (Kwon et al., 2024b; Shen & Chen, 2023), which relies solely on first-order gradients rather than computationally inefficient second-order Hessian or Jacobian matrices. However, developing a decentralized Hessian-/Jacobian-free (i.e., first-order) algorithm for Eq. (1), where the lower-level loss function satisfies the PL condition, presents unique challenges as follows.

**Challenges in algorithm design: It remains unclear whether a decentralized stochastic Hessian-/Jacobian-free algorithm without additional operations can be achieved.** In the single-agent setting, (Shen & Chen, 2023) developed a double-loop first-order algorithm that uses a *deterministic gradient*, which is infeasible for large-scale machine learning models. (Kwon et al., 2024b) showed that a double-loop algorithm is required to ensure convergence when using first-order *stochastic gradients*, and subsequently proposed a single-loop first-order algorithm based on variance-reduced gradients. However, both *stochastic* algorithms in (Kwon et al., 2024b) introduce additional computational steps. Specifically, *(Kwon et al.,*

*2024b) added a regularization term to the lower-level optimization problem to make it strongly convex.* As a result, the algorithm introduces additional variables along with their associated updates, increasing computational overhead. This leads to a natural question: *Is it possible to design a single-loop decentralized stochastic first-order algorithm for Eq. (1) without imposing artificial strong convexity?*

**Challenges in convergence analysis: It remains unclear how to establish the convergence rate for a single-loop decentralized stochastic first-order algorithm when only the pure PL condition is satisfied.** As mentioned earlier, in the single-agent setting, (Kwon et al., 2024b) enforced artificial strong convexity on the lower-level loss function via a regularization term. As a result, their convergence analysis for the lower-level optimization problem relies on techniques developed for strongly convex optimization (see the proof of its Lemma C.1). Although (Chen et al., 2024) established a convergence rate under the pure PL condition, it focuses only on the double-loop algorithm. Thus, it remains unclear whether the single-loop stochastic first-order algorithm can converge under the pure PL condition. Moreover, the decentralized setting introduces additional challenges, such as non-zero consensus error and larger gradient estimation error. This raises another natural question: *Is it possible to establish the convergence rate of a single-loop decentralized stochastic first-order algorithm without enforcing artificial strong convexity?*

To address these challenges, we develop *a novel single-loop decentralized Hessian-/Jacobian-free algorithm based on fully first-order stochastic gradients* for solving Eq. (1). Specifically, we convert the decentralized stochastic bi-level optimization problem in Eq. (1) into a single-level minimax problem based on the penalty framework, and then develop a decentralized stochastic variance-reduced gradient descent ascent algorithm to solve it. Moreover, we establish the convergence rate of our algorithm based on the pure PL condition. More specifically, we have made the following contributions in algorithm design and theoretical analysis.

**Contributions in algorithm design.** Firstly, unlike the existing single-agent method (Kwon et al., 2024b), *our algorithm does not introduce a regularization term to enforce artificial strong convexity*. As a result, it reduces computational overhead by avoiding the update of additional variables. Secondly, (Kwon et al., 2024b) employs a time-dependent learning rate and a two-timescale schedule for different variables, which is difficult to tune in practice and may leak training stage information to malicious workers in the decentralized setting. In contrast, our algorithm *uses a single-timescale constant learning rate for all variables, while still guaranteeing convergence*.

**Contributions in convergence analysis.** Unlike the existing single-agent method (Kwon et al., 2024b), which introduces a regularization term to strengthen the PL condition into strong convexity and then applies techniques developed for strongly convex optimization to establish convergence, we propose a new convergence analysis framework that relies purely on the PL condition. Specifically, *we introduce a novel approach to quantify how close the current iterate of the lower-level variable is to its optimal value at the $t$-th iteration. Based on this new metric, we construct a novel potential function to derive the convergence rate of our algorithm.* Our analysis shows that the proposed algorithm achieves a convergence rate of $O\left(\frac{\kappa^9}{K(1-\lambda)^2\epsilon^5}\right)$, where $\kappa$ is the condition number and $1-\lambda$ represents the spectral gap.

To the best of our knowledge, this is the first decentralized stochastic first-order algorithm that provides rigorous convergence guarantees without imposing artificial strong convexity. Finally, we conduct extensive experiments on hyperparameter optimization and model pruning tasks, and the results validate the effectiveness of our algorithm.

## 2. Related Work

### 2.1. Centralized Stochastic Bi-level Optimization

Stochastic bi-level optimization has gained tremendous attention in recent years because it covers various machine learning models, such as meta learning (Finn et al., 2017), hyperparameter optimization (Franceschi et al., 2018), neural architecture search (Liu et al., 2019). The main difficulty in solving stochastic bi-level optimization problems lies in estimating the hypergradient of the upper-level loss function. When the lower-level loss function is strongly convex, its optimal solution $Y^*(x)$ is singleton, and then the hypergradient is well defined. Various algorithms (Ghadimi & Wang, 2018; Ji et al., 2021; Chen et al., 2021; Hong et al., 2023; Guo et al., 2025; Yang et al., 2021; Khanduri et al., 2021b; Dagréou et al., 2022; 2024; Li et al., 2022) have been developed recently for this problem class. For instance, (Ghadimi & Wang, 2018) uses the Neumann series expansion approach to approximately compute Hessian-inverse-vector product for estimating hypergradient, which is a double-loop algorithm. Inspired by this, (Ji et al., 2021; Chen et al., 2021; Hong et al., 2023; Yang et al., 2021; Khanduri et al., 2021b) developed faster algorithms based on different techniques, such as the mini-batch gradient and variance-reduced gradient. On the other hand, (Dagréou et al., 2022; 2024; Li et al., 2022) developed the single-loop algorithm for nonconvex-strongly-convex bi-level problems. However, they all need to compute second-order Hessian and Jacobian matrices, which is computationally expensive for practical machine learning

models. To address this issue, (Kwon et al., 2023; Shen & Chen, 2023; Liu et al., 2024; Lu & Mei, 2024) developed the fully first-order algorithm, which converts the bi-level optimization problem into a minimax problem with the penalty framework. This method is then extensively studied in (Kwon et al., 2024b; Chen et al., 2024; 2025a; Kwon et al., 2024a; Lu & Mei, 2026; Zeng et al., 2026). Moreover, there are some other studies (Jiang et al., 2023; Sow et al., 2022; Liu et al., 2021; 2023a; 2024) for the lower-level loss function without strong convexity. However, they either focus on the deterministic gradient or require second-order Hessian or Jacobian matrices.

### 2.2. Decentralized Stochastic Bi-level Optimization

To optimize bi-level optimization problems in the distributed setting to train machine learning models on distributed data, a series of decentralized stochastic bi-level optimization algorithms (Chen et al., 2025b; Yang et al., 2022; Gao et al., 2023; Chen et al., 2023; Dong et al., 2025; Zhang et al., 2026b; Kong et al., 2025; Zhu et al., 2024; Liu et al., 2022b; Lu et al., 2022; Liu et al., 2023b; Niu et al., 2025), federated learning algorithms (Gao, 2022; Tarzanagh et al., 2022; Li et al., 2023; Zhang et al., 2025; Yang et al., 2025), and distributed algorithms for the special compositional optimization problem (Gao & Huang, 2021; Gao et al., 2022; Gao, 2024a;b; Zhang et al., 2023; 2024; Gao et al., 2024) have been developed. For example, (Chen et al., 2025b) proposed to compute and communicate the Jacobian-Hessian-inverse product in an inner loop, while (Chen et al., 2023) handled the Hessian-inverse-vector product in the inner loop. (Gao et al., 2023; Yang et al., 2022) used the Neumann series expansion approach to approximately compute Hessian-inverse-vector product in the inner loop. (Zhang et al., 2026b; Dong et al., 2025; Kong et al., 2025; Zhu et al., 2024) developed the single-loop algorithm, where the Hessian-inverse-vector product is approximated by an additional gradient descent procedure. However, all these methods require the computation of second-order Jacobian and Hessian matrices, leading to high computational and communication costs. To address this issue, (Wang et al., 2025) recently developed a fully first-order method based on (Kwon et al., 2023), which avoids computing second-order Hessian and Jacobian matrices. However, all these existing decentralized stochastic bi-level optimization algorithms focus on strongly convex lower-level loss functions. Only a recent work (Zhang et al., 2026a) studies the decentralized bi-level optimization with nonconvex lower-level loss functions. However, it focuses on the heavy-tailed noise setting, where a normalized gradient estimator is used to update variables. Such a gradient estimator requires a small learning rate to accommodate the heavy-tailed noise, which is unnecessary for the finite-variance setting.

# 3. Algorithm Design

## 3.1. Problem Reformulation

Following (Shen & Chen, 2023; Kwon et al., 2024b; Chen et al., 2024), we convert Eq. (1) into a constrained single-level optimization problem as follows:

$$\min_{x \in \mathbb{R}^{d_x}, y \in \mathbb{R}^{d_y}} f(x, y) ,$$
$$s.t. \quad g(x, y) \leq \min_{z \in \mathbb{R}^{d_y}} g(x, z) . \quad (2)$$

Then, based on the penalty strategy, we further convert it into an unconstrained problem as follows:

$$\min_{x \in \mathbb{R}^{d_x}, y \in \mathbb{R}^{d_y}} \max_{z \in \mathbb{R}^{d_y}} f(x, y) + \frac{1}{\rho}(g(x, y) - g(x, z)) , \quad (3)$$

where $\rho > 0$ is the penalty parameter. In terms of existing studies (Kwon et al., 2024b; Chen et al., 2024), the reformulated Eq. (3) can approximate the original bi-level optimization problem in Eq. (1) as good as possible by controlling the penalty parameter $\rho$. Specifically, denoting

$$\mathcal{L}(x) = \min_{y \in Y^*(x)} f(x, y) , \quad (4)$$

$$\mathcal{L}_\rho(x) = \frac{1}{\rho} \Big( \min_{y \in \mathbb{R}^{d_y}} \underbrace{(\rho f(x, y) + g(x, y))}_{h_\rho(x,y)} - \min_{z \in \mathbb{R}^{d_y}} g(x, z) \Big) ,$$

the existing studies (Kwon et al., 2024b; Chen et al., 2024) have characterized how $\mathcal{L}_\rho(x)$ approximates $\mathcal{L}(x)$ and the tractability of $\min_{x \in \mathbb{R}^{d_x}} \mathcal{L}_\rho(x)$. More specifically, given appropriate assumptions, Lemma A.14 indicates that the solution set $Y_\rho^*(x) = \arg\min_{y \in \mathbb{R}^{d_y}} h_\rho(x, y)$ is Lipschitz continuous. Then, based on Lemma A.14, Lemma A.15 demonstrates the existence of $\nabla \mathcal{L}_\rho(x)$, which confirms the tractability of $\min_{x \in \mathbb{R}^{d_x}} \mathcal{L}_\rho(x)$. Then we can solve the reformulated Eq. (3) to find the solution of $\min_{x \in \mathbb{R}^{d_x}} \mathcal{L}_\rho(x)$. Finally, based on Lemma A.16, which characterizes how well $\mathcal{L}_\rho(x)$ approximates $\mathcal{L}(x)$, we can know how the solution of $\min_{x \in \mathbb{R}^{d_x}} \mathcal{L}_\rho(x)$ approximates that of $\min_{x \in \mathbb{R}^{d_x}} \mathcal{L}(x)$.

Based on the aforementioned reformulation, we will develop a new single-loop decentralized optimization algorithm to solve the following problem:

$$\min_{x \in \mathbb{R}^{d_x}, y \in \mathbb{R}^{d_y}} \max_{z \in \mathbb{R}^{d_y}} \mathcal{L}_\rho(x, y, z) \quad (5)$$

$$\triangleq \frac{1}{K} \sum_{k=1}^{K} \Big( f^{(k)}(x, y) + \frac{1}{\rho}(g^{(k)}(x, y) - g^{(k)}(x, z)) \Big) ,$$

where $\rho > 0$ is the penalty parameter, which will be determined in our convergence analysis. Note that, even though Eq. (5) is a decentralized minimax optimization problem, the existing convergence analysis for standard minimax optimization cannot be directly applied to Eq. (5) due to the

existence of the penalty parameter $\rho$, which can significantly affect the convergence rate.

To summarize, the reformulated Eq. (5) is a reasonable approximation for Eq. (1). Therefore, we will design an efficient algorithm to solve the reformulated Eq. (5) and study study how fast the algorithm designed for Eq. (5) can find the stationary point of the original decentralized stochastic bi-level optimization problem in Eq. (1).

**Terminology.** In this paper, we use the matrix $W \in \mathbb{R}^{K \times K}$ to denote the communication graph composed by workers $\{1, 2, \cdots, K\}$. Specifically, for any element $w_{ij}$ of $W$ where $i \in \{1, \cdots, K\}$ and $j \in \{1, \cdots, K\}$, it is always non-negative. More specifically, $w_{ij} > 0$ indicates that the worker $i$ and the worker $j$ are connected. Otherwise, they are disconnected. Moreover, we use $\nabla_i$ to denote the gradient with respect to the $i$-th variable, where $i \in \{1, 2\}$, and use $\bar{a} = \frac{1}{K} \sum_{k=1}^{K} a^{(k)}$ to denote the averaged variable across workers for any variable $a$. In addition, we denote $X_t = [x_t^{(1)}, \cdots, x_t^{(K)}] \in \mathbb{R}^{d_x \times K}$, where $x_t^{(k)}$ denotes the variable $x$ on the $k$-th worker at the $t$-th iteration. Similarly, we define $Y_t = [y_t^{(1)}, \cdots, y_t^{(K)}] \in \mathbb{R}^{d_y \times K}$ and $Z_t = [z_t^{(1)}, \cdots, z_t^{(K)}] \in \mathbb{R}^{d_y \times K}$. We further introduce $M_{x,1,t} = [m_{x,1,t}^{(1)}, \cdots, m_{x,1,t}^{(K)}] \in \mathbb{R}^{d_x \times K}$, where $m_{x,1,t}^{(k)}$ denotes the estimator for $\nabla_1 f^{(k)}(x_t^{(k)}, y_t^{(k)})$ on the $k$-th worker at the $t$-th iteration. Likewise, we define $M_{x,2,t} = [m_{x,2,t}^{(1)}, \cdots, m_{x,2,t}^{(K)}] \in \mathbb{R}^{d_x \times K}$, where $m_{x,2,t}^{(k)}$ denotes the estimator for $\nabla_1 g^{(k)}(x_t^{(k)}, y_t^{(k)})$, and $M_{x,3,t} = [m_{x,3,t}^{(1)}, \cdots, m_{x,3,t}^{(K)}] \in \mathbb{R}^{d_x \times K}$ where $m_{x,3,t}^{(k)}$ denotes the estimator for $\nabla_1 g^{(k)}(x_t^{(k)}, z_t^{(k)})$. Similarly, we introduce $M_{y,1,t} = [m_{y,1,t}^{(1)}, \cdots, m_{y,1,t}^{(K)}] \in \mathbb{R}^{d_y \times K}$ and $M_{y,2,t} = [m_{y,2,t}^{(1)}, \cdots, m_{y,2,t}^{(K)}] \in \mathbb{R}^{d_y \times K}$, where $m_{y,1,t}^{(k)}$ denotes the estimator for $\nabla_2 f^{(k)}(x_t^{(k)}, y_t^{(k)})$, and $m_{y,2,t}^{(k)}$ denotes the estimator for $\nabla_2 g^{(k)}(x_t^{(k)}, y_t^{(k)})$. Finally, we define $M_{z,1,t} = [m_{z,1,t}^{(1)}, \cdots, m_{z,1,t}^{(K)}] \in \mathbb{R}^{d_y \times K}$, where $m_{z,1,t}^{(k)}$ denotes the estimator for $\nabla_2 g^{(k)}(x_t^{(k)}, z_t^{(k)})$. Finally, the *two-timescale* learning rate indicates the ratio of the two learning rates, $\eta_{x,t}$ and $\eta_{y,t}$, approaches to zero or infinity, e.g., $\lim_{t \to \infty} \eta_{x,t}/\eta_{y,t} = 0$. The *single-timescale* learning rate indicates that their ratio converges to a positive constant, e.g., $\lim_{t \to \infty} \eta_{x,t}/\eta_{y,t} = O(1)$.

## 3.2. Our Algorithm

To solve Eq. (5), we develop a novel single-loop decentralized stochastic first-order gradient descent algorithm (FO-DSVRBGD) in Algorithm 1. The key idea is to compute the STORM gradient estimator (Cutkosky & Orabona, 2019) on each worker $k \in \{1, 2, \cdots, K\}$ for the deterministic gradient and then use it to update each variable. For example, we compute the gradient estimator $m_{x,1,t}^{(k)}$ for the

**Algorithm 1** Decentralized stochastic first-order gradient descent algorithm (FO-DSVRBGD)

---

**Input:** $\eta_x > 0$, $\eta_y > 0$, $\eta_z > 0$, $\gamma_x > 0$, $\gamma_y > 0$, $\gamma_z > 0$, $\rho > 0$. Initialization on each worker $k \in \{1, \cdots, K\}$:

$x_0^{(k)} = x_0, \qquad y_0^{(k)} = y_0, \qquad z_0^{(k)} = z_0,$

$m_{x,1,0}^{(k)} = \nabla_1 f^{(k)}(x_0^{(k)}, y_0^{(k)}; \xi_0^{(k)}),$

$m_{x,2,0}^{(k)} = \nabla_1 g^{(k)}(x_0^{(k)}, y_0^{(k)}; \zeta_0^{(k)}),$

$m_{x,3,0}^{(k)} = \nabla_1 g^{(k)}(x_0^{(k)}, z_0^{(k)}; \zeta_0^{(k)}),$

$m_{y,1,0}^{(k)} = \nabla_2 f^{(k)}(x_0^{(k)}, y_0^{(k)}; \xi_0^{(k)}),$

$m_{y,2,0}^{(k)} = \nabla_2 g^{(k)}(x_0^{(k)}, y_0^{(k)}; \zeta_0^{(k)}),$

$m_{z,1,0}^{(k)} = \nabla_2 g^{(k)}(x_0^{(k)}, z_0^{(k)}; \zeta_0^{(k)}),$

$U_{x,0} = M_{x,0}, \qquad U_{y,0} = M_{y,0}, \qquad U_{z,0} = M_{z,0}.$

1: **for** $t = 0, \cdots, T-1$ **do**
2:     Update three variables:
    $X_{t+1} = X_t W - \eta_x U_{x,t}$ ,
    $Y_{t+1} = Y_t W - \eta_y U_{y,t}$ ,
    $Z_{t+1} = Z_t W - \eta_z U_{z,t}$ ,
3:     Compute three gradient estimators:
    $M_{x,t+1} = M_{x,1,t+1} + \frac{1}{\rho}(M_{x,2,t+1} - M_{x,3,t+1})$ ,
    $M_{y,t+1} = M_{y,1,t+1} + \frac{1}{\rho} M_{y,2,t+1}$ ,
    $M_{z,t+1} = \frac{1}{\rho} M_{z,1,t+1}$ ,
4:     Perform gradient tracking:
    $U_{x,t+1} = U_{x,t} W + M_{x,t+1} - M_{x,t}$ ,
    $U_{y,t+1} = U_{y,t} W + M_{y,t+1} - M_{y,t}$ ,
    $U_{z,t+1} = U_{z,t} W + M_{z,t+1} - M_{z,t}$ ,
5: **end for**

---

deterministic $\nabla_1 f^{(k)}(x_t^{(k)}, y_t^{(k)})$ as follows:

$$
\begin{aligned}
m_{x,1,t}^{(k)} = {} & (1 - \gamma_x \eta_x^2)(m_{x,1,t-1}^{(k)} - \nabla_1 f^{(k)}(x_{t-1}^{(k)}, y_{t-1}^{(k)}; \xi_t^{(k)})) \\
& + \nabla_1 f^{(k)}(x_t^{(k)}, y_t^{(k)}; \xi_t^{(k)}) ,
\end{aligned} \tag{6}
$$

where $\gamma_x > 0$ is a hyperparameter, $\eta_x > 0$ is the learning learning for updating $x$, and $\gamma_x \eta_x^2 < 1$. We use the same approach and hyperparameters to compute $m_{x,2,t}^{(k)}$ and $m_{x,3,t}^{(k)}$, which estimate $\nabla_1 g^{(k)}(x_t^{(k)}, y_t^{(k)})$ and $\nabla_1 g^{(k)}(x_t^{(k)}, z_t^{(k)})$, respectively. Similarly, we compute the STORM gradient estimators $m_{y,1,t}^{(k)}$ and $m_{y,2,t}^{(k)}$ for the deterministic gradient $\nabla_2 f^{(k)}(x_t^{(k)}, y_t^{(k)})$ and $\nabla_2 g^{(k)}(x_t^{(k)}, z_t^{(k)})$, respectively, using the hyperparameter $\gamma_y > 0$, $\eta_y > 0$, and $\gamma_y \eta_y^2 < 1$. In addition, we compute the STORM gradient estimator $m_{z,1,t}^{(k)}$ for the deterministic gradient $\nabla_2 g^{(k)}(x_t^{(k)}, z_t^{(k)})$ with the hyperparameter $\gamma_z > 0$, $\eta_z > 0$, and $\gamma_z \eta_z^2 < 1$. With the aforementioned gradient estimators for each component, we can construct the gradient estimators with respect to each variable as described in Step 3 of Algorithm 1. Then, we apply the gradient tracking communication strategy to communicate gradients as shown in Step 4 of Algorithm 1. Finally, we can update variables according to Step 2 of Algorithm 1.

### 3.3. Key Innovations in Algorithm 1

Compared to the existing single-agent method (Kwon et al., 2024b), our algorithm introduces several key innovations, as outlined below.

Firstly, (Kwon et al., 2024b) introduces a regularization term for the optimization problem with respect to $y$ and $z$ such that their loss functions are strongly convex. For example, to update the variable $z$, (Kwon et al., 2024b) proposed to solve the following strongly convex optimization problem

$$
\min_{z \in \mathbb{R}^{d_y}} g(x, z) + \frac{1}{2\beta} \|z - w\|^2 , \tag{7}
$$

where $w$ is an auxiliary variable, $\beta > 0$ is hyperparameter such that $g(x, z) + \frac{1}{2\beta} \|z - w\|^2$ is strongly convex with respect to $z$. This approach introduces additional variables and their associated updates, leading to increased computational overhead. In contrast, *our algorithm does NOT introduce such a regularization term and directly uses the gradient estimator of $\nabla_2 g(x, z)$ to update the variable $z$.*

Secondly, (Kwon et al., 2024b) employs a time-dependent learning rate and a two-timescale schedule for different variables. For example, its learning rate used to update the variable $x$ is $O(\frac{1}{t^{2/5}})$. Moreover, as shown in Algorithm 2 and Corollary 5.5 in (Kwon et al., 2024b), the actual learning rate for updating the variable $y$ is $\beta_t \gamma_t = O(\frac{1}{t^{2/5}} \frac{1}{t^{2/5}})$ while that for $x$ is $O(\frac{1}{t^{2/5}})$. It is easy to see that $\lim_{t \to \infty} t^{-4/5}/t^{-2/5} = 0$, indicating the two-timescale learning schedule. This kind of time-dependent learning rate can potentially leak training stage information to malicious workers in the decentralized setting, and the two-timescale schedule is difficult to tune in practice. In contrast, the learning rates for all variables in our Algorithm 1 are constant, independent of the current iteration, and share the same order dependence on the solution accuracy $\epsilon$. Specifically, $\eta_x$, $\eta_y$, and $\eta_z$ are **of the same scale in terms of the solution accuracy $\epsilon$** as shown in Theorem 4.6. Therefore, it is much easier to tune the learning rate for our algorithm.

These key innovations make our Algorithm 1 significantly different from existing single-agent methods (Kwon et al., 2024b; Chen et al., 2024; Shen & Chen, 2023). Hence, we have to develop new approaches to establish the convergence rate of our Algorithm 1.

## 4. Convergence Analysis

### 4.1. Assumptions

To establish the convergence rate of our algorithm, we introduce the following assumptions, which are also used in the existing nonconvex bi-level optimization literature

(Kwon et al., 2024b; Chen et al., 2024).

**Assumption 4.1.** For $k \in \{1, \cdots, K\}$, the upper-level loss function $f^{(k)}(\cdot, \cdot)$ satisfies the mean-squared smoothness with the constant $L_f > 0$. In addition, $\|\nabla_2 f^{(k)}(\cdot, \cdot)\| \leq C_f$, where $C_f > 0$ is a constant. Moreover, the second-order gradients of $f^{(k)}(\cdot, \cdot)$ satisfy the Lipschitz continuity with the constant $\ell_f > 0$.

**Assumption 4.2.** For $k \in \{1, \cdots, K\}$, the lower-level loss function $g^{(k)}(\cdot, \cdot)$ satisfies the mean-squared smoothness with the constant $L_g > 0$. The second-order gradients of $g^{(k)}(\cdot, \cdot)$ satisfy the Lipschitz continuity with the constant $\ell_g > 0$.

**Assumption 4.3.** For $n \in \{1, \cdots, N\}$, the lower-level function $g^{(k)}(\cdot, \cdot)$ is $\mu$-PL with respect to the second variable, where $\mu > 0$ is a constant. In addition, the penalty function $h_\rho(\cdot, \cdot) = \rho f(\cdot, \cdot) + g(\cdot, \cdot)$ is $\mu$-PL with respect to the second variable.

**Assumption 4.4.** The stochastic gradients of the upper-level and lower-level loss functions on each worker $k \in \{1, \cdots, K\}$ have an upper bounded variance $\sigma^2$ where $\sigma > 0$ is a constant.

**Assumption 4.5.** The adjacency matrix $W$ is doubly stochastic and symmetric. In addition, its eigenvalues satisfy $|\lambda_K| \leq |\lambda_{K-1}| \leq \cdots \leq |\lambda_2| < |\lambda_1| = 1$.

In terms of Assumption 4.5, we denote the spectral gap of $W$ by $1 - \lambda$ where $\lambda = |\lambda_2|$. Additionally, we denote $\ell = \max\{L_f, L_g, \ell_f, \ell_g\}$ and $\kappa = \ell/\mu$.

### 4.2. Convergence Rate

Based on Assumptions 4.1-4.5 and the fundamental Lemmas A.14-A.16 that characterize how $\mathcal{L}_\rho(x)$ approximates $\mathcal{L}(x)$ and the tractability of $\min_{x \in \mathbb{R}^{d_x}} \mathcal{L}_\rho(x)$, we establish the convergence rate of Algorithm 1 in Theorem 4.6.

**Theorem 4.6.** *Suppose Assumptions 4.1-4.5 hold, when*
$\gamma_x = O\left(\frac{\kappa^2}{(1-\lambda)^4 \rho^2 K}\right)$, $\gamma_y = O\left(\frac{1}{(1-\lambda)^4 \rho^2 K}\right)$, $\gamma_z = O\left(\frac{1}{(1-\lambda)^4 \rho^2 K}\right)$, $\eta_x = O\left(\frac{K(1-\lambda)^2 \epsilon^3}{\kappa^9}\right)$, $\eta_y = O\left(\frac{K(1-\lambda)^2 \epsilon^3}{\kappa^7}\right)$, $\eta_z = O\left(\frac{K(1-\lambda)^2 \epsilon^3}{\kappa^7}\right)$, $S = O\left(\frac{\kappa^7}{\epsilon^3}\right)$, $T = O\left(\frac{\kappa^9}{K(1-\lambda)^2 \epsilon^5}\right)$, $\rho = O\left(\frac{\epsilon}{\kappa^3}\right)$, *Algorithm 1 can achieve the $\epsilon$-accuracy solution:* $\frac{1}{T}\sum_{t=0}^{T-1} \mathbb{E}[\|\nabla \mathcal{L}(\bar{x}_t)\|^2] \leq O(\epsilon^2)$, *where $\epsilon > 0$, $S$ is the batch size in the initial iteration.*

*Remark* 4.7. Theorem 4.6 shows that the learning rates $\eta_x$, $\eta_y$, and $\eta_z$ for three variables are of the same scale, in contrast to the two-timescale schedule used in (Kwon et al., 2024b). Moreover, unlike (Kwon et al., 2024b), our learning rates are not time-dependent. Therefore, our learning rate strategy is fundamentally different from that in (Kwon et al., 2024b).

*Remark* 4.8. When the number of workers is 1, i.e., $K = 1$ and $1 - \lambda = 1$, our iteration complexity $T = O\left(\frac{\kappa^9}{\epsilon^5}\right)$

can match the learning rate in (Kwon et al., 2024b) in the single-agent setting in terms of $\epsilon$. Moreover, in terms of the condition number $\kappa$, our convergence rate $O(\kappa^9)$ is better than that established in (Chen et al., 2024), i.e., $O(\kappa^{12})$ where a standard stochastic gradient is used for update.

*Remark* 4.9. Theorem 4.6 indicates that the communication complexity of Algorithm 1 is $O\left(\frac{\kappa^9}{K(1-\lambda)^2 \epsilon^5}\right)$. The dependence over the spectral gap is in the order of $O\left(\frac{1}{(1-\lambda)^2}\right)$, which is consistent with existing second-order methods (Gao et al., 2023). Moreover, if ignoring the spectral gap, this complexity indicates a linear speedup with respect to the number of workers $K$, which is consistent with existing second-order methods (Gao et al., 2023; Zhang et al., 2026b).

To the best of our knowledge, this is the first time to achieve such a favorable convergence rate for the decentralized nonconvex bi-level optimization problem.

### 4.3. Key Innovations in Convergence Analysis

As mentioned in Subsection 3.3, the existing single-agent method (Kwon et al., 2024b) introduces a regularization term to make the loss function with respect to the variable $y$ and $z$ strongly convex. *This allows (Kwon et al., 2024b) to apply techniques developed for strongly convex optimization in their convergence analysis.* For example, when analyzing the optimization error for the subproblems involving $y$ and $z$, (Kwon et al., 2024b) bounds the optimization errors $\|y_t - y_\rho^*(x)\|^2$ and $\|z_t - y^*(x)\|^2$ using methods tailored for strongly convex objectives (see Lemma D.3 in (Kwon et al., 2024b)). In contrast, since our Algorithm 1 does not impose artificial strong convexity, the approach in (Kwon et al., 2024b) is not applicable to our setting, where the loss function with respect to $y$ and $z$ is nonconvex but satisfies only the PL condition.

**Novel lemmas for characterizing the optimization error with respect to the variable $y$ and $z$ .** To establish the convergence rate of our Algorithm 1 without imposing the artificial strong convexity, we propose a novel approach for measuring the optimality of $\bar{y}_t$ and $\bar{z}_t$ in the $t$-th iteration. Specifically, we measure the optimality of $\bar{y}_t$ and $\bar{z}_t$ in the $t$-th iteration based on the following two metrics:

$$\frac{1}{\rho}\mathbb{E}[h_\rho(\bar{x}_t, \bar{y}_t) - \hat{h}_\rho(\bar{x}_t)] ,$$
$$\frac{1}{\rho}\mathbb{E}[g(\bar{x}_t, \bar{z}_t) - \hat{g}(\bar{x}_t)] ,$$
(8)

where $\hat{h}_\rho(x) \triangleq h_\rho(x, y_\rho^*(x))$ with $y_\rho^*(x) = \arg\min_y h_\rho(x, y)$ and $\hat{g}(x) \triangleq g(x, y^*(x))$ with $y^*(x) = \arg\min_z g(x, z)$. Then, we investigate how these two metrics evolve across iterations. Specifically, in Lemma A.3, we prove that $\hat{h}_\rho(x)$ and $\hat{g}(x)$ are smooth.

Based on that, we develop two novel lemmas for these metrics in Lemma 4.10 and Lemma 4.11 without relying on strong convexity.

**Lemma 4.10.** *Suppose Assumptions 4.1-4.5 hold, when* $\eta_x \leq \eta_y \frac{\mu^2}{4L_{\hat{h}_\rho}^2}$ *and* $\eta_y \leq \frac{\rho}{2L_{h_\rho}}$, *where* $L_{h_\rho} = \rho L_f + L_g$, *we have that*

$$
\frac{1}{\rho}\mathbb{E}[h_\rho(\bar{x}_{t+1}, \bar{y}_{t+1}) - \hat{h}_\rho(\bar{x}_{t+1})] \leq \frac{1}{\rho}\mathbb{E}[h_\rho(\bar{x}_t, \bar{y}_t) - \hat{h}_\rho(\bar{x}_t)]
$$

$$
- \frac{1}{\rho^2}\frac{\eta_y}{8}\mathbb{E}[\|\nabla_2 h_\rho(\bar{x}_t, \bar{y}_t)\|^2] - \frac{\eta_y}{4}\mathbb{E}[\|\bar{m}_{y,t}\|^2]
$$

$$
+ \left(\frac{\eta_x}{2} + \frac{1}{\rho}\frac{\eta_x^2 L_{h_\rho}}{2} + \frac{1}{\rho}\frac{3\eta_x^2 L_{h_\rho}}{2} + \frac{1}{\rho}\frac{\eta_x^2 L_{\hat{h}_\rho}}{2}\right)\mathbb{E}[\|\bar{m}_{x,t}\|^2]
$$

$$
+ 2\eta_y \frac{L_{h_\rho}^2}{\rho^2}\frac{1}{K}\mathbb{E}[\|X_t - \bar{X}_t\|_F^2] + 2\eta_y \frac{L_{h_\rho}^2}{\rho^2}\frac{1}{K}\mathbb{E}[\|Y_t - \bar{Y}_t\|_F^2]
$$

$$
+ 4\eta_y \mathbb{E}[\|\frac{1}{K}\sum_{k=1}^{K}\nabla_2 f^{(k)}(x_t^{(k)}, y_t^{(k)}) - \frac{1}{K}\sum_{k=1}^{K}m_{y,1,t+1}^{(k)}\|^2]
$$

$$
+ 4\eta_y \frac{1}{\rho^2}\mathbb{E}[\|\frac{1}{K}\sum_{k=1}^{K}\nabla_2 g^{(k)}(x_t^{(k)}, y_t^{(k)}) - \frac{1}{K}\sum_{k=1}^{K}m_{y,2,t+1}^{(k)}\|^2].
$$

**Lemma 4.11.** *Suppose Assumptions 4.1-4.5 hold, when* $\eta_x \leq \frac{\mu^2}{4L_g^2}\eta_z$ *and* $\eta_z \leq \frac{\rho}{2L_g}$, *we have that*

$$
\frac{1}{\rho}\mathbb{E}[g(\bar{x}_{t+1}, \bar{z}_{t+1}) - \hat{g}(\bar{x}_{t+1})] \leq \frac{1}{\rho}\mathbb{E}[g(\bar{x}_t, \bar{z}_t) - \hat{g}(\bar{x}_t)]
$$

$$
- \frac{1}{\rho^2}\frac{\eta_z}{8}\mathbb{E}[\|\nabla_2 g(\bar{x}_t, \bar{z}_t)\|^2] - \frac{\eta_z}{4}\mathbb{E}[\|\bar{m}_{z,t}\|^2]
$$

$$
+ \left(\frac{\eta_x}{2} + 2\eta_x^2 L_g \frac{1}{\rho} + \frac{\eta_x^2 L_{\hat{g}}}{2}\frac{1}{\rho}\right)\mathbb{E}[\|\bar{m}_{x,t}\|^2]
$$

$$
+ 2\eta_z \frac{L_g^2}{\rho^2}\frac{1}{K}\mathbb{E}[\|X_t - \bar{X}_t\|_F^2] + 2\eta_z \frac{L_g^2}{\rho^2}\frac{1}{K}\mathbb{E}[\|Z_t - \bar{Z}_t\|_F^2]
$$

$$
+ \frac{2\eta_z}{\rho^2}\mathbb{E}[\|\frac{1}{K}\sum_{k=1}^{K}\nabla_2 g^{(k)}(x_t^{(k)}, z_t^{(k)}) - \frac{1}{K}\sum_{k=1}^{K}m_{z,1,t}^{(k)}\|^2].
$$

From the above two lemmas, it can be seen that there are gradient estimation errors like $\mathbb{E}[\|\frac{1}{K}\sum_{k=1}^{K}\nabla_1 f^{(k)}(x_t^{(k)}, y_t^{(k)}) - \frac{1}{K}\sum_{k=1}^{K}m_{x,1,t}^{(k)}\|^2]$ and consensus errors like $\mathbb{E}[\|X_t - \bar{X}_t\|_F^2]$. Therefore, to handle these errors, we establish the upper bound for the consensus error in Lemma A.10 and Lemma A.11 and the gradient estimation error in Lemma A.8 and Lemma A.9.

**Novel potential function.** To prove Theorem 4.6, we develop a novel potential function in Eq. (12), which includes $\mathbb{E}[\mathcal{L}(\bar{x}_t)]$, *the above two metrics*, *the consensus error*, and *the gradient estimation error*. A novel design for this potential function is that the coefficients $\{c_9, c_{10}, c_{12}, c_{13}, c_{15}, c_{16}, c_{18}, c_{19}\}$ are explicitly attached by $\frac{1}{\rho^2}$. As a result, these coefficients themselves are independent of $\frac{1}{\rho}$, as shown in Eq. (82). Then, these coefficients will not affect the dependence on $\frac{1}{\rho}$ for the variance terms

in Eq. (76). Otherwise, it might lead to a higher-order dependence, degenerating the convergence rate.

To summarize, we develop a new approach to establish the convergence rate of our algorithm for decentralized nonconvex bi-level optimization problems, which is significantly different from existing studies (Kwon et al., 2024b; Chen et al., 2024). In fact, our proof strategy is general and easy to understand. We believe that it can also be applied to the single-agent setting.

# 5. Experiments

## 5.1. Experiment Setup

In our experiments, we apply Algorithm 1 to a hyperparameter optimization task, formally defined as:

$$
\min_x \frac{1}{K}\sum_{k=1}^{K}\frac{1}{m^{(k)}}\sum_{i=1}^{m^{(k)}}\ell(y^*(x); a_{v,i}^{(k)}, b_{v,i}^{(k)})
$$

$$
s.t.\ y^*(x) = \arg\min_y \frac{1}{K}\sum_{k=1}^{K}\frac{1}{m^{(k)}}\sum_{i=1}^{m^{(k)}}\ell(y; a_{t,i}^{(k)}, b_{t,i}^{(k)})
$$

$$
+ \frac{1}{d_1 d_2}\sum_{p=1}^{d_1}\sum_{q=1}^{d_2}\exp(x_{1,q})y_{1,pq}^2 + \frac{1}{d_2 d_3}\sum_{p=1}^{d_2}\sum_{q=1}^{d_3}\exp(x_{2,q})y_{2,pq}^2,
$$

$$
\tag{9}
$$

where $(a_{v,i}^{(k)}, b_{v,i}^{(k)})$ denotes the $i$-th validation sample's feature and label on the $k$-th worker, $(a_{t,i}^{(k)}, b_{t,i}^{(k)})$ denotes the $i$-th training sample's feature and label, $\ell$ denotes the cross-entropy loss function. Specifically, the lower-level optimization problem is to learn the weight $y = \{y_1, y_2\}$ of a two-layer neural network, where $y_1 \in \mathbb{R}^{d_1 \times d_2}$ denotes the weight of the first layer, $y_2 \in \mathbb{R}^{d_2 \times d_3}$ denotes the weight of the second layer, and $d_2 = 100$ in our experiment. The upper-level optimization problem is to learn the regularization coefficient $x = \{x_1, x_2\}$. The existing study (Liu et al., 2022a) has shown that an over-parameterized neural network can satisfy the PL condition. Therefore, the lower-level loss function in Eq. (9) can satisfy the PL condition, and the upper-level one is non-convex.

To evaluate the performance of our algorithm, we use three LIBSVM benchmark datasets [1]: a9a, w8a, covtype, and an image dataset: MNIST (Lecun et al., 1998). For the first three datasets, we allocate 10% of samples as a test set, 70% of the remaining samples for training, and use the rest as a validation set. Regarding baseline methods, since all existing state-of-the-art decentralized bi-level optimization algorithms (Chen et al., 2025b; Yang et al., 2022; Gao et al., 2023; Chen et al., 2023; Dong et al., 2025; Zhang et al., 2026b; Kong et al., 2025; Zhu et al., 2024) are designed for the problem with a strongly-convex lower-level subprob-

---

[1] https://www.csie.ntu.edu.tw/~cjlin/libsvmtools/datasets/

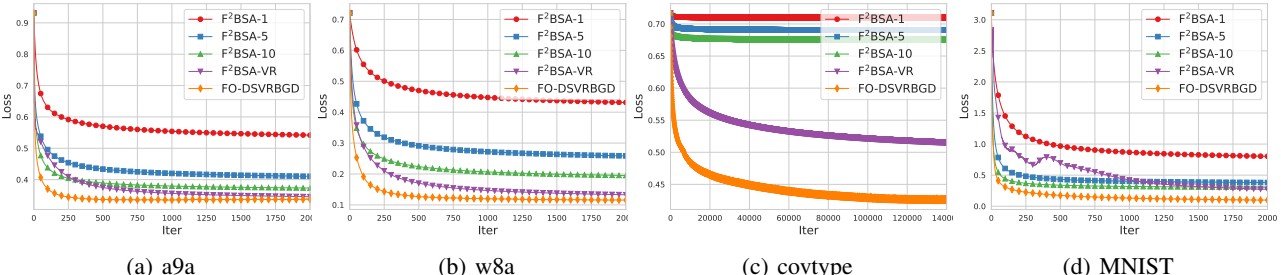

(a) a9a      (b) w8a      (c) covtype      (d) MNIST

*Figure 1.* The upper-level loss function value with respect to the number of iterations.

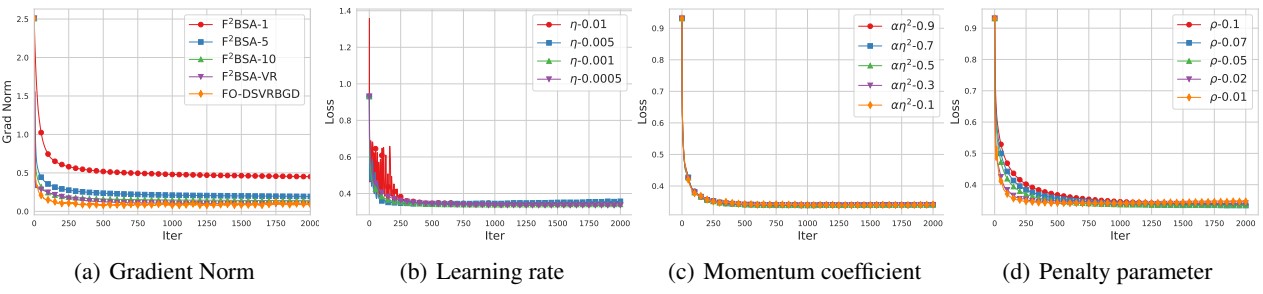

(a) Gradient Norm      (b) Learning rate      (c) Momentum coefficient      (d) Penalty parameter

*Figure 2.* (a) The gradient norm of upper-level loss function versus iterations for a9a dataset. (b-c) The upper-level loss function value versus iterations under different settings for a9a dataset.

lem, they cannot be used to solve the non-convex bi-level optimization problem in Eq. (9). Therefore, we compare our algorithm with $F^2$BSA (Kwon et al., 2024b), which is designed for the single-machine setting. We consider both single-loop and double-loop variants of $F^2$BSA. For the double-loop approach, we evaluate versions with one, five, and ten inner iterations, denoted as $F^2$BSA-1, $F^2$BSA-5, and $F^2$BSA-10, respectively. The single-loop variant, $F^2$BSA-VR, incorporates the same variance reduction technique as our algorithm. To ensure a fair comparison, we run all baseline methods in a decentralized setting similar to ours, using a complete communication graph with the all-reduce operation, since using a non-complete graph could potentially degrade the performance of these single-agent methods.

### 5.2. Result and Analysis

In Figure 1, we report the upper level loss function value with respect to the number of iterations. Specifically, we use eight workers in this experiment and we use a ring graph to connect these workers. Moreover, the solution accuracy $\epsilon$ is set to $0.1$. Then, according to Theorem 4.6, we set the learning rate of our algorithm as $\eta_x = \eta_y = \eta_z = \epsilon^3$, the momentum coefficient $\gamma_x \eta_x^2 = \gamma_y \eta_y^2 = \gamma_z \eta_z^2 = 0.9$. The penalty parameter is set to $\rho = 0.2\epsilon$. For all variants of $F^2$BSA, we set their learning rates and penalty parameter according to Corollary 5.2 and 5.5 in (Kwon et al., 2024b). In addition, the batch size is set to 100. From Fig-

ure 1, we can find that our algorithm converges much faster than baseline methods, confirming the effectiveness of our Algorithm 1. In Figure 2(a), we report the gradient norm of the upper-level loss function versus the number of iterations, which also confirms that our algorithm converges faster than baselines.

**Hyperparameter.** To examine the influence of different hyperparameters, including the learning rate $\eta$, the momentum coefficient $\alpha\eta^2$, and the penalty parameter $\rho$, on our algorithm, we performed three additional experiments with the a9a dataset. Figure 2(b) shows the loss function value when using different values for $\eta$. It can be seen that a larger $\eta$ leads to faster convergence but may cause instability. Figure 2(c) shows that the momentum coefficient does not affect the convergence speed of our algorithm too much. Additionally, Figure 2(d) shows that a smaller $\rho$, i.e., a larger penalty $1/\rho$, can accelerate the convergence of our algorithm.

### 5.3. More Applications on Model Pruning

To further evaluate the performance of our algorithm, we apply it to the model pruning task (Zhang et al., 2022), which involves pruning certain parameters of a deep neural network by learning a binary mask. Specifically, the lower-level optimization problem focuses on learning the parameters of the pruned network, while the upper-level optimization problem is responsible for learning the prun-

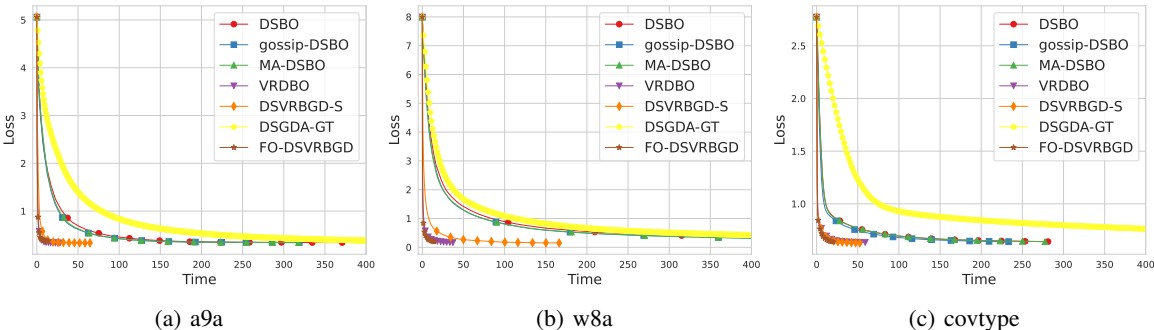

| (a) a9a | (b) w8a | (c) covtype |
|---------|---------|-------------|

*Figure 3.* The upper-level loss function value the nonconvex-strongly-convex bilevel problem (hyperparameter optimization task) with respect to the time consumed (seconds).

ing mask. Formally, its loss function is defined as:

$$\min_x \frac{1}{K} \sum_{k=1}^{K} \ell^{(k)}(x \odot y^*(x))$$

$$s.t. \quad y^*(x) = \arg\min_y \frac{1}{K} \sum_{k=1}^{K} \ell^{(k)}(x \odot y), \qquad (10)$$

where $y \in \mathbb{R}^d$ is the parameters of the deep neural network, $x \in \{0,1\}^d$ denotes the binary pruning mask, with a value of 0 indicating that the corresponding parameter in $y$ is pruned.

In our experiment, we conduct model pruning on ResNet20, where the dataset is CIFAR10. Obviously, this deep neural network is overparameterized compared to the number of raw feature in CIFAR10. Thus, the lower-level

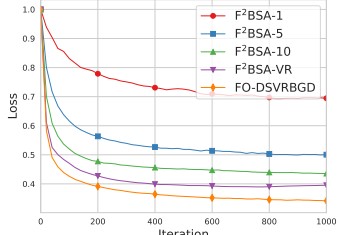

*Figure 4.* The upper-level loss function value with respect to the number of iterations (model pruning task) with ResNet20 and CIFAR10.

loss function satisfies the PL condition according to the existing study (Liu et al., 2022a). Then, we still use our algorithm to solve Eq. (10), where $y$ is the parameters of ResNet20 and $x$ is the pruning mask. In this experiment, we use the same experimental setup as the first experiment, and we only keep 20% parameters. The loss function value versus the number of iterations is reported in Figure 4. We can observe that our algorithm converges faster than baseline methods, which further confirms the efficacy of our algorithm.

### 5.4. More Applications in the Strongly Convex Setting

Since the strongly convexity indicates the PL condition, we apply our algorithm to the nonconvex-strongly-convex hy-

perparameter optimization problem to further demonstrate its performance. In particular, we change the two-layer neural network in Eq. (9) into a linear logistic regression model. Then, we compare our algorithm with those that require second-order gradients: DSBO (Chen et al., 2025b), gossip-DSBO (Yang et al., 2022), MA-DSBO (Chen et al., 2023), VRDBO (Gao et al., 2023), DSVRBGD-S (Zhang et al., 2026b), and those using fully first-order gradients: DSFDA-GT (Wang et al., 2025). We use the same way to set the learning rate for our algorithm and DSFDA-GT in terms of the theoretical result in (Wang et al., 2025). As for other baseline methods, we also set the learning rate according to their theoretical results, i.e., using $\epsilon^2$ as the learning rate for DSBO, gossip-DSBO, and MA-DSBO and using $\epsilon$ for VRDBO and DSVRBGD-S. The other settings are the same as the first experiment. From Figure 3, we can find that our algorithm consumed much less time to convergence than all the baseline methods that require second-order Jacobian and Hessian matrices, which confirms the efficacy of using fully first-order gradients. In addition, our algorithm outperforms DSFDA-GT that uses the fully first-order gradient, which confirms the efficacy of using variance-reduced gradients.

In summary, these experimental results across different applications confirm the efficacy of our algorithm.

## 6. Conclusions

In this paper, we develop a novel decentralized stochastic first-order method for nonconvex bi-level optimization problems without introducing the artificial strong convexity for the lower-level loss function as existing methods, where a novel single-timescale constant learning rate schedule is proposed. Moreover, we established its convergence rate, where a new proof strategy is proposed to handle the pure PL condition. The extensive experimental results confirm the effectiveness of our algorithm.

## Acknowledgements

We thank anonymous reviewers for constructive comments. Y. Zhang, X. Zhang, and H. Gao was partially supported by U.S. NSF CAREER 2339545, NSF IIS 2416607, NSF CNS 2107014. M. Thai was partially supported by U.S. NSF IIS 2416606.

## Impact Statement

This paper presents work whose goal is to advance the field of Machine Learning. There are many potential societal consequences of our work, none which we feel must be specifically highlighted here.

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

# A. Appendix

## A.1. More Applications on Toy Model

To further verify the performance of our algorithm, we conducted an additional experiment on a synthetic model, which has a lower-level problem that is nonconvex but satisfies the PL condition. Specifically, following Eq.(12) in (Shen & Chen, 2023), we construct the following stochastic bilevel optimization problem:

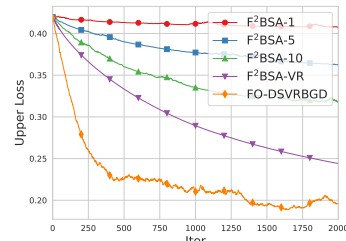

$$\min_{x\in[0,3],y} \frac{1}{K}\sum_{k=1}^{K}\frac{1}{m}\sum_{m=1}^{M}\left(a_m^{(k)}\frac{cos(4y+2)}{1+\exp^{2-4x}}+b_m^{(k)}\frac{1}{2}\ln((4x-2)^2+1)\right)$$

$$s.t. \quad y\in\arg\min_{y}\frac{1}{K}\sum_{k=1}^{K}\frac{1}{n}\sum_{n=1}^{N}\left((y+x-c_n^{(k)})^2+x\sin^2(y+x-c_n^{(k)})\right).$$

$$(11)$$

*Figure 5.* The upper-level loss function value with respect to the number of iterations on toy model.

According to (Shen & Chen, 2023), it is easy to know the lower-level loss function is still nonconvex but satisfies the PL condition because our variant just adds constant terms to it.

In our experiment, we generate $a_m$, $b_m$, and $c_n$ from a standard Gaussian distribution. Then, we use the same experimental setting as other experiments. Figure 5 shows the upper-level loss function value in different iterations. We can find that our algorithm still outperforms all baselines for this toy model, which further verifies our theoretical results.

## A.2. Proof Sketch

To establish the convergence rate of Algorithm 1, we establish the following potential function:

**Step 1:** To prove Theorem 4.6, we propose a novel potential function, which is defined as follows:

$$P_t = \mathbb{E}[\mathcal{L}(\bar{x}_t)] \qquad\qquad \text{Lemmas A.5}$$

$$+ \mathbb{E}[\frac{c_0\eta_x}{\eta_y}\frac{1}{\rho}\mathbb{E}[h_\rho(\bar{x}_t,\bar{y}_t)-\hat{h}_\rho(\bar{x}_t)] \qquad\qquad \text{Lemmas A.7}$$

$$+ \frac{c_1\eta_x}{\eta_z}\frac{1}{\rho}\mathbb{E}[g(\bar{x}_t,\bar{z}_t)-\hat{g}(\bar{x}_t)] \qquad\qquad \text{Lemmas A.6}$$

$$+ c_2\frac{1}{K}\mathbb{E}[\|X_t-\bar{X}_t\|_F^2]+c_3\frac{1}{K}\mathbb{E}[\|Y_t-\bar{Y}_t\|_F^2]+c_4\frac{1}{K}\mathbb{E}[\|Z_t-\bar{Z}_t\|_F^2] \qquad \text{Lemma A.10}$$

$$+ c_5\frac{1}{K}\mathbb{E}[\|U_{x,t}-\bar{U}_{x,t}\|_F^2]+c_6\frac{1}{K}\mathbb{E}[\|U_{y,t}-\bar{U}_{y,t}\|_F^2]+c_7\frac{1}{K}\mathbb{E}[\|U_{z,t}-\bar{U}_{z,t}\|_F^2] \qquad \text{Lemma A.11}$$

$$+ c_8\mathbb{E}[\|\frac{1}{K}\sum_{k=1}^{K}\nabla_1 f^{(k)}(x_t^{(k)},y_t^{(k)})-\frac{1}{K}\sum_{k=1}^{K}m_{x,1,t}^{(k)}\|^2] \qquad\qquad \text{Lemma A.8}$$

$$+ c_9\frac{1}{\rho^2}\mathbb{E}[\|\frac{1}{K}\sum_{k=1}^{K}\nabla_1 g^{(k)}(x_t^{(k)},y_t^{(k)})-\frac{1}{K}\sum_{k=1}^{K}m_{x,2,t}^{(k)}\|^2] \qquad\qquad \text{Lemma A.8}$$

$$+ c_{10}\frac{1}{\rho^2}\mathbb{E}[\|\frac{1}{K}\sum_{k=1}^{K}\nabla_1 g^{(k)}(x_t^{(k)},z_t^{(k)})-\frac{1}{K}\sum_{k=1}^{K}m_{x,3,t}^{(k)}\|^2] \qquad\qquad \text{Lemma A.8}$$

$$+ c_{11}\mathbb{E}[\|\frac{1}{K}\sum_{k=1}^{K}\nabla_2 f^{(k)}(x_t^{(k)},y_t^{(k)})-\frac{1}{K}\sum_{k=1}^{K}m_{y,1,t}^{(k)}\|^2] \qquad\qquad \text{Lemma A.8}$$

$$+ c_{12}\frac{1}{\rho^2}\mathbb{E}[\|\frac{1}{K}\sum_{k=1}^{K}\nabla_2 g^{(k)}(x_t^{(k)},y_t^{(k)})-\frac{1}{K}\sum_{k=1}^{K}m_{y,2,t}^{(k)}\|^2] \qquad\qquad \text{Lemma A.8}$$

$$+ c_{13}\frac{1}{\rho^2}\mathbb{E}[\|\frac{1}{K}\sum_{k=1}^{K}\nabla_2 g^{(k)}(x_t^{(k)},z_t^{(k)})-\frac{1}{K}\sum_{k=1}^{K}m_{z,1,t}^{(k)}\|^2] \qquad\qquad \text{Lemma A.8}$$

$$+ c_{14} \frac{1}{K} \sum_{k=1}^{K} \mathbb{E}[\|\nabla_1 f^{(k)}(x_t^{(k)}, y_t^{(k)}) - m_{x,1,t}^{(k)}\|^2] \qquad \text{Lemma A.9}$$

$$+ c_{15} \frac{1}{\rho^2} \frac{1}{K} \sum_{k=1}^{K} \mathbb{E}[\|\nabla_1 g^{(k)}(x_t^{(k)}, y_t^{(k)}) - m_{x,2,t}^{(k)}\|^2] \qquad \text{Lemma A.9}$$

$$+ c_{16} \frac{1}{\rho^2} \frac{1}{K} \sum_{k=1}^{K} \mathbb{E}[\|\nabla_1 g^{(k)}(x_t^{(k)}, z_t^{(k)}) - m_{x,3,t}^{(k)}\|^2] \qquad \text{Lemma A.9}$$

$$+ c_{17} \frac{1}{K} \sum_{k=1}^{K} \mathbb{E}[\|\nabla_2 f^{(k)}(x_t^{(k)}, y_t^{(k)}) - m_{y,1,t}^{(k)}\|^2] \qquad \text{Lemma A.9}$$

$$+ c_{18} \frac{1}{\rho^2} \frac{1}{K} \sum_{k=1}^{K} \mathbb{E}[\|\nabla_2 g^{(k)}(x_t^{(k)}, y_t^{(k)}) - m_{y,2,t}^{(k)}\|^2] \qquad \text{Lemma A.9}$$

$$+ c_{19} \frac{1}{\rho^2} \frac{1}{K} \sum_{k=1}^{K} \mathbb{E}[\|\nabla_2 g^{(k)}(x_t^{(k)}, z_t^{(k)}) - m_{z,1,t}^{(k)}\|^2] . \qquad \text{Lemma A.9} \qquad (12)$$

This potential function includes the optimization error, consensus error, and gradient estimation error.

**Step 2:** We establish the upper bound for the optimization error in Lemmas A.5-A.7 of Appendix A.4, that for the consensus error in Lemmas A.10-A.11 of Appendix A.6, and that for the gradient estimation error in Lemmas A.8-A.9 of Appendix A.5.

**Step 3:** Based on those upper bounds, we establish the upper bound for $P_{t+1} - P_t$ as shown in Eq. (76).

**Step 4:** We select feasible hyperparameters $\{c_i\}_{i=0}^{19}$ such that the coefficients for the consensus error, gradient estimation error, $\mathbb{E}[\|\nabla_2 h_\rho(\bar{x}_t, \bar{y}_t)\|^2]$, $\mathbb{E}[\|\nabla_2 g(\bar{x}_t, \bar{z}_t)\|^2]$, $\mathbb{E}[\|\bar{m}_{x,t}\|^2]$, $\mathbb{E}[\|\bar{m}_{y,t}\|^2]$, and $\mathbb{E}[\|\bar{m}_{z,t}\|^2]$ in Eq. (76) are non-positive.

### A.3. Fundamental Lemmas

**Lemma A.1.** *(Chen et al., 2024) Suppose Assumptions 4.1-4.5 hold, $\mathcal{L}(x)$ is $L_{\mathcal{L}}$-Lipschitz smooth, where $L_{\mathcal{L}} = O(\ell \kappa^3)$.*

**Lemma A.2.** *Suppose Assumptions 4.1-4.5 hold, denoting $Y^*(x) = \arg\min_{y \in \mathbb{R}^{d_y}} g(x, y)$, we have that*

$$dist(Y^*(x_1), Y^*(x_2)) \le C_{y^*} \|x_1 - x_2\|, \qquad (13)$$

*for any $x_1 \in \mathbb{R}^{d_x}$ and $x_2 \in \mathbb{R}^{d_x}$, where $C_{y^*} = \frac{L_g}{\mu} = O(\kappa)$.*

*Proof.* For any $y_1 \in Y^*(x_1)$, there exists $y_2 \in Y^*(x_2)$ such that

$$\begin{aligned}
\mu\|y_2 - y_1\| \\
\le \|\nabla_2 g(x_2, y_1)\| \\
= \|\nabla_2 g(x_2, y_1) - \nabla_2 g(x_1, y_1)\| \\
\le L_g \|x_2 - x_1\| .
\end{aligned} \qquad (14)$$

Similarly, for any $y_2 \in Y^*(x_2)$, there exists $y_1 \in Y^*(x_1)$ such that the above inequality holds. This completes the proof. $\square$

**Lemma A.3.** *Suppose Assumptions 4.1-4.5 hold, we have that*

$$\begin{aligned}
\|\nabla \hat{g}(x_1) - \nabla \hat{g}(x_2)\| \le L_{\hat{g}} \|x_1 - x_2\|, \\
\|\nabla \hat{h}_\rho(x_1) - \nabla \hat{h}_\rho(x_2)\| \le L_{\hat{h}} \|x_1 - x_2\|,
\end{aligned} \qquad (15)$$

*where $L_{\hat{g}} = L_g(1 + C_{y^*}) = O(\kappa)$ and $L_{\hat{h}} = (\rho L_f + L_g)(1 + C_{y^*_\rho}) = O(\kappa)$.*

*Proof.* At first, we have

$$
\begin{aligned}
\nabla \hat{g}(x) &= \nabla_x g(x, y^*(x)) \\
&= \nabla_1 g(x, y^*(x)) + (\nabla y^*(x))^T \nabla_2 g(x, y^*(x)) \\
&= \nabla_1 g(x, y^*(x)) \, ,
\end{aligned}
\tag{16}
$$

where the last step follows from the fact that $y^*(x) \in \arg\min_{z \in \mathbb{R}^{d_y}} g(x, z)$. Then, for any $x_1 \in \mathbb{R}^{d_x}$ and $x_2 \in \mathbb{R}^{d_x}$, we have

$$
\begin{aligned}
&\|\nabla \hat{g}(x_1) - \nabla \hat{g}(x_2)\| \\
&= \|\nabla_1 g(x_1, y^*(x_1)) - \nabla_1 g(x_2, y^*(x_2))\| \\
&\leq L_g \|x_1 - x_2\| + L_g \|y^*(x_1) - y^*(x_2)\| \\
&\leq L_g \|x_1 - x_2\| + L_g L_{y^*} \|x_1 - x_2\| \\
&= L_{\hat{g}} \|x_1 - x_2\| \, ,
\end{aligned}
\tag{17}
$$

where $L_{\hat{g}} = L_g(1 + C_{y^*})$.

As for $\hat{h}_\rho(x)$, we have

$$
\begin{aligned}
\nabla \hat{h}_\rho(x) &= \rho \nabla_1 f(x, y_\rho^*(x)) + \nabla_1 g(x, y_\rho^*(x)) + (\nabla y_\rho^*(x))^T (\rho \nabla_2 f(x, y_\rho^*(x)) + \nabla_2 g(x, y_\rho^*(x))) \\
&= \rho \nabla_1 f(x, y_\rho^*(x)) + \nabla_1 g(x, y_\rho^*(x)) \, ,
\end{aligned}
\tag{18}
$$

where the second step follows from the fact that $y_\rho^*(x) \in \arg\min_{y \in \mathbb{R}^{d_y}} h_\rho(x, y)$.

Then, for any $x_1 \in \mathbb{R}^{d_x}$ and $x_2 \in \mathbb{R}^{d_x}$, we have

$$
\begin{aligned}
&\|\nabla \hat{h}_\rho(x_1) - \nabla \hat{h}_\rho(x_2)\| \\
&= \|\rho \nabla_1 f(x_1, y_\rho^*(x_1)) + \nabla_1 g(x_1, y_\rho^*(x_1)) - \rho \nabla_2 f(x_2, y_\rho^*(x_2)) - \nabla_2 g(x_2, y_\rho^*(x_2))\| \\
&\leq \rho \|\nabla_1 f(x_1, y_\rho^*(x_1)) - \nabla_2 f(x_2, y_\rho^*(x_2))\| + \|\nabla_1 g(x_1, y_\rho^*(x_1)) - \nabla_2 g(x_2, y_\rho^*(x_2))\| \\
&\leq \rho L_f \|x_1 - x_2\| + \rho L_f \|y_\rho^*(x_1) - y_\rho^*(x_2)\| + L_g \|x_1 - x_2\| + L_g \|y_\rho^*(x_1) - y_\rho^*(x_2)\| \\
&\leq (\rho L_f + L_g)\|x_1 - x_2\| + (\rho L_f + L_g) C_{y_\rho^*} \|x_1 - x_2\| \\
&= L_{\hat{h}} \|x_1 - x_2\| \, ,
\end{aligned}
\tag{19}
$$

where $L_{\hat{h}} = (\rho L_f + L_g)(1 + C_{y_\rho^*})$, the second to last step follows from Lemma A.14.

$\square$

**Lemma A.4.** *Suppose Assumptions 4.1-4.5 hold, we have that*

$$
\begin{aligned}
\|y^*(x) - z\|^2 &\leq \frac{2}{\mu}(g(x, z) - g(x, y^*(x))) \, , \\
\|y_\rho^*(x) - y\|^2 &\leq \frac{2}{\mu}(h_\rho(x, y) - h_\rho(x, y_\rho^*(x))) \, .
\end{aligned}
\tag{20}
$$

*Proof.* According to Assumption 4.3, the penalty function $h_\rho(x, y)$ is $\mu$-PL in $y$. Then, in terms of Lemma A.1 in (Yang et al., 2020), it satisfies the quadratic growth condition as follows:

$$
\frac{\mu}{2}\|y_\rho^*(x) - y\|^2 \leq h_\rho(x, y) - h_\rho(x, y_\rho^*(x)) \, ,
\tag{21}
$$

which completes the proof.

Similarly, according to Assumption 4.3, the function $g(x, z)$ is $\mu$-PL in $z$. Then, in terms of Lemma A.1 in (Yang et al., 2020), it satisfies the quadratic growth condition as follows:

$$
\frac{\mu}{2}\|y^*(x) - z\|^2 \leq g(x, z) - g(x, y^*(x)) \, ,
\tag{22}
$$

which completes the proof.

$\square$

## A.4. Optimization Error

**Lemma A.5.** *Suppose Assumptions 4.1-4.5 hold, when $\eta_x \leq \frac{1}{2L_{\mathcal{L}}}$, we have that*

$$\mathbb{E}[\mathcal{L}(\bar{x}_{t+1})] \leq \mathbb{E}[\mathcal{L}(\bar{x}_t)] - \frac{\eta_x}{2}\mathbb{E}[\|\nabla\mathcal{L}(\bar{x}_t)\|^2] - \frac{\eta_x}{4}\mathbb{E}[\|\bar{m}_{x,t}\|^2] + \frac{3\eta_x}{2}\mathbb{E}[\|\nabla\mathcal{L}(\bar{x}_t) - \nabla\mathcal{L}_\rho(\bar{x}_t)\|^2]$$

$$+ \frac{9\eta_x}{2\mu^2}(L_f^2 + \frac{L_g^2}{\rho^2})\mathbb{E}[\|\nabla_2 h_\rho(\bar{x}_t, \bar{y}_t)\|^2] + \frac{9\eta_x}{2\mu^2}\frac{L_g^2}{\rho^2}\mathbb{E}[\|\nabla_2 g(\bar{x}_t, \bar{z}_t)\|^2] + 9\eta_x\frac{1}{\rho^2}L_g^2\frac{1}{K}\mathbb{E}[\|Z_t - \bar{Z}_t\|_F^2]$$

$$+ 9\eta_x(L_f^2 + 2\frac{1}{\rho^2}L_g^2)\frac{1}{K}\mathbb{E}[\|X_t - \bar{X}_t\|_F^2] + 9\eta_x(L_f^2 + \frac{1}{\rho^2}L_g^2)\frac{1}{K}\mathbb{E}[\|Y_t - \bar{Y}_t\|_F^2]$$

$$+ 9\eta_x\mathbb{E}[\|\frac{1}{K}\sum_{k=1}^K \nabla_1 f^{(k)}(x_t^{(k)}, y_t^{(k)}) - \frac{1}{K}\sum_{k=1}^K m_{x,1,t}^{(k)}\|^2]$$

$$+ 9\eta_x\frac{1}{\rho^2}\mathbb{E}[\|\frac{1}{K}\sum_{k=1}^K \nabla_1 g^{(k)}(x_t^{(k)}, y_t^{(k)}) - \frac{1}{K}\sum_{k=1}^K m_{x,2,t}^{(k)}\|^2]$$

$$+ 9\eta_x\frac{1}{\rho^2}\mathbb{E}[\|\frac{1}{K}\sum_{k=1}^K \nabla_1 g^{(k)}(x_t^{(k)}, z_t^{(k)}) - \frac{1}{K}\sum_{k=1}^K m_{x,3,t}^{(k)}\|^2]. \tag{23}$$

*Proof.* Because $\mathcal{L}(x)$ is $L_{\mathcal{L}}$-Lipschitz smooth, we have

$$\mathbb{E}[\mathcal{L}(\bar{x}_{t+1})] \leq \mathbb{E}[\mathcal{L}(\bar{x}_t)] + \mathbb{E}[\langle\nabla\mathcal{L}(\bar{x}_t), \bar{x}_{t+1} - \bar{x}_t\rangle] + \frac{L_{\mathcal{L}}}{2}\mathbb{E}[\|\bar{x}_{t+1} - \bar{x}_t\|^2]$$

$$= \mathbb{E}[\mathcal{L}(\bar{x}_t)] - \eta_x\mathbb{E}[\langle\nabla\mathcal{L}(\bar{x}_t), \bar{m}_{x,t}\rangle] + \frac{\eta_x^2 L_{\mathcal{L}}}{2}\mathbb{E}[\|\bar{m}_{x,t}\|^2]$$

$$= \mathbb{E}[\mathcal{L}(\bar{x}_t)] - \frac{\eta_x}{2}\mathbb{E}[\|\nabla\mathcal{L}(\bar{x}_t)\|^2] - \frac{\eta_x}{2}\mathbb{E}[\|\bar{m}_{x,t}\|^2] + \frac{\eta_x}{2}\mathbb{E}[\|\nabla\mathcal{L}(\bar{x}_t) - \bar{m}_{x,t}\|^2] + \frac{\eta_x^2 L_{\mathcal{L}}}{2}\mathbb{E}[\|\bar{m}_{x,t}\|^2]$$

$$\leq \mathbb{E}[\mathcal{L}(\bar{x}_t)] - \frac{\eta_x}{2}\mathbb{E}[\|\nabla\mathcal{L}(\bar{x}_t)\|^2] - \frac{\eta_x}{4}\mathbb{E}[\|\bar{m}_{x,t}\|^2] + \frac{\eta_x}{2}\mathbb{E}[\|\nabla\mathcal{L}(\bar{x}_t) - \bar{m}_{x,t}\|^2]$$

$$\leq \mathbb{E}[\mathcal{L}(\bar{x}_t)] - \frac{\eta_x}{2}\mathbb{E}[\|\nabla\mathcal{L}(\bar{x}_t)\|^2] - \frac{\eta_x}{4}\mathbb{E}[\|\bar{m}_{x,t}\|^2] + \frac{3\eta_x}{2}\mathbb{E}[\|\nabla\mathcal{L}(\bar{x}_t) - \nabla\mathcal{L}_\rho(\bar{x}_t)\|^2]$$

$$+ \frac{3\eta_x}{2}\mathbb{E}[\|\nabla\mathcal{L}_\rho(\bar{x}_t) - \nabla_x\mathcal{L}_\rho(\bar{x}_t, \bar{y}_t, \bar{z}_t)\|^2] + \frac{3\eta_x}{2}\mathbb{E}[\|\nabla_x\mathcal{L}_\rho(\bar{x}_t, \bar{y}_t, \bar{z}_t) - \bar{m}_{x,t}\|^2], \tag{24}$$

where the fourth step follows from $\eta_x \leq \frac{1}{2L_{\mathcal{L}}}$.

**Bounding** $\mathbb{E}[\|\nabla\mathcal{L}_\rho(\bar{x}_t) - \nabla_x\mathcal{L}_\rho(\bar{x}_t, \bar{y}_t, \bar{z}_t)\|^2]$. For $\nabla\mathcal{L}_\rho(\bar{x}_t)$, we have

$$\nabla\mathcal{L}_\rho(\bar{x}_t) = \nabla_x f(\bar{x}_t, y_\rho^*(\bar{x}_t)) + \frac{1}{\rho}\nabla_x g(\bar{x}_t, y_\rho^*(\bar{x}_t)) - \frac{1}{\rho}\nabla_x g(\bar{x}_t, y^*(\bar{x}_t))$$

$$= \nabla_1 f(\bar{x}_t, y_\rho^*(\bar{x}_t)) + (\nabla y_\rho^*(\bar{x}_t))^T \nabla_2 f(\bar{x}_t, y_\rho^*(\bar{x}_t))$$

$$+ \frac{1}{\rho}\nabla_1 g(\bar{x}_t, y_\rho^*(\bar{x}_t)) + \frac{1}{\rho}(\nabla y^*(\bar{x}_t))^T \nabla_2 g(\bar{x}_t, y_\rho^*(\bar{x}_t))$$

$$- \frac{1}{\rho}\nabla_1 g(\bar{x}_t, y^*(\bar{x}_t)) - \frac{1}{\rho}(\nabla z^*(\bar{x}_t))^T \nabla_2 g(\bar{x}_t, y^*(\bar{x}_t)). \tag{25}$$

Because $y_\rho^*(\bar{x}_t) = \arg\min_y \rho f(\bar{x}_t, y) + g(\bar{x}_t, y)$, we can obtain $\nabla_2 f(\bar{x}_t, y_\rho^*(\bar{x}_t)) + \frac{1}{\rho}\nabla_2 g(\bar{x}_t, y_\rho^*(\bar{x}_t)) = 0$. Similarly, we can obtain $\nabla_2 g(\bar{x}_t, y^*(\bar{x}_t)) = 0$, as $y^*(\bar{x}_t) = \arg\min_z g(\bar{x}_t, z)$. As a result, we can know

$$\nabla\mathcal{L}_\rho(\bar{x}_t) = \nabla_1 f(\bar{x}_t, y_\rho^*(\bar{x}_t)) + \frac{1}{\rho}\nabla_1 g(\bar{x}_t, y_\rho^*(\bar{x}_t)) - \frac{1}{\rho}\nabla_1 g(\bar{x}_t, y^*(\bar{x}_t)). \tag{26}$$

Based on this gradient, we can bound $\mathbb{E}[\|\nabla\mathcal{L}_\rho(\bar{x}_t) - \nabla_x\mathcal{L}_\rho(\bar{x}_t, \bar{y}_t, \bar{z}_t)\|^2]$ as follows:

$$\mathbb{E}[\|\nabla\mathcal{L}_\rho(\bar{x}_t) - \nabla_x\mathcal{L}_\rho(\bar{x}_t, \bar{y}_t, \bar{z}_t)\|^2]$$

$$= \mathbb{E}[\|\nabla_1 f(\bar{x}_t, y_\rho^*(\bar{x}_t)) + \frac{1}{\rho}\nabla_1 g(\bar{x}_t, y_\rho^*(\bar{x}_t)) - \frac{1}{\rho}\nabla_1 g(\bar{x}_t, y^*(\bar{x}_t))$$

$$- (\nabla_1 f(\bar{x}_t, \bar{y}_t) + \frac{1}{\rho}\nabla_1 g(\bar{x}_t, \bar{y}_t) - \frac{1}{\rho}\nabla_1 g(\bar{x}_t, \bar{z}_t))\|^2]$$

$$\leq 3\mathbb{E}[\|\nabla_1 f(\bar{x}_t, y_\rho^*(\bar{x}_t)) - \nabla_1 f(\bar{x}_t, \bar{y}_t)\|^2] + 3\mathbb{E}[\|\frac{1}{\rho}\nabla_1 g(\bar{x}_t, y_\rho^*(\bar{x}_t)) - \frac{1}{\rho}\nabla_1 g(\bar{x}_t, \bar{y}_t)\|^2]$$

$$+ 3\mathbb{E}[\|\frac{1}{\rho}\nabla_1 g(\bar{x}_t, y^*(\bar{x}_t)) - \frac{1}{\rho}\nabla_1 g(\bar{x}_t, \bar{z}_t)\|^2]$$

$$\leq 3(L_f^2 + \frac{L_g^2}{\rho^2})\mathbb{E}[\|\bar{y}_t - y_\rho^*(\bar{x}_t)\|^2] + 3\frac{L_g^2}{\rho^2}\mathbb{E}[\|\bar{z}_t - y^*(\bar{x}_t)\|^2]$$

$$\leq 3(L_f^2 + \frac{L_g^2}{\rho^2})\frac{2}{\mu}\mathbb{E}[h_\rho(\bar{x}_t, \bar{y}_t) - h_\rho(\bar{x}_t, y_\rho^*(\bar{x}_t))] + 3\frac{L_g^2}{\rho^2}\frac{2}{\mu}\mathbb{E}[g(\bar{x}_t, \bar{z}_t) - g(\bar{x}_t, y^*(\bar{x}_t))]$$

$$\leq \frac{3}{\mu^2}(L_f^2 + \frac{L_g^2}{\rho^2})\mathbb{E}[\|\nabla_2 h_\rho(\bar{x}_t, \bar{y}_t)\|^2] + \frac{3}{\mu^2}\frac{L_g^2}{\rho^2}\mathbb{E}[\|\nabla_2 g(\bar{x}_t, \bar{z}_t)\|^2] \,, \tag{27}$$

where the third to last step follows from Assumptions 4.1, 4.2, the second to last step follows from Lemma A.4, and the last step follows from Assumption 4.3.

**Bounding $\mathbb{E}[\|\nabla_x \mathcal{L}_\rho(\bar{x}_t, \bar{y}_t, \bar{z}_t) - \bar{m}_{x,t}\|^2]$.** As for $\mathbb{E}[\|\nabla_x \mathcal{L}_\rho(\bar{x}_t, \bar{y}_t, \bar{z}_t) - \bar{m}_{x,t}\|^2]$, we have

$$\mathbb{E}[\|\nabla_x \mathcal{L}_\rho(\bar{x}_t, \bar{y}_t, \bar{z}_t) - \bar{m}_{x,t}\|^2]$$

$$\leq 3\mathbb{E}[\|\nabla_1 f(\bar{x}_t, \bar{y}_t) - \bar{m}_{x,1,t}\|^2] + 3\frac{1}{\rho^2}\mathbb{E}[\|\nabla_1 g(\bar{x}_t, \bar{y}_t) - \bar{m}_{x,2,t}\|^2] + 3\frac{1}{\rho^2}\mathbb{E}[\|\nabla_1 g(\bar{x}_t, \bar{z}_t) - \bar{m}_{x,3,t}\|^2]$$

$$\leq 6\mathbb{E}[\|\nabla_1 f(\bar{x}_t, \bar{y}_t) - \frac{1}{K}\sum_{k=1}^{K}\nabla_1 f^{(k)}(x_t^{(k)}, y_t^{(k)})\|^2] + 6\mathbb{E}[\|\frac{1}{K}\sum_{k=1}^{K}\nabla_1 f^{(k)}(x_t^{(k)}, y_t^{(k)}) - \frac{1}{K}\sum_{k=1}^{K}m_{x,1,t}^{(k)}\|^2]$$

$$+ 6\frac{1}{\rho^2}\mathbb{E}[\|\nabla_1 g(\bar{x}_t, \bar{y}_t) - \frac{1}{K}\sum_{k=1}^{K}\nabla_1 g^{(k)}(x_t^{(k)}, y_t^{(k)})\|^2] + 6\frac{1}{\rho^2}\mathbb{E}[\|\frac{1}{K}\sum_{k=1}^{K}\nabla_1 g^{(k)}(x_t^{(k)}, y_t^{(k)}) - \frac{1}{K}\sum_{k=1}^{K}m_{x,2,t}^{(k)}\|^2]$$

$$+ 6\frac{1}{\rho^2}\mathbb{E}[\|\nabla_1 g(\bar{x}_t, \bar{z}_t) - \frac{1}{K}\sum_{k=1}^{K}\nabla_1 g^{(k)}(x_t^{(k)}, z_t^{(k)})\|^2] + 6\frac{1}{\rho^2}\mathbb{E}[\|\frac{1}{K}\sum_{k=1}^{K}\nabla_1 g^{(k)}(x_t^{(k)}, z_t^{(k)}) - \frac{1}{K}\sum_{k=1}^{K}m_{x,3,t}^{(k)}\|^2]$$

$$\leq 6(L_f^2 + 2\frac{L_g^2}{\rho^2})\frac{1}{K}\mathbb{E}[\|X_t - \bar{X}_t\|_F^2] + 6(L_f^2 + \frac{L_g^2}{\rho^2})\frac{1}{K}\mathbb{E}[\|Y_t - \bar{Y}_t\|_F^2] + 6\frac{L_g^2}{\rho^2}\frac{1}{K}\mathbb{E}[\|Z_t - \bar{Z}_t\|_F^2]$$

$$+ 6\mathbb{E}[\|\frac{1}{K}\sum_{k=1}^{K}\nabla_1 f^{(k)}(x_t^{(k)}, y_t^{(k)}) - \frac{1}{K}\sum_{k=1}^{K}m_{x,1,t}^{(k)}\|^2] + 6\frac{1}{\rho^2}\mathbb{E}[\|\frac{1}{K}\sum_{k=1}^{K}\nabla_1 g^{(k)}(x_t^{(k)}, y_t^{(k)}) - \frac{1}{K}\sum_{k=1}^{K}m_{x,2,t}^{(k)}\|^2]$$

$$+ 6\frac{1}{\rho^2}\mathbb{E}[\|\frac{1}{K}\sum_{k=1}^{K}\nabla_1 g^{(k)}(x_t^{(k)}, z_t^{(k)}) - \frac{1}{K}\sum_{k=1}^{K}m_{x,3,t}^{(k)}\|^2] \,, \tag{28}$$

where the last step follows from Assumptions 4.1, 4.2.

By plugging Eq. (27) and Eq. (28) into Eq. (24), we complete the proof.

$\square$

**Lemma A.6.** *(Restatement of Lemma 4.11) Suppose Assumptions 4.1-4.5 hold, when $\eta_x \leq \frac{\mu^2}{4L_g^2}\eta_z$ and $\eta_z \leq \frac{\rho}{2L_g}$, we have that*

$$\frac{1}{\rho}\mathbb{E}[g(\bar{x}_{t+1}, \bar{z}_{t+1}) - \hat{g}(\bar{x}_{t+1})] \leq \frac{1}{\rho}\mathbb{E}[g(\bar{x}_t, \bar{z}_t) - \hat{g}(\bar{x}_t)] - \frac{1}{\rho^2}\frac{\eta_z}{8}\mathbb{E}[\|\nabla_2 g(\bar{x}_t, \bar{z}_t)\|^2] - \frac{\eta_z}{4}\mathbb{E}[\|\bar{m}_{z,t}\|^2]$$

$$+ \left(\frac{\eta_x}{2} + 2\eta_x^2 L_g\frac{1}{\rho} + \frac{\eta_x^2 L_{\hat{g}}}{2}\frac{1}{\rho}\right)\mathbb{E}[\|\bar{m}_{x,t}\|^2] + \frac{2\eta_z}{\rho^2}\mathbb{E}[\|\frac{1}{K}\sum_{k=1}^{K}\nabla_2 g^{(k)}(x_t^{(k)}, z_t^{(k)}) - \frac{1}{K}\sum_{k=1}^{K}m_{z,1,t}^{(k)}\|^2]$$

$$+ 2\eta_z \frac{L_g^2}{\rho^2} \frac{1}{K} \mathbb{E}[\|X_t - \bar{X}_t\|_F^2] + 2\eta_z \frac{L_g^2}{\rho^2} \frac{1}{K} \mathbb{E}[\|Z_t - \bar{Z}_t\|_F^2] \,. \tag{29}$$

*Proof.* On the one hand, we have

$$\frac{1}{\rho}\mathbb{E}[g(\bar{x}_{t+1}, \bar{z}_{t+1})] \leq \frac{1}{\rho}\mathbb{E}[g(\bar{x}_{t+1}, \bar{z}_t)] + \frac{1}{\rho}\mathbb{E}[\langle \nabla_2 g(\bar{x}_{t+1}, \bar{z}_t), \bar{z}_{t+1} - \bar{z}_t \rangle] + \frac{1}{\rho}\frac{L_g}{2}\mathbb{E}[\|\bar{z}_{t+1} - \bar{z}_t\|^2]$$

$$= \frac{1}{\rho}\mathbb{E}[g(\bar{x}_{t+1}, \bar{z}_t)] - \eta_z \mathbb{E}[\langle \frac{1}{\rho}\nabla_2 g(\bar{x}_{t+1}, \bar{z}_t), \bar{m}_{z,t} \rangle] + \frac{1}{\rho}\frac{\eta_z^2 L_g}{2}\mathbb{E}[\|\bar{m}_{z,t}\|^2]$$

$$= \frac{1}{\rho}\mathbb{E}[g(\bar{x}_{t+1}, \bar{z}_t)] - \frac{\eta_z}{2}\mathbb{E}[\|\frac{1}{\rho}\nabla_2 g(\bar{x}_{t+1}, \bar{z}_t)\|^2] - \frac{\eta_z}{2}\mathbb{E}[\|\bar{m}_{z,t}\|^2]$$

$$\quad + \frac{\eta_z}{2}\mathbb{E}[\|\frac{1}{\rho}\nabla_2 g(\bar{x}_{t+1}, \bar{z}_t) - \bar{m}_{z,t}\|^2] + \frac{1}{\rho}\frac{\eta_z^2 L_g}{2}\mathbb{E}[\|\bar{m}_{z,t}\|^2]$$

$$\leq \frac{1}{\rho}\mathbb{E}[g(\bar{x}_{t+1}, \bar{z}_t)] - \frac{1}{\rho^2}\frac{\eta_z}{4}\mathbb{E}[\|\nabla_2 g(\bar{x}_t, \bar{z}_t)\|^2] + \frac{1}{\rho^2}\frac{\eta_z}{2}\mathbb{E}[\|\nabla_2 g(\bar{x}_t, \bar{z}_t) - \nabla_2 g(\bar{x}_{t+1}, \bar{z}_t)\|^2]$$

$$\quad - \frac{\eta_z}{4}\mathbb{E}[\|\bar{m}_{z,t}\|^2] + \eta_z\frac{1}{\rho^2}\mathbb{E}[\|\nabla_2 g(\bar{x}_t, \bar{z}_t) - \nabla_2 g(\bar{x}_{t+1}, \bar{z}_t)\|^2] + \eta_z\mathbb{E}[\|\frac{1}{\rho}\nabla_2 g(\bar{x}_t, \bar{z}_t) - \bar{m}_{z,t}\|^2]$$

$$\leq \frac{1}{\rho}\mathbb{E}[g(\bar{x}_{t+1}, \bar{z}_t)] - \frac{1}{\rho^2}\frac{\eta_z}{4}\mathbb{E}[\|\nabla_2 g(\bar{x}_t, \bar{z}_t)\|^2] - \frac{\eta_z}{4}\mathbb{E}[\|\bar{m}_{z,t}\|^2] + \frac{1}{\rho^2}\frac{3\eta_x^2\eta_z L_g^2}{2}\mathbb{E}[\|\bar{m}_{x,t}\|^2]$$

$$\quad + \eta_z\mathbb{E}[\|\frac{1}{\rho}\nabla_2 g(\bar{x}_t, \bar{z}_t) - \bar{m}_{z,t}\|^2]$$

$$\leq \frac{1}{\rho}\mathbb{E}[g(\bar{x}_{t+1}, \bar{z}_t)] - \frac{1}{\rho^2}\frac{\eta_z}{4}\mathbb{E}[\|\nabla_2 g(\bar{x}_t, \bar{z}_t)\|^2] - \frac{\eta_z}{4}\mathbb{E}[\|\bar{m}_{z,t}\|^2] + \frac{1}{\rho}\frac{3\eta_x^2 L_g}{2}\mathbb{E}[\|\bar{m}_{x,t}\|^2]$$

$$\quad + \eta_z\mathbb{E}[\|\frac{1}{\rho}\nabla_2 g(\bar{x}_t, \bar{z}_t) - \bar{m}_{z,t}\|^2]$$

$$\leq \frac{1}{\rho}\mathbb{E}[g(\bar{x}_{t+1}, \bar{z}_t)] - \frac{1}{\rho^2}\frac{\eta_z}{4}\mathbb{E}[\|\nabla_2 g(\bar{x}_t, \bar{z}_t)\|^2] - \frac{\eta_z}{4}\mathbb{E}[\|\bar{m}_{z,t}\|^2] + \frac{1}{\rho}\frac{3\eta_x^2 L_g}{2}\mathbb{E}[\|\bar{m}_{x,t}\|^2]$$

$$\quad + 2\eta_z\mathbb{E}[\|\frac{1}{\rho}\nabla_2 g(\bar{x}_t, \bar{z}_t) - \frac{1}{\rho}\frac{1}{K}\sum_{k=1}^{K} \nabla_2 g^{(k)}(x_t^{(k)}, z_t^{(k)})\|^2]$$

$$\quad + 2\eta_z\mathbb{E}[\|\frac{1}{\rho}\frac{1}{K}\sum_{k=1}^{K} \nabla_2 g^{(k)}(x_t^{(k)}, z_t^{(k)}) - \frac{1}{\rho}\frac{1}{K}\sum_{k=1}^{K} m_{z,1,t}^{(k)}\|^2]$$

$$\leq \frac{1}{\rho}\mathbb{E}[g(\bar{x}_{t+1}, \bar{z}_t)] - \frac{1}{\rho^2}\frac{\eta_z}{4}\mathbb{E}[\|\nabla_2 g(\bar{x}_t, \bar{z}_t)\|^2] - \frac{\eta_z}{4}\mathbb{E}[\|\bar{m}_{z,t}\|^2] + \frac{1}{\rho}\frac{3\eta_x^2 L_g}{2}\mathbb{E}[\|\bar{m}_{x,t}\|^2]$$

$$\quad + 2\eta_z\frac{L_g^2}{\rho^2}\frac{1}{K}\mathbb{E}[\|X_t - \bar{X}_t\|_F^2] + 2\eta_z\frac{L_g^2}{\rho^2}\frac{1}{K}\mathbb{E}[\|Z_t - \bar{Z}_t\|_F^2]$$

$$\quad + \frac{2\eta_z}{\rho^2}\mathbb{E}[\|\frac{1}{K}\sum_{k=1}^{K} \nabla_2 g^{(k)}(x_t^{(k)}, z_t^{(k)}) - \frac{1}{K}\sum_{k=1}^{K} m_{z,1,t}^{(k)}\|^2] \,, \tag{30}$$

where the fourth step follows from $\eta_z \leq \frac{\rho}{2L_g}$ and $-\mathbb{E}[\|\nabla_2 g(\bar{x}_{t+1}, \bar{z}_t)\|^2] \leq -\frac{1}{2}\mathbb{E}[\|\nabla_2 g(\bar{x}_t, \bar{z}_t)\|^2] + \mathbb{E}[\|\nabla_2 g(\bar{x}_t, \bar{z}_t) - \nabla_2 g(\bar{x}_{t+1}, \bar{z}_t)\|^2]$.

On the other hand, we have

$$\frac{1}{\rho}\mathbb{E}[g(\bar{x}_{t+1}, \bar{z}_t)] \leq \frac{1}{\rho}\mathbb{E}[g(\bar{x}_t, \bar{z}_t)] + \frac{1}{\rho}\mathbb{E}[\langle \nabla_1 g(\bar{x}_t, \bar{z}_t), \bar{x}_{t+1} - \bar{x}_t \rangle] + \frac{1}{\rho}\frac{L_g}{2}\mathbb{E}[\|\bar{x}_{t+1} - \bar{x}_t\|^2]$$

$$= \frac{1}{\rho}\mathbb{E}[g(\bar{x}_t, \bar{z}_t)] + \frac{1}{\rho}\mathbb{E}[\langle \nabla_1 g(\bar{x}_t, \bar{z}_t) - \nabla_x g(\bar{x}_t, y^*(\bar{x}_t)), \bar{x}_{t+1} - \bar{x}_t \rangle]$$

$$\quad + \frac{1}{\rho}\mathbb{E}[\langle \nabla_x g(\bar{x}_t, y^*(\bar{x}_t)), \bar{x}_{t+1} - \bar{x}_t \rangle] + \frac{1}{\rho}\frac{\eta_x^2 L_g}{2}\mathbb{E}[\|\bar{m}_{x,t}\|^2]$$

$$= \frac{1}{\rho}\mathbb{E}[g(\bar{x}_t, \bar{z}_t)] - \eta_x \mathbb{E}[\langle \frac{1}{\rho}(\nabla_1 g(\bar{x}_t, \bar{z}_t) - \nabla_x g(\bar{x}_t, y^*(\bar{x}_t))), \bar{m}_{x,t}\rangle]$$

$$+ \frac{1}{\rho}\mathbb{E}[\langle \nabla_x g(\bar{x}_t, y^*(\bar{x}_t)), \bar{x}_{t+1} - \bar{x}_t\rangle] + \frac{1}{\rho}\frac{\eta_x^2 L_g}{2}\mathbb{E}[\|\bar{m}_{x,t}\|^2]$$

$$\leq \frac{1}{\rho}\mathbb{E}[g(\bar{x}_t, \bar{z}_t)] + \frac{1}{\rho^2}\frac{\eta_x}{2}\mathbb{E}[\|\nabla_1 g(\bar{x}_t, \bar{z}_t) - \nabla_x g(\bar{x}_t, y^*(\bar{x}_t))\|^2] + \frac{\eta_x}{2}\mathbb{E}[\|\bar{m}_{x,t}\|^2] + \frac{1}{\rho}\frac{\eta_x^2 L_g}{2}\mathbb{E}[\|\bar{m}_{x,t}\|^2]$$

$$+ \frac{1}{\rho}\mathbb{E}[\langle \nabla_x g(\bar{x}_t, y^*(\bar{x}_t)), \bar{x}_{t+1} - \bar{x}_t\rangle]$$

$$= \frac{1}{\rho}\mathbb{E}[g(\bar{x}_t, \bar{z}_t)] + \frac{1}{\rho^2}\frac{\eta_x}{2}\mathbb{E}[\|\nabla_1 g(\bar{x}_t, \bar{z}_t) - \nabla_1 g(\bar{x}_t, y^*(\bar{x}_t))\|^2] + \left(\frac{\eta_x}{2} + \frac{1}{\rho}\frac{\eta_x^2 L_g}{2}\right)\mathbb{E}[\|\bar{m}_{x,t}\|^2]$$

$$+ \frac{1}{\rho}\mathbb{E}[\langle \nabla_x g(\bar{x}_t, y^*(\bar{x}_t)), \bar{x}_{t+1} - \bar{x}_t\rangle]$$

$$\leq \frac{1}{\rho}\mathbb{E}[g(\bar{x}_t, \bar{z}_t)] + \frac{1}{\rho^2}\frac{\eta_x L_g^2}{2}\mathbb{E}[\|\bar{z}_t - y^*(\bar{x}_t)\|^2] + \left(\frac{\eta_x}{2} + \frac{1}{\rho}\frac{\eta_x^2 L_g}{2}\right)\mathbb{E}[\|\bar{m}_{x,t}\|^2]$$

$$+ \frac{1}{\rho}\mathbb{E}[\langle \nabla_x g(\bar{x}_t, y^*(\bar{x}_t)), \bar{x}_{t+1} - \bar{x}_t\rangle]$$

$$\leq \frac{1}{\rho}\mathbb{E}[g(\bar{x}_t, \bar{z}_t)] + \frac{1}{\rho^2}\frac{\eta_x L_g^2}{2\mu^2}\mathbb{E}[\|\nabla_2 g(\bar{x}_t, \bar{z}_t)\|^2] + \left(\frac{\eta_x}{2} + \frac{1}{\rho}\frac{\eta_x^2 L_g}{2}\right)\mathbb{E}[\|\bar{m}_{x,t}\|^2]$$

$$+ \frac{1}{\rho}\mathbb{E}[\langle \nabla_x g(\bar{x}_t, y^*(\bar{x}_t)), \bar{x}_{t+1} - \bar{x}_t\rangle] . \tag{31}$$

Then, by combining the above two inequalities, we have

$$\frac{1}{\rho}\mathbb{E}[g(\bar{x}_{t+1}, \bar{z}_{t+1})] \leq \frac{1}{\rho}\mathbb{E}[g(\bar{x}_{t+1}, \bar{z}_t)] - \frac{1}{\rho^2}\frac{\eta_z}{4}\mathbb{E}[\|\nabla_2 g(\bar{x}_t, \bar{z}_t)\|^2] - \frac{\eta_z}{4}\mathbb{E}[\|\bar{m}_{z,t}\|^2]$$

$$+ \frac{1}{\rho}\frac{3\eta_x^2 L_g}{2}\mathbb{E}[\|\bar{m}_{x,t}\|^2] + 2\eta_z \frac{L_g^2}{\rho^2}\frac{1}{K}\mathbb{E}[\|X_t - \bar{X}_t\|_F^2] + 2\eta_z \frac{L_g^2}{\rho^2}\frac{1}{K}\mathbb{E}[\|Z_t - \bar{Z}_t\|_F^2]$$

$$+ \frac{2\eta_z}{\rho^2}\mathbb{E}[\|\frac{1}{K}\sum_{k=1}^K \nabla_2 g^{(k)}(x_t^{(k)}, z_t^{(k)}) - \frac{1}{K}\sum_{k=1}^K m_{z,1,t}^{(k)}\|^2]$$

$$\leq \frac{1}{\rho}\mathbb{E}[g(\bar{x}_t, \bar{z}_t)] - \frac{1}{\rho^2}\frac{\eta_z}{4}\mathbb{E}[\|\nabla_2 g(\bar{x}_t, \bar{z}_t)\|^2] - \frac{\eta_z}{4}\mathbb{E}[\|\bar{m}_{z,t}\|^2] + \frac{1}{\rho}\frac{3\eta_x^2 L_g}{2}\mathbb{E}[\|\bar{m}_{x,t}\|^2]$$

$$+ 2\eta_z \frac{L_g^2}{\rho^2}\frac{1}{K}\mathbb{E}[\|X_t - \bar{X}_t\|_F^2] + 2\eta_z \frac{L_g^2}{\rho^2}\frac{1}{K}\mathbb{E}[\|Z_t - \bar{Z}_t\|_F^2]$$

$$+ \frac{2\eta_z}{\rho^2}\mathbb{E}[\|\frac{1}{K}\sum_{k=1}^K \nabla_2 g^{(k)}(x_t^{(k)}, z_t^{(k)}) - \frac{1}{K}\sum_{k=1}^K m_{z,1,t}^{(k)}\|^2]$$

$$+ \frac{1}{\rho^2}\frac{\eta_x L_g^2}{2\mu^2}\mathbb{E}[\|\nabla_2 g(\bar{x}_t, \bar{z}_t)\|^2] + \left(\frac{\eta_x}{2} + \frac{1}{\rho}\frac{\eta_x^2 L_g}{2}\right)\mathbb{E}[\|\bar{m}_{x,t}\|^2]$$

$$+ \frac{1}{\rho}\mathbb{E}[\langle \nabla_x g(\bar{x}_t, y^*(\bar{x}_t)), \bar{x}_{t+1} - \bar{x}_t\rangle]$$

$$= \frac{1}{\rho}\mathbb{E}[g(\bar{x}_t, \bar{z}_t)] + \frac{1}{\rho^2}\left(\frac{\eta_x L_g^2}{2\mu^2} - \frac{\eta_z}{4}\right)\mathbb{E}[\|\nabla_2 g(\bar{x}_t, \bar{z}_t)\|^2] - \frac{\eta_z}{4}\mathbb{E}[\|\bar{m}_{z,t}\|^2]$$

$$+ \left(\frac{\eta_x}{2} + 2\eta_x^2 L_g \frac{1}{\rho}\right)\mathbb{E}[\|\bar{m}_{x,t}\|^2] + 2\eta_z \frac{L_g^2}{\rho^2}\frac{1}{K}\mathbb{E}[\|X_t - \bar{X}_t\|_F^2] + 2\eta_z \frac{L_g^2}{\rho^2}\frac{1}{K}\mathbb{E}[\|Z_t - \bar{Z}_t\|_F^2]$$

$$+ \frac{2\eta_z}{\rho^2}\mathbb{E}[\|\frac{1}{K}\sum_{k=1}^K \nabla_2 g^{(k)}(x_t^{(k)}, z_t^{(k)}) - \frac{1}{K}\sum_{k=1}^K m_{z,1,t}^{(k)}\|^2] + \frac{1}{\rho}\mathbb{E}[\langle \nabla_x g(\bar{x}_t, y^*(\bar{x}_t)), \bar{x}_{t+1} - \bar{x}_t\rangle] . \tag{32}$$

Moreover, because $\hat{g}(x)$ is $L_{\hat{g}}$-smooth, we have

$$\hat{g}(\bar{x}_{t+1}) \geq \hat{g}(\bar{x}_t) + \langle \nabla \hat{g}(\bar{x}_t), \bar{x}_{t+1} - \bar{x}_t \rangle - \frac{L_{\hat{g}}}{2} \|\bar{x}_{t+1} - \bar{x}_t\|^2$$

$$= \hat{g}(\bar{x}_t) + \langle \nabla_x g(\bar{x}_t, y^*(\bar{x}_t)), \bar{x}_{t+1} - \bar{x}_t \rangle - \frac{L_{\hat{g}}}{2} \|\bar{x}_{t+1} - \bar{x}_t\|^2 . \tag{33}$$

Therefore, we have

$$\frac{1}{\rho} \hat{g}(\bar{x}_t) - \frac{1}{\rho} \hat{g}(\bar{x}_{t+1}) \leq -\frac{1}{\rho} \langle \nabla_x g(\bar{x}_t, y^*(\bar{x}_t)), \bar{x}_{t+1} - \bar{x}_t \rangle + \frac{1}{\rho} \frac{L_{\hat{g}}}{2} \|\bar{x}_{t+1} - \bar{x}_t\|^2 . \tag{34}$$

As a result, we have

$$\frac{1}{\rho} \mathbb{E}[g(\bar{x}_{t+1}, \bar{z}_{t+1}) - \hat{g}(\bar{x}_{t+1})]$$

$$= \frac{1}{\rho} \mathbb{E}[g(\bar{x}_{t+1}, \bar{z}_{t+1}) - g(\bar{x}_t, \bar{z}_t) + g(\bar{x}_t, \bar{z}_t) - \hat{g}(\bar{x}_t) + \hat{g}(\bar{x}_t) - \hat{g}(\bar{x}_{t+1})]$$

$$\leq \frac{1}{\rho} \mathbb{E}[g(\bar{x}_t, \bar{z}_t) - \hat{g}(\bar{x}_t)] + \frac{1}{\rho} \mathbb{E}[g(\bar{x}_{t+1}, \bar{z}_{t+1}) - g(\bar{x}_t, \bar{z}_t)] + \frac{1}{\rho} \mathbb{E}[\hat{g}(\bar{x}_t) - \hat{g}(\bar{x}_{t+1})]$$

$$\leq \frac{1}{\rho} \mathbb{E}[g(\bar{x}_t, \bar{z}_t) - \hat{g}(\bar{x}_t)] + \frac{1}{\rho^2} \left( \frac{\eta_x L_g^2}{2\mu^2} - \frac{\eta_z}{4} \right) \mathbb{E}[\|\nabla_2 g(\bar{x}_t, \bar{z}_t)\|^2] - \frac{\eta_z}{4} \mathbb{E}[\|\bar{m}_{z,t}\|^2]$$

$$+ \left( \frac{\eta_x}{2} + 2\eta_x^2 L_g \frac{1}{\rho} + \frac{\eta_x^2 L_{\hat{g}}}{2} \frac{1}{\rho} \right) \mathbb{E}[\|\bar{m}_{x,t}\|^2] + 2\eta_z \frac{L_g^2}{\rho^2} \frac{1}{K} \mathbb{E}[\|X_t - \bar{X}_t\|_F^2] + 2\eta_z \frac{L_g^2}{\rho^2} \frac{1}{K} \mathbb{E}[\|Z_t - \bar{Z}_t\|_F^2]$$

$$+ \frac{1}{\rho} \mathbb{E}[\langle \nabla_x g(\bar{x}_t, y^*(\bar{x}_t)), \bar{x}_{t+1} - \bar{x}_t \rangle] - \frac{1}{\rho} \mathbb{E}[\langle \nabla \hat{g}(\bar{x}_t), \bar{x}_{t+1} - \bar{x}_t \rangle]$$

$$+ \frac{2\eta_z}{\rho^2} \mathbb{E}[\| \frac{1}{K} \sum_{k=1}^K \nabla_2 g^{(k)}(x_t^{(k)}, z_t^{(k)}) - \frac{1}{K} \sum_{k=1}^K m_{z,1,t}^{(k)} \|^2]$$

$$\leq \frac{1}{\rho} \mathbb{E}[g(\bar{x}_t, \bar{z}_t) - \hat{g}(\bar{x}_t)] - \frac{1}{\rho^2} \frac{\eta_z}{8} \mathbb{E}[\|\nabla_2 g(\bar{x}_t, \bar{z}_t)\|^2] - \frac{\eta_z}{4} \mathbb{E}[\|\bar{m}_{z,t}\|^2]$$

$$+ \left( \frac{\eta_x}{2} + 2\eta_x^2 L_g \frac{1}{\rho} + \frac{\eta_x^2 L_{\hat{g}}}{2} \frac{1}{\rho} \right) \mathbb{E}[\|\bar{m}_{x,t}\|^2] + 2\eta_z \frac{L_g^2}{\rho^2} \frac{1}{K} \mathbb{E}[\|X_t - \bar{X}_t\|_F^2] + 2\eta_z \frac{L_g^2}{\rho^2} \frac{1}{K} \mathbb{E}[\|Z_t - \bar{Z}_t\|_F^2]$$

$$+ \frac{2\eta_z}{\rho^2} \mathbb{E}[\| \frac{1}{K} \sum_{k=1}^K \nabla_2 g^{(k)}(x_t^{(k)}, z_t^{(k)}) - \frac{1}{K} \sum_{k=1}^K m_{z,1,t}^{(k)} \|^2] , \tag{35}$$

where the last step follows from $\eta_x \leq \frac{\mu^2}{4L_g^2} \eta_z$.

$\square$

**Lemma A.7.** *(Restatement of Lemma 4.10) Suppose Assumptions 4.1-4.5 hold, when $\eta_x \leq \eta_y \frac{\mu^2}{4L_{\hat{h}_\rho}^2}$ and $\eta_y \leq \frac{\rho}{2L_{h_\rho}}$, where $L_{h_\rho} = \rho L_f + L_g$, we have that*

$$\frac{1}{\rho} \mathbb{E}[h_\rho(\bar{x}_{t+1}, \bar{y}_{t+1}) - \hat{h}_\rho(\bar{x}_{t+1})] \leq \frac{1}{\rho} \mathbb{E}[h_\rho(\bar{x}_t, \bar{y}_t) - \hat{h}_\rho(\bar{x}_t)] - \frac{1}{\rho^2} \frac{\eta_y}{8} \mathbb{E}[\|\nabla_2 h_\rho(\bar{x}_t, \bar{y}_t)\|^2]$$

$$- \frac{\eta_y}{4} \mathbb{E}[\|\bar{m}_{y,t}\|^2] + \left( \frac{\eta_x}{2} + \frac{1}{\rho} \frac{\eta_x^2 L_{h_\rho}}{2} + \frac{1}{\rho} \frac{3\eta_x^2 L_{h_\rho}}{2} + \frac{1}{\rho} \frac{\eta_x^2 L_{\hat{h}_\rho}}{2} \right) \mathbb{E}[\|\bar{m}_{x,t}\|^2]$$

$$+ 2\eta_y \frac{L_{h_\rho}^2}{\rho^2} \frac{1}{K} \mathbb{E}[\|X_t - \bar{X}_t\|_F^2] + 2\eta_y \frac{L_{h_\rho}^2}{\rho^2} \frac{1}{K} \mathbb{E}[\|Y_t - \bar{Y}_t\|_F^2]$$

$$+ 4\eta_y \mathbb{E}[\| \frac{1}{K} \sum_{k=1}^K \nabla_2 f^{(k)}(x_t^{(k)}, y_t^{(k)}) - \frac{1}{K} \sum_{k=1}^K m_{y,1,t+1}^{(k)} \|^2]$$

$$+ 4\eta_y \frac{1}{\rho^2} \mathbb{E}[\|\frac{1}{K} \sum_{k=1}^{K} \nabla_2 g^{(k)}(x_t^{(k)}, y_t^{(k)}) - \frac{1}{K} \sum_{k=1}^{K} m_{y,2,t+1}^{(k)}\|^2] \, . \tag{36}$$

*Proof.* According to the definition of the penalty function $h_\rho(x, y)$, it is easy to know that this function is $L_{h_\rho}$-smooth where $L_{h_\rho} = \rho L_f + L_g$.

Then, on the one hand, we have

$$\frac{1}{\rho} \mathbb{E}[h_\rho(\bar{x}_{t+1}, \bar{y}_{t+1})] \leq \frac{1}{\rho} \mathbb{E}[h_\rho(\bar{x}_{t+1}, \bar{y}_t)] + \frac{1}{\rho} \mathbb{E}[\langle \nabla_2 h_\rho(\bar{x}_{t+1}, \bar{y}_t), \bar{y}_{t+1} - \bar{y}_t \rangle] + \frac{1}{\rho} \frac{L_{h_\rho}}{2} \mathbb{E}[\|\bar{y}_{t+1} - \bar{y}_t\|^2]$$

$$= \frac{1}{\rho} \mathbb{E}[h_\rho(\bar{x}_{t+1}, \bar{y}_t)] - \eta_y \mathbb{E}[\langle \frac{1}{\rho} \nabla_2 h_\rho(\bar{x}_{t+1}, \bar{y}_t), \bar{m}_{y,t} \rangle] + \frac{1}{\rho} \frac{\eta_y^2 L_{h_\rho}}{2} \mathbb{E}[\|\bar{m}_{y,t}\|^2]$$

$$= \frac{1}{\rho} \mathbb{E}[h_\rho(\bar{x}_{t+1}, \bar{y}_t)] - \frac{\eta_y}{2} \mathbb{E}[\|\frac{1}{\rho} \nabla_2 h_\rho(\bar{x}_{t+1}, \bar{y}_t)\|^2] - \frac{\eta_y}{2} \mathbb{E}[\|\bar{m}_{y,t}\|^2]$$

$$+ \frac{\eta_y}{2} \mathbb{E}[\|\frac{1}{\rho} \nabla_2 h_\rho(\bar{x}_{t+1}, \bar{y}_t) - \bar{m}_{y,t}\|^2] + \frac{1}{\rho} \frac{\eta_y^2 L_{h_\rho}}{2} \mathbb{E}[\|\bar{m}_{y,t}\|^2]$$

$$\leq \frac{1}{\rho} \mathbb{E}[h_\rho(\bar{x}_{t+1}, \bar{y}_t)] - \frac{\eta_y}{2} \mathbb{E}[\|\frac{1}{\rho} \nabla_2 h_\rho(\bar{x}_{t+1}, \bar{y}_t)\|^2] - \frac{\eta_y}{4} \mathbb{E}[\|\bar{m}_{y,t}\|^2] + \frac{\eta_y}{2} \mathbb{E}[\|\frac{1}{\rho} \nabla_2 h_\rho(\bar{x}_{t+1}, \bar{y}_t) - \bar{m}_{y,t}\|^2]$$

$$\leq \frac{1}{\rho} \mathbb{E}[h_\rho(\bar{x}_{t+1}, \bar{y}_t)] - \frac{1}{\rho^2} \frac{\eta_y}{4} \mathbb{E}[\|\nabla_2 h_\rho(\bar{x}_t, \bar{y}_t)\|^2] - \frac{\eta_y}{4} \mathbb{E}[\|\bar{m}_{y,t}\|^2]$$

$$+ \frac{1}{\rho^2} \frac{3\eta_y}{2} \mathbb{E}[\|\nabla_2 h_\rho(\bar{x}_{t+1}, \bar{y}_t) - \nabla_2 h_\rho(\bar{x}_t, \bar{y}_t)\|^2] + \eta_y \mathbb{E}[\|\frac{1}{\rho} \nabla_2 h_\rho(\bar{x}_t, \bar{y}_t) - \bar{m}_{y,t}\|^2]$$

$$\leq \frac{1}{\rho} \mathbb{E}[h_\rho(\bar{x}_{t+1}, \bar{y}_t)] - \frac{1}{\rho^2} \frac{\eta_y}{4} \mathbb{E}[\|\nabla_2 h_\rho(\bar{x}_t, \bar{y}_t)\|^2] - \frac{\eta_y}{4} \mathbb{E}[\|\bar{m}_{y,t}\|^2]$$

$$+ \frac{1}{\rho^2} \frac{3\eta_y \eta_x^2 L_{h_\rho}^2}{2} \mathbb{E}[\|\bar{m}_{x,t}\|^2] + \eta_y \mathbb{E}[\|\frac{1}{\rho} \nabla_2 h_\rho(\bar{x}_t, \bar{y}_t) - \bar{m}_{y,t}\|^2]$$

$$\leq \frac{1}{\rho} \mathbb{E}[h_\rho(\bar{x}_{t+1}, \bar{y}_t)] - \frac{1}{\rho^2} \frac{\eta_y}{4} \mathbb{E}[\|\nabla_2 h_\rho(\bar{x}_t, \bar{y}_t)\|^2] - \frac{\eta_y}{4} \mathbb{E}[\|\bar{m}_{y,t}\|^2]$$

$$+ \frac{1}{\rho} \frac{3\eta_x^2 L_{h_\rho}}{2} \mathbb{E}[\|\bar{m}_{x,t}\|^2] + \eta_y \mathbb{E}[\|\frac{1}{\rho} \nabla_2 h_\rho(\bar{x}_t, \bar{y}_t) - \bar{m}_{y,t}\|^2]$$

$$\leq \frac{1}{\rho} \mathbb{E}[h_\rho(\bar{x}_{t+1}, \bar{y}_t)] - \frac{1}{\rho^2} \frac{\eta_y}{4} \mathbb{E}[\|\nabla_2 h_\rho(\bar{x}_t, \bar{y}_t)\|^2] - \frac{\eta_y}{4} \mathbb{E}[\|\bar{m}_{y,t}\|^2] + \frac{1}{\rho} \frac{3\eta_x^2 L_{h_\rho}}{2} \mathbb{E}[\|\bar{m}_{x,t}\|^2]$$

$$+ 2\eta_y \mathbb{E}[\|\frac{1}{\rho} \nabla_2 h_\rho(\bar{x}_t, \bar{y}_t) - \frac{1}{\rho} \frac{1}{K} \sum_{k=1}^{K} \nabla_2 h_\rho^{(k)}(x_t^{(k)}, y_t^{(k)})\|^2] + 2\eta_y \mathbb{E}[\|\frac{1}{\rho} \frac{1}{K} \sum_{k=1}^{K} \nabla_2 h_\rho^{(k)}(x_t^{(k)}, y_t^{(k)}) - \bar{m}_{y,t}\|^2]$$

$$\leq \frac{1}{\rho} \mathbb{E}[h_\rho(\bar{x}_{t+1}, \bar{y}_t)] - \frac{1}{\rho^2} \frac{\eta_y}{4} \mathbb{E}[\|\nabla_2 h_\rho(\bar{x}_t, \bar{y}_t)\|^2] - \frac{\eta_y}{4} \mathbb{E}[\|\bar{m}_{y,t}\|^2] + \frac{1}{\rho} \frac{3\eta_x^2 L_{h_\rho}}{2} \mathbb{E}[\|\bar{m}_{x,t}\|^2]$$

$$+ 2\eta_y \frac{L_{h_\rho}^2}{\rho^2} \frac{1}{K} \mathbb{E}[\|X_t - \bar{X}_t\|_F^2] + 2\eta_y \frac{L_{h_\rho}^2}{\rho^2} \frac{1}{K} \mathbb{E}[\|Y_t - \bar{Y}_t\|_F^2] + 4\eta_y \mathbb{E}[\|\frac{1}{K} \sum_{k=1}^{K} \nabla_2 f^{(k)}(x_t^{(k)}, y_t^{(k)}) - \frac{1}{K} \sum_{k=1}^{K} m_{y,1,t+1}^{(k)}\|^2]$$

$$+ 4\eta_y \frac{1}{\rho^2} \mathbb{E}[\|\frac{1}{K} \sum_{k=1}^{K} \nabla_2 g^{(k)}(x_t^{(k)}, y_t^{(k)}) - \frac{1}{K} \sum_{k=1}^{K} m_{y,2,t+1}^{(k)}\|^2] \, , \tag{37}$$

where the fourth step follows from $\eta_y \leq \frac{\rho}{2L_{h_\rho}}$.

On the other hand, we have

$$\frac{1}{\rho} \mathbb{E}[h_\rho(\bar{x}_{t+1}, \bar{y}_t)] \leq \frac{1}{\rho} \mathbb{E}[h_\rho(\bar{x}_t, \bar{y}_t)] + \frac{1}{\rho} \mathbb{E}[\langle \nabla_1 h_\rho(\bar{x}_t, \bar{y}_t), \bar{x}_{t+1} - \bar{x}_t \rangle] + \frac{1}{\rho} \frac{L_{h_\rho}}{2} \mathbb{E}[\|\bar{x}_{t+1} - \bar{x}_t\|^2]$$

$$= \frac{1}{\rho} \mathbb{E}[h_\rho(\bar{x}_t, \bar{y}_t)] + \mathbb{E}[\langle \frac{1}{\rho} (\nabla_1 h_\rho(\bar{x}_t, \bar{y}_t) - \nabla \hat{h}_\rho(\bar{x}_t)), \bar{x}_{t+1} - \bar{x}_t \rangle]$$

$$+ \mathbb{E}[\langle \frac{1}{\rho}\nabla\hat{h}_\rho(\bar{x}_t), \bar{x}_{t+1} - \bar{x}_t\rangle] + \frac{1}{\rho}\frac{L_{h_\rho}}{2}\mathbb{E}[\|\bar{x}_{t+1} - \bar{x}_t\|^2]$$

$$= \frac{1}{\rho}\mathbb{E}[h_\rho(\bar{x}_t, \bar{y}_t)] + \frac{1}{\rho^2}\frac{\eta_x}{2}\mathbb{E}[\|\nabla_1 h_\rho(\bar{x}_t, \bar{y}_t) - \nabla\hat{h}_\rho(\bar{x}_t)\|^2] + \frac{\eta_x}{2}\mathbb{E}[\|\bar{m}_{x,t}\|^2]$$

$$+ \frac{1}{\rho}\mathbb{E}[\langle\nabla\hat{h}_\rho(\bar{x}_t), \bar{x}_{t+1} - \bar{x}_t\rangle] + \frac{1}{\rho}\frac{\eta_x^2 L_{h_\rho}}{2}\mathbb{E}[\|\bar{m}_{x,t}\|^2]$$

$$= \frac{1}{\rho}\mathbb{E}[h_\rho(\bar{x}_t, \bar{y}_t)] + \frac{1}{\rho^2}\frac{\eta_x}{2}\mathbb{E}[\|\nabla_1 h_\rho(\bar{x}_t, \bar{y}_t) - \nabla\hat{h}_\rho(\bar{x}_t)\|^2] + \frac{1}{\rho}\mathbb{E}[\langle\nabla\hat{h}_\rho(\bar{x}_t), \bar{x}_{t+1} - \bar{x}_t\rangle] + \left(\frac{\eta_x}{2} + \frac{\eta_x^2 L_{h_\rho}}{2\rho}\right)\mathbb{E}[\|\bar{m}_{x,t}\|^2]$$

$$\leq \frac{1}{\rho}\mathbb{E}[h_\rho(\bar{x}_t, \bar{y}_t)] + \frac{1}{\rho^2}\frac{\eta_x L_{h_\rho}^2}{2}\mathbb{E}[\|\bar{y}_t - y_\rho^*(\bar{x}_t)\|^2] + \frac{1}{\rho}\mathbb{E}[\langle\nabla\hat{h}_\rho(\bar{x}_t), \bar{x}_{t+1} - \bar{x}_t\rangle] + \left(\frac{\eta_x}{2} + \frac{\eta_x^2 L_{h_\rho}}{2\rho}\right)\mathbb{E}[\|\bar{m}_{x,t}\|^2]$$

$$\leq \frac{1}{\rho}\mathbb{E}[h_\rho(\bar{x}_t, \bar{y}_t)] + \frac{1}{\rho^2}\frac{\eta_x L_{h_\rho}^2}{2\mu^2}\mathbb{E}[\|\nabla_2 h_\rho(\bar{x}_t, \bar{y}_t)\|^2] + \frac{1}{\rho}\mathbb{E}[\langle\nabla\hat{h}_\rho(\bar{x}_t), \bar{x}_{t+1} - \bar{x}_t\rangle] + \left(\frac{\eta_x}{2} + \frac{\eta_x^2 L_{h_\rho}}{2\rho}\right)\mathbb{E}[\|\bar{m}_{x,t}\|^2], \quad (38)$$

where the last step follows from Lemma A.4 and Assumption 4.3.

By combining the above two inequalities, we have

$$\frac{1}{\rho}\mathbb{E}[h_\rho(\bar{x}_{t+1}, \bar{y}_{t+1})] \leq \frac{1}{\rho}\mathbb{E}[h_\rho(\bar{x}_{t+1}, \bar{y}_t)] - \frac{1}{\rho^2}\frac{\eta_y}{4}\mathbb{E}[\|\nabla_2 h_\rho(\bar{x}_t, \bar{y}_t)\|^2] - \frac{\eta_y}{4}\mathbb{E}[\|\bar{m}_{y,t}\|^2] + \frac{1}{\rho}\frac{3\eta_x^2 L_{h_\rho}}{2}\mathbb{E}[\|\bar{m}_{x,t}\|^2]$$

$$+ 2\eta_y\frac{L_{h_\rho}^2}{\rho^2}\frac{1}{K}\mathbb{E}[\|X_t - \bar{X}_t\|_F^2] + 2\eta_y\frac{L_{h_\rho}^2}{\rho^2}\frac{1}{K}\mathbb{E}[\|Y_t - \bar{Y}_t\|_F^2] + 4\eta_y\mathbb{E}[\|\frac{1}{K}\sum_{k=1}^K \nabla_2 f^{(k)}(x_t^{(k)}, y_t^{(k)}) - \frac{1}{K}\sum_{k=1}^K m_{y,1,t+1}^{(k)}\|^2]$$

$$+ 4\eta_y\frac{1}{\rho^2}\mathbb{E}[\|\frac{1}{K}\sum_{k=1}^K \nabla_2 g^{(k)}(x_t^{(k)}, y_t^{(k)}) - \frac{1}{K}\sum_{k=1}^K m_{y,2,t+1}^{(k)}\|^2]$$

$$\leq \frac{1}{\rho}\mathbb{E}[h_\rho(\bar{x}_t, \bar{y}_t)] - \frac{1}{\rho^2}\frac{\eta_y}{4}\mathbb{E}[\|\nabla_2 h_\rho(\bar{x}_t, \bar{y}_t)\|^2] - \frac{\eta_y}{4}\mathbb{E}[\|\bar{m}_{y,t}\|^2] + \frac{1}{\rho}\frac{3\eta_x^2 L_{h_\rho}}{2}\mathbb{E}[\|\bar{m}_{x,t}\|^2]$$

$$+ 2\eta_y\frac{L_{h_\rho}^2}{\rho^2}\frac{1}{K}\mathbb{E}[\|X_t - \bar{X}_t\|_F^2] + 2\eta_y\frac{L_{h_\rho}^2}{\rho^2}\frac{1}{K}\mathbb{E}[\|Y_t - \bar{Y}_t\|_F^2] + 4\eta_y\mathbb{E}[\|\frac{1}{K}\sum_{k=1}^K \nabla_2 f^{(k)}(x_t^{(k)}, y_t^{(k)}) - \frac{1}{K}\sum_{k=1}^K m_{y,1,t+1}^{(k)}\|^2]$$

$$+ 4\eta_y\frac{1}{\rho^2}\mathbb{E}[\|\frac{1}{K}\sum_{k=1}^K \nabla_2 g^{(k)}(x_t^{(k)}, y_t^{(k)}) - \frac{1}{K}\sum_{k=1}^K m_{y,2,t+1}^{(k)}\|^2]$$

$$+ \frac{1}{\rho^2}\frac{\eta_x L_{h_\rho}^2}{2\mu^2}\mathbb{E}[\|\nabla_2 h_\rho(\bar{x}_t, \bar{y}_t)\|^2] + \frac{1}{\rho}\mathbb{E}[\langle\nabla\hat{h}_\rho(\bar{x}_t), \bar{x}_{t+1} - \bar{x}_t\rangle] + \left(\frac{\eta_x}{2} + \frac{1}{\rho}\frac{\eta_x^2 L_{h_\rho}}{2}\right)\mathbb{E}[\|\bar{m}_{x,t}\|^2]$$

$$= \frac{1}{\rho}\mathbb{E}[h_\rho(\bar{x}_t, \bar{y}_t)] - \frac{1}{\rho^2}\left(\frac{\eta_y}{4} - \frac{\eta_x L_{h_\rho}^2}{2\mu^2}\right)\mathbb{E}[\|\nabla_2 h_\rho(\bar{x}_t, \bar{y}_t)\|^2] - \frac{\eta_y}{4}\mathbb{E}[\|\bar{m}_{y,t}\|^2] + \frac{1}{\rho}\mathbb{E}[\langle\nabla\hat{h}_\rho(\bar{x}_t), \bar{x}_{t+1} - \bar{x}_t\rangle]$$

$$+ \left(\frac{\eta_x}{2} + \frac{1}{\rho}\frac{\eta_x^2 L_{h_\rho}}{2} + \frac{1}{\rho}\frac{3\eta_x^2 L_{h_\rho}}{2}\right)\mathbb{E}[\|\bar{m}_{x,t}\|^2] + 2\eta_y\frac{L_{h_\rho}^2}{\rho^2}\frac{1}{K}\mathbb{E}[\|X_t - \bar{X}_t\|_F^2] + 2\eta_y\frac{L_{h_\rho}^2}{\rho^2}\frac{1}{K}\mathbb{E}[\|Y_t - \bar{Y}_t\|_F^2]$$

$$+ 4\eta_y\mathbb{E}[\|\frac{1}{K}\sum_{k=1}^K \nabla_2 f^{(k)}(x_t^{(k)}, y_t^{(k)}) - \frac{1}{K}\sum_{k=1}^K m_{y,1,t+1}^{(k)}\|^2]$$

$$+ 4\eta_y\frac{1}{\rho^2}\mathbb{E}[\|\frac{1}{K}\sum_{k=1}^K \nabla_2 g^{(k)}(x_t^{(k)}, y_t^{(k)}) - \frac{1}{K}\sum_{k=1}^K m_{y,2,t+1}^{(k)}\|^2]. \quad (39)$$

Similarly, due to the smoothness of $\hat{h}_\rho(x_t)$, we have

$$\frac{1}{\rho}\hat{h}_\rho(\bar{x}_t) - \frac{1}{\rho}\hat{h}_\rho(\bar{x}_{t+1}) \leq -\frac{1}{\rho}\langle\nabla\hat{h}_\rho(\bar{x}_t), \bar{x}_{t+1} - \bar{x}_t\rangle + \frac{1}{\rho}\frac{\eta_x^2 L_{\hat{h}_\rho}}{2}\|\bar{m}_{x,t}\|^2. \quad (40)$$

As a result, we have

$$\frac{1}{\rho}\mathbb{E}[h_\rho(\bar{x}_{t+1}, \bar{y}_{t+1}) - \hat{h}_\rho(\bar{x}_{t+1})]$$

$$= \frac{1}{\rho}\mathbb{E}[h_\rho(\bar{x}_{t+1}, \bar{y}_{t+1}) - h_\rho(\bar{x}_t, \bar{y}_t) + h_\rho(\bar{x}_t, \bar{y}_t) - \hat{h}_\rho(\bar{x}_t) + \hat{h}_\rho(\bar{x}_t) - \hat{h}_\rho(\bar{x}_{t+1})]$$

$$= \frac{1}{\rho}\mathbb{E}[h_\rho(\bar{x}_t, \bar{y}_t) - \hat{h}_\rho(\bar{x}_t)] + \frac{1}{\rho}\mathbb{E}[h_\rho(\bar{x}_{t+1}, \bar{y}_{t+1}) - h_\rho(\bar{x}_t, \bar{y}_t)] + \frac{1}{\rho}\mathbb{E}[\hat{h}_\rho(\bar{x}_t) - \hat{h}_\rho(\bar{x}_{t+1})]$$

$$\leq \frac{1}{\rho}\mathbb{E}[h_\rho(\bar{x}_t, \bar{y}_t) - \hat{h}_\rho(\bar{x}_t)] - \frac{1}{\rho^2}\left(\frac{\eta_y}{4} - \frac{\eta_x L_{h_\rho}^2}{2\mu^2}\right)\mathbb{E}[\|\nabla_2 h_\rho(\bar{x}_t, \bar{y}_t)\|^2] - \frac{\eta_y}{4}\mathbb{E}[\|\bar{m}_{y,t}\|^2]$$

$$+ \left(\frac{\eta_x}{2} + \frac{1}{\rho}\frac{\eta_x^2 L_{h_\rho}}{2} + \frac{1}{\rho}\frac{3\eta_x^2 L_{h_\rho}}{2}\right)\mathbb{E}[\|\bar{m}_{x,t}\|^2] + 2\eta_y\frac{L_{h_\rho}^2}{\rho^2}\frac{1}{K}\mathbb{E}[\|X_t - \bar{X}_t\|_F^2] + 2\eta_y\frac{L_{h_\rho}^2}{\rho^2}\frac{1}{K}\mathbb{E}[\|Y_t - \bar{Y}_t\|_F^2]$$

$$+ 4\eta_y\mathbb{E}[\|\frac{1}{K}\sum_{k=1}^{K}\nabla_2 f^{(k)}(x_t^{(k)}, y_t^{(k)}) - \frac{1}{K}\sum_{k=1}^{K}m_{y,1,t+1}^{(k)}\|^2]$$

$$+ 4\eta_y\frac{1}{\rho^2}\mathbb{E}[\|\frac{1}{K}\sum_{k=1}^{K}\nabla_2 g^{(k)}(x_t^{(k)}, y_t^{(k)}) - \frac{1}{K}\sum_{k=1}^{K}m_{y,2,t+1}^{(k)}\|^2]$$

$$+ \frac{1}{\rho}\mathbb{E}[\langle\nabla\hat{h}_\rho(\bar{x}_t), \bar{x}_{t+1} - \bar{x}_t\rangle] - \frac{1}{\rho}\mathbb{E}[\langle\nabla\hat{h}_\rho(\bar{x}_t), \bar{x}_{t+1} - \bar{x}_t\rangle] + \frac{1}{\rho}\frac{\eta_x^2 L_{\hat{h}_\rho}}{2}\mathbb{E}[\|\bar{m}_{x,t}\|^2]$$

$$\leq \frac{1}{\rho}\mathbb{E}[h_\rho(\bar{x}_t, \bar{y}_t) - \hat{h}_\rho(\bar{x}_t)] - \frac{1}{\rho^2}\frac{\eta_y}{8}\mathbb{E}[\|\nabla_2 h_\rho(\bar{x}_t, \bar{y}_t)\|^2] - \frac{\eta_y}{4}\mathbb{E}[\|\bar{m}_{y,t}\|^2]$$

$$+ \left(\frac{\eta_x}{2} + \frac{1}{\rho}\frac{\eta_x^2 L_{h_\rho}}{2} + \frac{1}{\rho}\frac{3\eta_x^2 L_{h_\rho}}{2} + \frac{1}{\rho}\frac{\eta_x^2 L_{\hat{h}_\rho}}{2}\right)\mathbb{E}[\|\bar{m}_{x,t}\|^2]$$

$$+ 2\eta_y\frac{L_{h_\rho}^2}{\rho^2}\frac{1}{K}\mathbb{E}[\|X_t - \bar{X}_t\|_F^2] + 2\eta_y\frac{L_{h_\rho}^2}{\rho^2}\frac{1}{K}\mathbb{E}[\|Y_t - \bar{Y}_t\|_F^2]$$

$$+ 4\eta_y\mathbb{E}[\|\frac{1}{K}\sum_{k=1}^{K}\nabla_2 f^{(k)}(x_t^{(k)}, y_t^{(k)}) - \frac{1}{K}\sum_{k=1}^{K}m_{y,1,t+1}^{(k)}\|^2]$$

$$+ 4\eta_y\frac{1}{\rho^2}\mathbb{E}[\|\frac{1}{K}\sum_{k=1}^{K}\nabla_2 g^{(k)}(x_t^{(k)}, y_t^{(k)}) - \frac{1}{K}\sum_{k=1}^{K}m_{y,2,t+1}^{(k)}\|^2], \tag{41}$$

where the last step follows from $\eta_x \leq \eta_y\frac{\mu^2}{4L_{h_\rho}^2}$.

$\square$

## A.5. Gradient Estimation Error

**Lemma A.8.** *Suppose Assumptions 4.1-4.5 hold, when $\eta_x \leq \frac{1}{\sqrt{\gamma_x}}$, $\eta_y \leq \frac{1}{\sqrt{\gamma_y}}$, and $\eta_z \leq \frac{1}{\sqrt{\gamma_z}}$, we have that*

$$\mathbb{E}[\|\frac{1}{K}\sum_{k=1}^{K}\nabla_1 f^{(k)}(x_{t+1}^{(k)}, y_{t+1}^{(k)}) - \frac{1}{K}\sum_{k=1}^{K}m_{x,1,t+1}^{(k)}\|^2] \leq (1 - \gamma_x\eta_x^2)\mathbb{E}[\|\frac{1}{K}\sum_{k=1}^{K}m_{x,1,t}^{(k)} - \frac{1}{K}\sum_{k=1}^{K}\nabla_1 f^{(k)}(x_t^{(k)}, y_t^{(k)})\|^2]$$

$$+ 2L_f^2\frac{1}{K^2}\mathbb{E}[\|X_{t+1} - X_t\|_F^2] + 2L_f^2\frac{1}{K^2}\mathbb{E}[\|Y_{t+1} - Y_t\|_F^2] + 2\gamma_x^2\eta_x^4\sigma^2\frac{1}{K}, \tag{42}$$

$$\mathbb{E}[\|\frac{1}{K}\sum_{k=1}^{K}\nabla_1 g^{(k)}(x_{t+1}^{(k)}, y_{t+1}^{(k)}) - \frac{1}{K}\sum_{k=1}^{K}m_{x,2,t+1}^{(k)}\|^2] \leq (1 - \gamma_x\eta_x^2)\mathbb{E}[\|\frac{1}{K}\sum_{k=1}^{K}m_{x,2,t}^{(k)} - \frac{1}{K}\sum_{k=1}^{K}\nabla_1 g^{(k)}(x_t^{(k)}, y_t^{(k)})\|^2]$$

$$+ 2L_g^2\frac{1}{K^2}\mathbb{E}[\|X_{t+1} - X_t\|_F^2] + 2L_g^2\frac{1}{K^2}\mathbb{E}[\|Y_{t+1} - Y_t\|_F^2] + 2\gamma_x^2\eta_x^4\sigma^2\frac{1}{K}, \tag{43}$$

$$\mathbb{E}[\|\frac{1}{K}\sum_{k=1}^{K}\nabla_1 g^{(k)}(x_{t+1}^{(k)}, z_{t+1}^{(k)}) - \frac{1}{K}\sum_{k=1}^{K}m_{x,3,t+1}^{(k)}\|^2] \leq (1-\gamma_x\eta_x^2)\mathbb{E}[\|\frac{1}{K}\sum_{k=1}^{K}m_{x,3,t}^{(k)} - \frac{1}{K}\sum_{k=1}^{K}\nabla_1 g^{(k)}(x_t^{(k)}, z_t^{(k)})\|^2]$$

$$+ 2L_g^2\frac{1}{K^2}\mathbb{E}[\|X_{t+1} - X_t\|_F^2] + 2L_g^2\frac{1}{K^2}\mathbb{E}[\|Z_{t+1} - Z_t\|_F^2] + 2\gamma_x^2\eta_x^4\sigma^2\frac{1}{K}, \tag{44}$$

$$\mathbb{E}[\|\frac{1}{K}\sum_{k=1}^{K}\nabla_2 f^{(k)}(x_{t+1}^{(k)}, y_{t+1}^{(k)}) - \frac{1}{K}\sum_{k=1}^{K}m_{y,1,t+1}^{(k)}\|^2] \leq (1-\gamma_y\eta_y^2)\mathbb{E}[\|\frac{1}{K}\sum_{k=1}^{K}m_{y,1,t}^{(k)} - \frac{1}{K}\sum_{k=1}^{K}\nabla_2 f^{(k)}(x_t^{(k)}, y_t^{(k)})\|^2]$$

$$+ 2L_f^2\frac{1}{K^2}\mathbb{E}[\|X_{t+1} - X_t\|_F^2] + 2L_f^2\frac{1}{K^2}\mathbb{E}[\|Y_{t+1} - Y_t\|_F^2] + 2\gamma_y^2\eta_y^4\sigma^2\frac{1}{K}, \tag{45}$$

$$\mathbb{E}[\|\frac{1}{K}\sum_{k=1}^{K}\nabla_2 g^{(k)}(x_{t+1}^{(k)}, y_{t+1}^{(k)}) - \frac{1}{K}\sum_{k=1}^{K}m_{y,2,t+1}^{(k)}\|^2] \leq (1-\gamma_y\eta_y^2)\mathbb{E}[\|\frac{1}{K}\sum_{k=1}^{K}m_{y,2,t}^{(k)} - \frac{1}{K}\sum_{k=1}^{K}\nabla_2 g^{(k)}(x_t^{(k)}, y_t^{(k)})\|^2]$$

$$+ 2L_g^2\frac{1}{K^2}\mathbb{E}[\|X_{t+1} - X_t\|_F^2] + 2L_g^2\frac{1}{K^2}\mathbb{E}[\|Y_{t+1} - Y_t\|_F^2] + 2\gamma_y^2\eta_y^4\sigma^2\frac{1}{K}, \tag{46}$$

$$\mathbb{E}[\|\frac{1}{K}\sum_{k=1}^{K}\nabla_2 g^{(k)}(x_{t+1}^{(k)}, z_{t+1}^{(k)}) - \frac{1}{K}\sum_{k=1}^{K}m_{z,1,t+1}^{(k)}\|^2] \leq (1-\gamma_z\eta_z^2)\mathbb{E}[\|\frac{1}{K}\sum_{k=1}^{K}m_{z,1,t}^{(k)} - \frac{1}{K}\sum_{k=1}^{K}\nabla_2 g^{(k)}(x_t^{(k)}, z_t^{(k)})\|^2]$$

$$+ 2L_g^2\frac{1}{K^2}\mathbb{E}[\|X_{t+1} - X_t\|_F^2] + 2L_g^2\frac{1}{K^2}\mathbb{E}[\|Z_{t+1} - Z_t\|_F^2] + 2\gamma_z^2\eta_z^4\sigma^2\frac{1}{K}. \tag{47}$$

Bounding the gradient estimation error has been studied in the existing literature (Cutkosky & Orabona, 2019; Khanduri et al., 2021a; Gao et al., 2023), we follow them to include the proof below for completeness.

*Proof.*

$$\mathbb{E}[\|\frac{1}{K}\sum_{k=1}^{K}\nabla_1 f^{(k)}(x_{t+1}^{(k)}, y_{t+1}^{(k)}) - \frac{1}{K}\sum_{k=1}^{K}m_{x,1,t+1}^{(k)}\|^2]$$

$$= \mathbb{E}[\|\frac{1}{K}\sum_{k=1}^{K}((1-\gamma_x\eta_x^2)(m_{x,1,t}^{(k)} - \nabla_1 f^{(k)}(x_t^{(k)}, y_t^{(k)}; \xi_{t+1}^{(k)})) + \nabla_1 f^{(k)}(x_{t+1}^{(k)}, y_{t+1}^{(k)}; \xi_{t+1}^{(k)})) - \frac{1}{K}\sum_{k=1}^{K}\nabla_1 f^{(k)}(x_{t+1}^{(k)}, y_{t+1}^{(k)})\|^2]$$

$$= \mathbb{E}[\|\frac{1}{K}\sum_{k=1}^{K}\left((1-\gamma_x\eta_x^2)(m_{x,1,t}^{(k)} - \nabla_1 f^{(k)}(x_t^{(k)}, y_t^{(k)}))\right)\|^2]$$

$$+ \mathbb{E}[\|(1-\gamma_x\eta_x^2)\frac{1}{K}\sum_{k=1}^{K}\left(\nabla_1 f^{(k)}(x_t^{(k)}, y_t^{(k)}) - \nabla_1 f^{(k)}(x_t^{(k)}, y_t^{(k)}; \xi_{t+1}^{(k)})\right)$$

$$+ \nabla_1 f^{(k)}(x_{t+1}^{(k)}, y_{t+1}^{(k)}; \xi_{t+1}^{(k)}) - \nabla_1 f^{(k)}(x_{t+1}^{(k)}, y_{t+1}^{(k)})\Big)$$

$$+ \gamma_x\eta_x^2\frac{1}{K}\sum_{k=1}^{K}(\nabla_1 f^{(k)}(x_{t+1}^{(k)}, y_{t+1}^{(k)}; \xi_{t+1}^{(k)}) - \nabla_1 f^{(k)}(x_{t+1}^{(k)}, y_{t+1}^{(k)}))\|^2]$$

$$\leq (1-\gamma_x\eta_x^2)^2\mathbb{E}[\|\frac{1}{K}\sum_{k=1}^{K}(m_{x,1,t}^{(k)} - \nabla_1 f^{(k)}(x_t^{(k)}, y_t^{(k)}))\|^2]$$

$$+ 2(1-\gamma_x\eta_x^2)^2\mathbb{E}[\|\frac{1}{K}\sum_{k=1}^{K}\left(\nabla_1 f^{(k)}(x_t^{(k)}, y_t^{(k)}) - \nabla_1 f^{(k)}(x_t^{(k)}, y_t^{(k)}; \xi_{t+1}^{(k)})\right)$$

$$+ \nabla_1 f^{(k)}(x_{t+1}^{(k)}, y_{t+1}^{(k)}; \xi_{t+1}^{(k)}) - \nabla_1 f^{(k)}(x_{t+1}^{(k)}, y_{t+1}^{(k)})\Big)\|^2]$$

$$+ 2\gamma_x^2\eta_x^4 \mathbb{E}[\| \frac{1}{K}\sum_{k=1}^{K}(\nabla_1 f^{(k)}(x_{t+1}^{(k)}, y_{t+1}^{(k)}; \xi_{t+1}^{(k)}) - \nabla_1 f^{(k)}(x_{t+1}^{(k)}, y_{t+1}^{(k)}))\|^2]$$

$$\leq (1 - \gamma_x\eta_x^2)\mathbb{E}[\| \frac{1}{K}\sum_{k=1}^{K} m_{x,1,t}^{(k)} - \frac{1}{K}\sum_{k=1}^{K}\nabla_1 f^{(k)}(x_t^{(k)}, y_t^{(k)})\|^2]$$

$$+ 2\frac{1}{K^2}\sum_{k=1}^{K}\mathbb{E}[\|\nabla_1 f^{(k)}(x_{t+1}^{(k)}, y_{t+1}^{(k)}; \xi_{t+1}^{(k)}) - \nabla_1 f^{(k)}(x_t^{(k)}, y_t^{(k)}; \xi_{t+1}^{(k)})\|^2]$$

$$+ 2\gamma_x^2\eta_x^4\frac{1}{K^2}\sum_{k=1}^{K}\mathbb{E}[\|\nabla_1 f^{(k)}(x_t^{(k)}, y_t^{(k)}; \xi_{t+1}^{(k)}) - \nabla_1 f^{(k)}(x_t^{(k)}, y_t^{(k)})\|^2]$$

$$\leq (1 - \gamma_x\eta_x^2)\mathbb{E}[\| \frac{1}{K}\sum_{k=1}^{K} m_{x,1,t}^{(k)} - \frac{1}{K}\sum_{k=1}^{K}\nabla_1 f^{(k)}(x_t^{(k)}, y_t^{(k)})\|^2]$$

$$+ 2L_f^2\frac{1}{K^2}\mathbb{E}[\|X_{t+1} - X_t\|_F^2] + 2L_f^2\frac{1}{K^2}\mathbb{E}[\|Y_{t+1} - Y_t\|_F^2] + 2\gamma_x^2\eta_x^4\sigma^2\frac{1}{K} . \tag{48}$$

The other inequalities can be proved in the same way so that we omit their detailed proof. $\qquad\square$

**Lemma A.9.** *Suppose Assumptions 4.1-4.5 hold, when $\eta_x \leq \frac{1}{\sqrt{\gamma_x}}$, $\eta_y \leq \frac{1}{\sqrt{\gamma_y}}$, and $\eta_z \leq \frac{1}{\sqrt{\gamma_z}}$, we have that*

$$\frac{1}{K}\sum_{k=1}^{K}\mathbb{E}[\|\nabla_1 f^{(k)}(x_{t+1}^{(k)}, y_{t+1}^{(k)}) - m_{x,1,t+1}^{(k)}\|^2] \leq (1 - \gamma_x\eta_x^2)\frac{1}{K}\sum_{k=1}^{K}\mathbb{E}[\|m_{x,1,t}^{(k)} - \nabla_1 f^{(k)}(x_t^{(k)}, y_t^{(k)})\|^2]$$

$$+ 2L_f^2\frac{1}{K}\mathbb{E}[\|X_{t+1} - X_t\|_F^2] + 2L_f^2\frac{1}{K}\mathbb{E}[\|Y_{t+1} - Y_t\|_F^2] + 2\gamma_x^2\eta_x^4\sigma^2 , \tag{49}$$

$$\frac{1}{K}\sum_{k=1}^{K}\mathbb{E}[\|\nabla_1 g^{(k)}(x_{t+1}^{(k)}, y_{t+1}^{(k)}) - m_{x,2,t+1}^{(k)}\|^2] \leq (1 - \gamma_x\eta_x^2)\frac{1}{K}\sum_{k=1}^{K}\mathbb{E}[\|m_{x,2,t}^{(k)} - \nabla_1 g^{(k)}(x_t^{(k)}, y_t^{(k)})\|^2]$$

$$+ 2L_g^2\frac{1}{K}\mathbb{E}[\|X_{t+1} - X_t\|_F^2] + 2L_g^2\frac{1}{K}\mathbb{E}[\|Y_{t+1} - Y_t\|_F^2] + 2\gamma_x^2\eta_x^4\sigma^2 , \tag{50}$$

$$\frac{1}{K}\sum_{k=1}^{K}\mathbb{E}[\|\nabla_1 g^{(k)}(x_{t+1}^{(k)}, z_{t+1}^{(k)}) - m_{x,3,t+1}^{(k)}\|^2] \leq (1 - \gamma_x\eta_x^2)\frac{1}{K}\sum_{k=1}^{K}\mathbb{E}[\|m_{x,3,t}^{(k)} - \nabla_1 g^{(k)}(x_t^{(k)}, z_t^{(k)})\|^2]$$

$$+ 2L_g^2\frac{1}{K}\mathbb{E}[\|X_{t+1} - X_t\|_F^2] + 2L_g^2\frac{1}{K}\mathbb{E}[\|Z_{t+1} - Z_t\|_F^2] + 2\gamma_x^2\eta_x^4\sigma^2 , \tag{51}$$

$$\frac{1}{K}\sum_{k=1}^{K}\mathbb{E}[\|\nabla_2 f^{(k)}(x_{t+1}^{(k)}, y_{t+1}^{(k)}) - m_{y,1,t+1}^{(k)}\|^2] \leq (1 - \gamma_y\eta_y^2)\frac{1}{K}\sum_{k=1}^{K}\mathbb{E}[\|m_{y,1,t}^{(k)} - \nabla_2 f^{(k)}(x_t^{(k)}, y_t^{(k)})\|^2]$$

$$+ 2L_f^2\frac{1}{K}\mathbb{E}[\|X_{t+1} - X_t\|_F^2] + 2L_f^2\frac{1}{K}\mathbb{E}[\|Y_{t+1} - Y_t\|_F^2] + 2\gamma_y^2\eta_y^4\sigma^2 , \tag{52}$$

$$\frac{1}{K}\sum_{k=1}^{K}\mathbb{E}[\|\nabla_2 g^{(k)}(x_{t+1}^{(k)}, y_{t+1}^{(k)}) - m_{y,2,t+1}^{(k)}\|^2] \leq (1 - \gamma_y\eta_y^2)\frac{1}{K}\sum_{k=1}^{K}\mathbb{E}[\|m_{y,2,t}^{(k)} - \nabla_2 g^{(k)}(x_t^{(k)}, y_t^{(k)})\|^2]$$

$$+ 2L_g^2\frac{1}{K}\mathbb{E}[\|X_{t+1} - X_t\|_F^2] + 2L_g^2\frac{1}{K}\mathbb{E}[\|Y_{t+1} - Y_t\|_F^2] + 2\gamma_y^2\eta_y^4\sigma^2 , \tag{53}$$

$$\frac{1}{K}\sum_{k=1}^{K}\mathbb{E}[\|\nabla_2 g^{(k)}(x_{t+1}^{(k)}, z_{t+1}^{(k)}) - m_{z,1,t+1}^{(k)}\|^2] \leq (1 - \gamma_z\eta_z^2)\frac{1}{K}\sum_{k=1}^{K}\mathbb{E}[\|m_{z,1,t}^{(k)} - \nabla_2 g^{(k)}(x_t^{(k)}, z_t^{(k)})\|^2]$$

$$+ 2L_g^2\frac{1}{K}\mathbb{E}[\|X_{t+1} - X_t\|_F^2] + 2L_g^2\frac{1}{K}\mathbb{E}[\|Z_{t+1} - Z_t\|_F^2] + 2\gamma_z^2\eta_z^4\sigma^2 . \tag{54}$$

This lemma can be proved by following Lemma A.8.

## A.6. Consensus Error

**Lemma A.10.** *Suppose Assumptions 4.1-4.5 hold, we have that*

$$\mathbb{E}[\|X_{t+1} - \bar{X}_{t+1}\|_F^2] \le \lambda \mathbb{E}[\|X_t - \bar{X}_t\|_F^2] + \frac{\eta_x^2}{1-\lambda}\mathbb{E}[\|U_{x,t} - \bar{U}_{x,t}\|_F^2] \,, \tag{55}$$

$$\mathbb{E}[\|Y_{t+1} - \bar{Y}_{t+1}\|_F^2] \le \lambda \mathbb{E}[\|Y_t - \bar{Y}_t\|_F^2] + \frac{\eta_y^2}{1-\lambda}\mathbb{E}[\|U_{y,t} - \bar{U}_{y,t}\|_F^2] \,, \tag{56}$$

$$\mathbb{E}[\|Z_{t+1} - \bar{Z}_{t+1}\|_F^2] \le \lambda \mathbb{E}[\|Z_t - \bar{Z}_t\|_F^2] + \frac{\eta_z^2}{1-\lambda}\mathbb{E}[\|U_{z,t} - \bar{U}_{z,t}\|_F^2] \,. \tag{57}$$

*Proof.*

$$\begin{aligned}
&\mathbb{E}[\|X_{t+1} - \bar{X}_{t+1}\|_F^2] \\
&= \mathbb{E}[\|X_t W - \eta_x U_{x,t} - (\bar{X}_t - \eta_x \bar{U}_{x,t})\|_F^2] \\
&\le \frac{1}{\lambda}\mathbb{E}[\|X_t W - \bar{X}_t\|_F^2] + \frac{\eta_x^2}{1-\lambda}\mathbb{E}[\|U_{x,t} - \bar{U}_{x,t}\|_F^2] \\
&\le \lambda \mathbb{E}[\|X_t - \bar{X}_t\|_F^2] + \frac{\eta_x^2}{1-\lambda}\mathbb{E}[\|U_{x,t} - \bar{U}_{x,t}\|_F^2] \,,
\end{aligned} \tag{58}$$

where the second step follows from the fact $\|a + b\|^2 \le (1 + c)\|a\|^2 + (1 + 1/c)\|b\|^2$ with $c = \frac{1}{\lambda} - 1$. $\qquad\square$

**Lemma A.11.** *Suppose Assumptions 4.1-4.5 hold, we have that*

$$\begin{aligned}
\mathbb{E}[\|U_{x,t+1} - \bar{U}_{x,t+1}\|_F^2] &\le \lambda \mathbb{E}[\|U_{x,t} - \bar{U}_{x,t}\|_F^2] + \frac{9}{1-\lambda}(L_f^2 + 2\frac{L_g^2}{\rho^2})\mathbb{E}[\|X_{t+1} - X_t\|_F^2] \\
&+ \frac{9}{1-\lambda}(L_f^2 + \frac{L_g^2}{\rho^2})\mathbb{E}[\|Y_{t+1} - Y_t\|_F^2] + \frac{9}{1-\lambda}\frac{L_g^2}{\rho^2}\mathbb{E}[\|Z_{t+1} - Z_t\|_F^2] + \frac{9K\gamma_x^2\eta_x^4\sigma^2}{1-\lambda}(1 + \frac{2}{\rho^2}) \\
&+ \frac{9\gamma_x^2\eta_x^4}{1-\lambda}\sum_{k=1}^{K}\mathbb{E}[\|m_{x,1,t}^{(k)} - \nabla_1 f^{(k)}(x_t^{(k)}, y_t^{(k)})\|^2] \\
&+ \frac{9\gamma_x^2\eta_x^4}{1-\lambda}\frac{1}{\rho^2}\sum_{k=1}^{K}\mathbb{E}[\|m_{x,2,t}^{(k)} - \nabla_1 g^{(k)}(x_t^{(k)}, y_t^{(k)})\|^2] + \frac{9\gamma_x^2\eta_x^4}{1-\lambda}\frac{1}{\rho^2}\sum_{k=1}^{K}\mathbb{E}[\|m_{x,3,t}^{(k)} - \nabla_1 g^{(k)}(x_t^{(k)}, z_t^{(k)})\|^2] \,, \tag{59}
\end{aligned}$$

$$\begin{aligned}
\mathbb{E}[\|U_{y,t+1} - \bar{U}_{y,t+1}\|_F^2] &\le \lambda \mathbb{E}[\|U_{y,t} - \bar{U}_{y,t}\|_F^2] + \frac{6}{1-\lambda}(L_f^2 + \frac{L_g^2}{\rho^2})\mathbb{E}[\|X_{t+1} - X_t\|_F^2] \\
&+ \frac{6}{1-\lambda}(L_f^2 + \frac{L_g^2}{\rho^2})\mathbb{E}[\|Y_{t+1} - Y_t\|_F^2] + \frac{6K\gamma_y^2\eta_y^4\sigma^2}{1-\lambda}(1 + \frac{1}{\rho^2}) \\
&+ \frac{6\gamma_y^2\eta_y^4}{1-\lambda}\sum_{k=1}^{K}\mathbb{E}[\|m_{y,1,t}^{(k)} - \nabla_1 f^{(k)}(x_t^{(k)}, y_t^{(k)})\|^2] + \frac{6\gamma_y^2\eta_y^4}{1-\lambda}\frac{1}{\rho^2}\sum_{k=1}^{K}\mathbb{E}[\|m_{y,2,t}^{(k)} - \nabla_2 g^{(k)}(x_t^{(k)}, y_t^{(k)})\|^2] \,, \tag{60}
\end{aligned}$$

$$\begin{aligned}
\mathbb{E}[\|U_{z,t+1} - \bar{U}_{z,t+1}\|_F^2] &\le \lambda \mathbb{E}[\|U_{z,t} - \bar{U}_{z,t}\|_F^2] + \frac{3}{1-\lambda}\frac{L_g^2}{\rho^2}\mathbb{E}[\|X_{t+1} - X_t\|_F^2] \\
&+ \frac{3}{1-\lambda}\frac{L_g^2}{\rho^2}\mathbb{E}[\|Z_{t+1} - Z_t\|_F^2] + \frac{3K\gamma_z^2\eta_z^4\sigma^2}{1-\lambda}\frac{1}{\rho^2} + \frac{3\gamma_z^2\eta_z^4}{1-\lambda}\frac{1}{\rho^2}\sum_{k=1}^{K}\mathbb{E}[\|m_{z,1,t}^{(k)} - \nabla_2 g^{(k)}(x_t^{(k)}, z_t^{(k)})\|^2] \,. \tag{61}
\end{aligned}$$

*Proof.* For the first inequality, we can prove it as follows:

$$\mathbb{E}[\|U_{x,t+1} - \bar{U}_{x,t+1}\|_F^2]$$

$$= \mathbb{E}[\|U_{x,t}W - M_{x,t} + M_{x,t+1} - (\bar{U}_{x,t} - \bar{M}_{x,t} + \bar{M}_{x,t+1})\|_F^2]$$

$$\leq \frac{1}{\lambda}\mathbb{E}[\|U_{x,t}W - \bar{U}_{x,t}\|_F^2] + \frac{1}{1-\lambda}\mathbb{E}[\|M_{x,t+1} - M_{x,t} - \bar{M}_{x,t+1} + \bar{M}_{x,t}\|_F^2]$$

$$\leq \lambda\mathbb{E}[\|U_{x,t} - \bar{U}_{x,t}\|_F^2] + \frac{1}{1-\lambda}\mathbb{E}[\|M_{x,t+1} - M_{x,t} - \bar{M}_{x,t+1} + \bar{M}_{x,t}\|_F^2]$$

$$\leq \lambda\mathbb{E}[\|U_{x,t} - \bar{U}_{x,t}\|_F^2] + \frac{1}{1-\lambda}\mathbb{E}[\|M_{x,t+1} - M_{x,t}\|_F^2]$$

$$\leq \lambda\mathbb{E}[\|U_{x,t} - \bar{U}_{x,t}\|_F^2] + \frac{3}{1-\lambda}\mathbb{E}[\|M_{x,1,t+1} - M_{x,1,t}\|_F^2]$$

$$+ \frac{3}{1-\lambda}\frac{1}{\rho^2}\mathbb{E}[\|M_{x,2,t+1} - M_{x,2,t}\|_F^2] + \frac{3}{1-\lambda}\frac{1}{\rho^2}\mathbb{E}[\|M_{x,3,t+1} - M_{x,3,t}\|_F^2]$$

$$\leq \lambda\mathbb{E}[\|U_{x,t} - \bar{U}_{x,t}\|_F^2]$$

$$+ \frac{3}{1-\lambda}3L_f^2\mathbb{E}[\|X_{t+1} - X_t\|_F^2] + \frac{3}{1-\lambda}3L_f^2\mathbb{E}[\|Y_{t+1} - Y_t\|_F^2] + 3K\gamma_x^2\eta_x^4\sigma^2\frac{3}{1-\lambda}$$

$$+ 3\gamma_x^2\eta_x^4\frac{3}{1-\lambda}\sum_{k=1}^{K}\mathbb{E}[\|m_{x,1,t}^{(k)} - \nabla_1 f^{(k)}(x_t^{(k)}, y_t^{(k)})\|^2]$$

$$+ \frac{3}{1-\lambda}\frac{1}{\rho^2}3L_g^2\mathbb{E}[\|X_{t+1} - X_t\|_F^2] + 3L_g^2\frac{3}{1-\lambda}\frac{1}{\rho^2}\mathbb{E}[\|Y_{t+1} - Y_t\|_F^2] + 3K\gamma_x^2\eta_x^4\sigma^2\frac{3}{1-\lambda}\frac{1}{\rho^2}$$

$$+ 3\gamma_x^2\eta_x^4\frac{3}{1-\lambda}\frac{1}{\rho^2}\sum_{k=1}^{K}\mathbb{E}[\|m_{x,2,t}^{(k)} - \nabla_1 g^{(k)}(x_t^{(k)}, y_t^{(k)})\|^2]$$

$$+ \frac{3}{1-\lambda}\frac{1}{\rho^2}3L_g^2\mathbb{E}[\|X_{t+1} - X_t\|_F^2] + 3L_g^2\frac{3}{1-\lambda}\frac{1}{\rho^2}\mathbb{E}[\|Z_{t+1} - Z_t\|_F^2] + 3K\gamma_x^2\eta_x^4\sigma^2\frac{3}{1-\lambda}\frac{1}{\rho^2}$$

$$+ 3\gamma_x^2\eta_x^4\frac{3}{1-\lambda}\frac{1}{\rho^2}\sum_{k=1}^{K}\mathbb{E}[\|m_{x,3,t}^{(k)} - \nabla_1 g^{(k)}(x_t^{(k)}, z_t^{(k)})\|^2]$$

$$\leq \lambda\mathbb{E}[\|U_{x,t} - \bar{U}_{x,t}\|_F^2] + \frac{9}{1-\lambda}(L_f^2 + 2\frac{L_g^2}{\rho^2})\mathbb{E}[\|X_{t+1} - X_t\|_F^2]$$

$$+ \frac{9}{1-\lambda}(L_f^2 + \frac{L_g^2}{\rho^2})\mathbb{E}[\|Y_{t+1} - Y_t\|_F^2] + \frac{9}{1-\lambda}\frac{L_g^2}{\rho^2}\mathbb{E}[\|Z_{t+1} - Z_t\|_F^2]$$

$$+ \frac{9\gamma_x^2\eta_x^4}{1-\lambda}\sum_{k=1}^{K}\mathbb{E}[\|m_{x,1,t}^{(k)} - \nabla_1 f^{(k)}(x_t^{(k)}, y_t^{(k)})\|^2] + \frac{9\gamma_x^2\eta_x^4}{1-\lambda}\frac{1}{\rho^2}\sum_{k=1}^{K}\mathbb{E}[\|m_{x,2,t}^{(k)} - \nabla_1 g^{(k)}(x_t^{(k)}, y_t^{(k)})\|^2]$$

$$+ \frac{9\gamma_x^2\eta_x^4}{1-\lambda}\frac{1}{\rho^2}\sum_{k=1}^{K}\mathbb{E}[\|m_{x,3,t}^{(k)} - \nabla_1 g^{(k)}(x_t^{(k)}, z_t^{(k)})\|^2] + \frac{9K\gamma_x^2\eta_x^4\sigma^2}{1-\lambda}(1 + \frac{2}{\rho^2}), \tag{62}$$

where the second step follows from the fact $\|a + b\|^2 \leq (1+c)\|a\|^2 + (1+1/c)\|b\|^2$ with $c = \frac{1}{\lambda} - 1$, the second to last step follows from Lemma A.12.

Similarly, for the second inequality, we can prove it as follows:

$$\mathbb{E}[\|U_{y,t+1} - \bar{U}_{y,t+1}\|_F^2]$$

$$\leq \lambda\mathbb{E}[\|U_{y,t} - \bar{U}_{y,t}\|_F^2] + \frac{1}{1-\lambda}\mathbb{E}[\|M_{y,t+1} - M_{y,t}\|_F^2]$$

$$\leq \lambda\mathbb{E}[\|U_{y,t} - \bar{U}_{y,t}\|_F^2] + \frac{2}{1-\lambda}\mathbb{E}[\|M_{y,1,t+1} - M_{y,1,t}\|_F^2] + \frac{2}{1-\lambda}\frac{1}{\rho^2}\mathbb{E}[\|M_{y,2,t+1} - M_{y,2,t}\|_F^2]$$

$$\leq \lambda\mathbb{E}[\|U_{y,t} - \bar{U}_{y,t}\|_F^2] + \frac{2}{1-\lambda}3L_f^2\mathbb{E}[\|X_{t+1} - X_t\|_F^2] + 3L_f^2\frac{2}{1-\lambda}\mathbb{E}[\|Y_{t+1} - Y_t\|_F^2] + 3K\gamma_y^2\eta_y^4\sigma^2\frac{2}{1-\lambda}$$

$$+ 3\gamma_y^2 \eta_y^4 \frac{2}{1-\lambda} \sum_{k=1}^{K} \mathbb{E}[\|m_{y,1,t}^{(k)} - \nabla_1 f^{(k)}(x_t^{(k)}, y_t^{(k)})\|^2]$$

$$+ \frac{2}{1-\lambda} \frac{1}{\rho^2} 3L_g^2 \mathbb{E}[\|X_{t+1} - X_t\|_F^2] + \frac{2}{1-\lambda} \frac{1}{\rho^2} 3L_g^2 \mathbb{E}[\|Y_{t+1} - Y_t\|_F^2] + 3K\gamma_y^2 \eta_y^4 \sigma^2 \frac{2}{1-\lambda} \frac{1}{\rho^2}$$

$$+ 3\gamma_y^2 \eta_y^4 \frac{2}{1-\lambda} \frac{1}{\rho^2} \sum_{k=1}^{K} \mathbb{E}[\|m_{y,2,t}^{(k)} - \nabla_2 g^{(k)}(x_t^{(k)}, y_t^{(k)})\|^2]$$

$$\leq \lambda \mathbb{E}[\|U_{y,t} - \bar{U}_{y,t}\|_F^2] + \frac{6}{1-\lambda}(L_f^2 + \frac{L_g^2}{\rho^2})\mathbb{E}[\|X_{t+1} - X_t\|_F^2] + \frac{6}{1-\lambda}(L_f^2 + \frac{L_g^2}{\rho^2})\mathbb{E}[\|Y_{t+1} - Y_t\|_F^2]$$

$$+ \frac{6\gamma_y^2 \eta_y^4}{1-\lambda} \sum_{k=1}^{K} \mathbb{E}[\|m_{y,1,t}^{(k)} - \nabla_1 f^{(k)}(x_t^{(k)}, y_t^{(k)})\|^2]$$

$$+ \frac{6\gamma_y^2 \eta_y^4}{1-\lambda} \frac{1}{\rho^2} \sum_{k=1}^{K} \mathbb{E}[\|m_{y,2,t}^{(k)} - \nabla_2 g^{(k)}(x_t^{(k)}, y_t^{(k)})\|^2] + \frac{6K\gamma_y^2 \eta_y^4 \sigma^2}{1-\lambda}(1 + \frac{1}{\rho^2}). \tag{63}$$

Moreover, for the third inequality, we can prove it as follows:

$$\mathbb{E}[\|U_{z,t+1} - \bar{U}_{z,t+1}\|_F^2]$$

$$\leq \lambda \mathbb{E}[\|U_{z,t} - \bar{U}_{z,t}\|_F^2] + \frac{1}{1-\lambda} \mathbb{E}[\|M_{z,t+1} - M_{z,t}\|_F^2]$$

$$= \lambda \mathbb{E}[\|U_{z,t} - \bar{U}_{z,t}\|_F^2] + \frac{1}{1-\lambda} \frac{1}{\rho^2} \mathbb{E}[\|M_{z,1,t+1} - M_{z,1,t}\|_F^2]$$

$$\leq \lambda \mathbb{E}[\|U_{z,t} - \bar{U}_{z,t}\|_F^2] + \frac{3}{1-\lambda} \frac{L_g^2}{\rho^2} \mathbb{E}[\|X_{t+1} - X_t\|_F^2] + \frac{3}{1-\lambda} \frac{L_g^2}{\rho^2} \mathbb{E}[\|Z_{t+1} - Z_t\|_F^2]$$

$$+ \frac{3\gamma_z^2 \eta_z^4}{1-\lambda} \frac{1}{\rho^2} \sum_{k=1}^{K} \mathbb{E}[\|m_{z,1,t}^{(k)} - \nabla_2 g^{(k)}(x_t^{(k)}, z_t^{(k)})\|^2] + \frac{3K\gamma_z^2 \eta_z^4 \sigma^2}{1-\lambda} \frac{1}{\rho^2}. \tag{64}$$

$\square$

## A.7. Auxiliary Lemma

**Lemma A.12.** *Suppose Assumptions 4.1-4.5 hold, we have that*

$$\mathbb{E}[\|M_{x,1,t+1} - M_{x,1,t}\|_F^2] \leq 3L_f^2 \mathbb{E}[\|X_{t+1} - X_t\|_F^2] + 3L_f^2 \mathbb{E}[\|Y_{t+1} - Y_t\|_F^2] + 3K\gamma_x^2 \eta_x^4 \sigma^2$$

$$+ 3\gamma_x^2 \eta_x^4 \sum_{k=1}^{K} \mathbb{E}[\|m_{x,1,t}^{(k)} - \nabla_1 f^{(k)}(x_t^{(k)}, y_t^{(k)})\|^2], \tag{65}$$

$$\mathbb{E}[\|M_{x,2,t+1} - M_{x,2,t}\|_F^2] \leq 3L_g^2 \mathbb{E}[\|X_{t+1} - X_t\|_F^2] + 3L_g^2 \mathbb{E}[\|Y_{t+1} - Y_t\|_F^2] + 3K\gamma_x^2 \eta_x^4 \sigma^2$$

$$+ 3\gamma_x^2 \eta_x^4 \sum_{k=1}^{K} \mathbb{E}[\|m_{x,2,t}^{(k)} - \nabla_1 g^{(k)}(x_t^{(k)}, y_t^{(k)})\|^2], \tag{66}$$

$$\mathbb{E}[\|M_{x,3,t+1} - M_{x,3,t}\|_F^2] \leq 3L_g^2 \mathbb{E}[\|X_{t+1} - X_t\|_F^2] + 3L_g^2 \mathbb{E}[\|Z_{t+1} - Z_t\|_F^2] + 3K\gamma_x^2 \eta_x^4 \sigma^2$$

$$+ 3\gamma_x^2 \eta_x^4 \sum_{k=1}^{K} \mathbb{E}[\|m_{x,3,t}^{(k)} - \nabla_1 g^{(k)}(x_t^{(k)}, z_t^{(k)})\|^2], \tag{67}$$

$$\mathbb{E}[\|M_{y,1,t+1} - M_{y,1,t}\|_F^2] \leq 3L_f^2 \mathbb{E}[\|X_{t+1} - X_t\|_F^2] + 3L_f^2 \mathbb{E}[\|Y_{t+1} - Y_t\|_F^2] + 3K\gamma_y^2 \eta_y^4 \sigma^2$$

$$+ 3\gamma_y^2\eta_y^4 \sum_{k=1}^K \mathbb{E}[\|m_{y,1,t}^{(k)} - \nabla_1 f^{(k)}(x_t^{(k)}, y_t^{(k)})\|^2] , \tag{68}$$

$$\mathbb{E}[\|M_{y,2,t+1} - M_{y,2,t}\|_F^2] \leq 3L_g^2 \mathbb{E}[\|X_{t+1} - X_t\|_F^2] + 3L_g^2 \mathbb{E}[\|Y_{t+1} - Y_t\|_F^2] + 3K\gamma_y^2\eta_y^4\sigma^2$$
$$+ 3\gamma_y^2\eta_y^4 \sum_{k=1}^K \mathbb{E}[\|m_{y,2,t}^{(k)} - \nabla_2 g^{(k)}(x_t^{(k)}, y_t^{(k)})\|^2] , \tag{69}$$

$$\mathbb{E}[\|M_{z,1,t+1} - M_{z,1,t}\|_F^2] \leq 3L_g^2 \mathbb{E}[\|X_{t+1} - X_t\|_F^2] + 3L_g^2 \mathbb{E}[\|Z_{t+1} - Z_t\|_F^2] + 3K\gamma_z^2\eta_z^4\sigma^2$$
$$+ 3\gamma_z^2\eta_z^4 \sum_{k=1}^K \mathbb{E}[\|m_{z,1,t}^{(k)} - \nabla_2 g^{(k)}(x_t^{(k)}, z_t^{(k)})\|^2] . \tag{70}$$

*Proof.*

$$\mathbb{E}[\|M_{x,1,t+1} - M_{x,1,t}\|_F^2]$$

$$\leq \sum_{k=1}^K \mathbb{E}[\|m_{x,1,t+1}^{(k)} - m_{x,1,t}^{(k)}\|^2]$$

$$\leq \sum_{k=1}^K \mathbb{E}[\|(1 - \gamma_x\eta^2)(m_{x,1,t}^{(k)} - \nabla_1 f^{(k)}(x_t^{(k)}, y_t^{(k)}; \xi_{t+1}^{(k)})) + \nabla_1 f^{(k)}(x_{t+1}^{(k)}, y_{t+1}^{(k)}; \xi_{t+1}^{(k)}) - m_{x,1,t}^{(k)}\|^2]$$

$$\leq 3\sum_{k=1}^K \mathbb{E}[\|\nabla_1 f^{(k)}(x_{t+1}^{(k)}, y_{t+1}^{(k)}; \xi_{t+1}^{(k)}) - \nabla_1 f^{(k)}(x_t^{(k)}, y_t^{(k)}; \xi_{t+1}^{(k)})\|^2]$$

$$+ 3\gamma_x^2\eta^4 \sum_{k=1}^K \mathbb{E}[\|m_{x,1,t}^{(k)} - \nabla_1 f^{(k)}(x_t^{(k)}, y_t^{(k)})\|^2] + 3\gamma_x^2\eta^4 \sum_{k=1}^K \mathbb{E}[\|\nabla_1 f^{(k)}(x_t^{(k)}, y_t^{(k)}) - \nabla_1 f^{(k)}(x_t^{(k)}, y_t^{(k)}; \xi_{t+1}^{(k)})\|^2]$$

$$\leq 3L_f^2 \mathbb{E}[\|X_{t+1} - X_t\|_F^2] + 3L_f^2 \mathbb{E}[\|Y_{t+1} - Y_t\|_F^2] + 3K\gamma_x^2\eta^4\sigma^2 + 3\gamma_x^2\eta^4 \sum_{k=1}^K \mathbb{E}[\|m_{x,1,t}^{(k)} - \nabla_1 f^{(k)}(x_t^{(k)}, y_t^{(k)})\|^2] , \tag{71}$$

where the last step follows from Assumption 4.1 and Assumption 4.4.

The other inequalities can be proved in the same way.

$\square$

**Lemma A.13.** *Suppose Assumptions 4.1-4.5 hold, we have that*

$$\|X_{t+1} - X_t\|_F^2 \leq 12\|X_t - \bar{X}_t\|_F^2 + 3\eta_x^2\|U_{x,t} - \bar{U}_{x,t}\|_F^2 + 3\eta_x^2 K\|\bar{m}_{x,t}\|^2 , \tag{72}$$

$$\|Y_{t+1} - Y_t\|_F^2 \leq 12\|Y_t - \bar{Y}_t\|_F^2 + 3\eta_y^2\|U_{y,t} - \bar{U}_{y,t}\|_F^2 + 3\eta_y^2 K\|\bar{m}_{y,t}\|^2 , \tag{73}$$

$$\|Z_{t+1} - Z_t\|_F^2 \leq 12\|Z_t - \bar{Z}_t\|_F^2 + 3\eta_z^2\|U_{z,t} - \bar{U}_{z,t}\|_F^2 + 3\eta_z^2 K\|\bar{m}_{z,t}\|^2 . \tag{74}$$

*Proof.*

$$\|X_{t+1} - X_t\|_F^2$$
$$= \|X_t W - \eta_x U_{x,t} - X_t\|_F^2$$
$$= \|X_t W - X_t - \eta_x U_{x,t} + \eta_x \bar{U}_{x,t} - \eta_x \bar{U}_{x,t}\|_F^2$$

$$\leq 3\|X_t W - X_t\|_F^2 + 3\eta_x^2\|U_{x,t} - \bar{U}_{x,t}\|_F^2 + 3\eta_x^2\|\bar{U}_{x,t}\|_F^2$$
$$= 3\|(X_t - \bar{X}_t)(W - I)\|_F^2 + 3\eta_x^2\|U_{x,t} - \bar{U}_{x,t}\|_F^2 + 3\eta_x^2\|\bar{U}_{x,t}\|_F^2$$
$$\leq 3\|X_t - \bar{X}_t\|_F^2\|W - I\|_2^2 + 3\eta_x^2\|U_{x,t} - \bar{U}_{x,t}\|_F^2 + 3\eta_x^2\|\bar{U}_{x,t}\|_F^2$$
$$\leq 12\|X_t - \bar{X}_t\|_F^2 + 3\eta_x^2\|U_{x,t} - \bar{U}_{x,t}\|_F^2 + 3\eta_x^2\|\bar{M}_{x,t}\|_F^2 . \tag{75}$$

The other inequalities can be proved in the same way. $\qquad\square$

### A.8. Proof of Theorem 4.6

*Proof.* Given the potential function $P_t$ in Eq. (12), by plugging Lemmas A.5, A.6, A.7, A.8, A.9, A.10, A.11, and A.13 into $P_{t+1} - P_t$, we have

$$P_{t+1} - P_t$$
$$\leq -\frac{\eta_x}{2}\mathbb{E}[\|\nabla\mathcal{L}(\bar{x}_t)\|^2] + \frac{3\eta_x}{2}\mathbb{E}[\|\nabla\mathcal{L}(\bar{x}_t) - \nabla\mathcal{L}_\rho(\bar{x}_t)\|^2]$$
$$+ c_5\frac{9\gamma_x^2\eta_x^4\sigma^2}{1-\lambda}(1+\frac{2}{\rho^2}) + c_6\frac{6\gamma_y^2\eta_y^4\sigma^2}{1-\lambda}(1+\frac{1}{\rho^2}) + c_7\frac{3\gamma_z^2\eta_z^4\sigma^2}{1-\lambda}\frac{1}{\rho^2}$$
$$+ 2c_8\gamma_x^2\eta_x^4\sigma^2\frac{1}{K} + 2c_9\frac{1}{\rho^2}\gamma_x^2\eta_x^4\sigma^2\frac{1}{K} + 2c_{10}\frac{1}{\rho^2}\gamma_x^2\eta_x^4\sigma^2\frac{1}{K} + 2c_{11}\gamma_y^2\eta_y^4\sigma^2\frac{1}{K} + 2c_{12}\frac{1}{\rho^2}\gamma_y^2\eta_y^4\sigma^2\frac{1}{K} + 2c_{13}\frac{1}{\rho^2}\gamma_z^2\eta_z^4\sigma^2\frac{1}{K}$$
$$+ 2c_{14}\gamma_x^2\eta_x^4\sigma^2 + 2c_{15}\frac{1}{\rho^2}\gamma_x^2\eta_x^4\sigma^2 + 2c_{16}\frac{1}{\rho^2}\gamma_x^2\eta_x^4\sigma^2 + 2c_{17}\gamma_y^2\eta_y^4\sigma^2 + 2c_{18}\frac{1}{\rho^2}\gamma_y^2\eta_y^4\sigma^2 + 2c_{19}\frac{1}{\rho^2}\gamma_z^2\eta_z^4\sigma^2$$
$$+ \left[\frac{9\eta_x}{2\mu^2}(L_f^2 + \frac{L_g^2}{\rho^2}) - \frac{c_0\eta_x}{\eta_y}\frac{1}{\rho^2}\frac{\eta_y}{8}\right]\mathbb{E}[\|\nabla_2 h_\rho(\bar{x}_t, \bar{y}_t)\|^2] + \left[\frac{9\eta_x}{2\mu^2}\frac{L_g^2}{\rho^2} - \frac{c_1\eta_x}{\eta_z}\frac{1}{\rho^2}\frac{\eta_z}{8}\right]\mathbb{E}[\|\nabla_2 g(\bar{x}_t, \bar{z}_t)\|^2]$$
$$+ \left[3\eta_y^2 C_2 - \frac{c_0\eta_x}{\eta_y}\frac{\eta_y}{4}\right]\mathbb{E}[\|\bar{m}_{y,t}\|^2] + \left[3\eta_z^2 C_3 - \frac{c_1\eta_x}{\eta_z}\frac{\eta_z}{4}\right]\mathbb{E}[\|\bar{m}_{z,t}\|^2]$$
$$+ \left[\frac{c_0\eta_x}{\eta_y}\left(\frac{\eta_x}{2} + \frac{1}{\rho}\frac{\eta_x^2 L_{h_\rho}}{2} + \frac{1}{\rho}\frac{3\eta_x^2 L_{h_\rho}}{2} + \frac{1}{\rho}\frac{\eta_x^2 L_{\hat{h}_\rho}}{2}\right) + \frac{c_1\eta_x}{\eta_z}\left(\frac{\eta_x}{2} + 2\eta_x^2 L_g\frac{1}{\rho} + \frac{\eta_x^2 L_{\hat{g}}}{2}\frac{1}{\rho}\right) + 3\eta_x^2 C_1 - \frac{\eta_x}{4}\right]\mathbb{E}[\|\bar{m}_{x,t}\|^2]$$
$$+ T_{\text{consensus-error-variable}} + T_{\text{consensus-error-gradient}} + T_{\text{gradient-estimation-error-global}} + T_{\text{gradient-estimation-error-local}} . \tag{76}$$

The last four terms, $T_{\text{consensus-error-variable}}$, $T_{\text{consensus-error-gradient}}$, $T_{\text{gradient-estimation-error-global}}$, and $T_{\text{gradient-estimation-error-local}}$, are defined as follows:

$$T_{\text{consensus-error-variable}} = \left[9\eta_x(L_f^2 + 2\frac{1}{\rho^2}L_g^2) + \frac{c_0\eta_x}{\eta_y}2\eta_y\frac{L_{h_\rho}^2}{\rho^2} + \frac{c_1\eta_x}{\eta_z}2\eta_z\frac{L_g^2}{\rho^2} + 12C_1 - (1-\lambda)c_2\right]\frac{1}{K}\mathbb{E}[\|X_t - \bar{X}_t\|_F^2]$$
$$+ \left[9\eta_x(L_f^2 + \frac{1}{\rho^2}L_g^2) + \frac{c_0\eta_x}{\eta_y}2\eta_y\frac{L_{h_\rho}^2}{\rho^2} + 12C_2 - (1-\lambda)c_3\right]\frac{1}{K}\mathbb{E}[\|Y_t - \bar{Y}_t\|_F^2]$$
$$+ \left[9\eta_x\frac{1}{\rho^2}L_g^2 + \frac{c_1\eta_x}{\eta_z}2\eta_z\frac{L_g^2}{\rho^2} + 12C_3 - (1-\lambda)c_4\right]\frac{1}{K}\mathbb{E}[\|Z_t - \bar{Z}_t\|_F^2] , \tag{77}$$

$$T_{\text{consensus-error-gradient}} = \left[c_3\frac{\eta_y^2}{1-\lambda} + 3\eta_y^2 C_2 - (1-\lambda)c_6\right]\frac{1}{K}\mathbb{E}[\|U_{y,t} - \bar{U}_{y,t}\|_F^2]$$
$$+ \left[c_4\frac{\eta_z^2}{1-\lambda} + 3\eta_z^2 C_3 - (1-\lambda)c_7\right]\frac{1}{K}\mathbb{E}[\|U_{z,t} - \bar{U}_{z,t}\|_F^2]$$
$$+ \left[c_2\frac{\eta_x^2}{1-\lambda} + 3\eta_x^2 C_1 - (1-\lambda)c_5\right]\frac{1}{K}\mathbb{E}[\|U_{x,t} - \bar{U}_{x,t}\|_F^2] , \tag{78}$$

$$T_{\text{gradient-estimation-error-global}} = \left[9\eta_x - c_8\gamma_x\eta_x^2\right]\mathbb{E}[\|\frac{1}{K}\sum_{k=1}^K m_{x,1,t}^{(k)} - \frac{1}{K}\sum_{k=1}^K \nabla_1 f^{(k)}(x_t^{(k)}, y_t^{(k)})\|^2]$$

$$+ \left[ 9\eta_x \frac{1}{\rho^2} - c_9 \frac{1}{\rho^2} \gamma_x \eta_x^2 \right] \mathbb{E}[\| \frac{1}{K} \sum_{k=1}^{K} m_{x,2,t}^{(k)} - \frac{1}{K} \sum_{k=1}^{K} \nabla_1 g^{(k)}(x_t^{(k)}, y_t^{(k)}) \|^2]$$

$$+ \left[ 9\eta_x \frac{1}{\rho^2} - c_{10} \frac{1}{\rho^2} \gamma_x \eta_x^2 \right] \mathbb{E}[\| \frac{1}{K} \sum_{k=1}^{K} m_{x,3,t}^{(k)} - \frac{1}{K} \sum_{k=1}^{K} \nabla_1 g^{(k)}(x_t^{(k)}, z_t^{(k)}) \|^2]$$

$$+ \left[ \frac{c_0 \eta_x}{\eta_y} 4\eta_y - c_{11} \gamma_y \eta_y^2 \right] \mathbb{E}[\| \frac{1}{K} \sum_{k=1}^{K} m_{y,1,t}^{(k)} - \frac{1}{K} \sum_{k=1}^{K} \nabla_2 f^{(k)}(x_t^{(k)}, y_t^{(k)}) \|^2]$$

$$+ \left[ \frac{c_0 \eta_x}{\eta_y} 4\eta_y \frac{1}{\rho^2} - c_{12} \frac{1}{\rho^2} \gamma_y \eta_y^2 \right] \mathbb{E}[\| \frac{1}{K} \sum_{k=1}^{K} m_{y,2,t}^{(k)} - \frac{1}{K} \sum_{k=1}^{K} \nabla_2 g^{(k)}(x_t^{(k)}, y_t^{(k)}) \|^2]$$

$$+ \left[ \frac{c_1 \eta_x}{\eta_z} \frac{2\eta_z}{\rho^2} - c_{13} \frac{1}{\rho^2} \gamma_z \eta_z^2 \right] \mathbb{E}[\| \frac{1}{K} \sum_{k=1}^{K} m_{z,1,t}^{(k)} - \frac{1}{K} \sum_{k=1}^{K} \nabla_2 g^{(k)}(x_t^{(k)}, z_t^{(k)}) \|^2] , \tag{79}$$

$$T_{\text{gradient-estimation-error-local}} = \left[ c_5 \frac{9\gamma_x^2 \eta_x^4}{1-\lambda} - c_{14} \gamma_x \eta_x^2 \right] \frac{1}{K} \sum_{k=1}^{K} \mathbb{E}[\| m_{x,1,t}^{(k)} - \nabla_1 f^{(k)}(x_t^{(k)}, y_t^{(k)}) \|^2]$$

$$+ \left[ c_5 \frac{9\gamma_x^2 \eta_x^4}{1-\lambda} \frac{1}{\rho^2} - c_{15} \frac{1}{\rho^2} \gamma_x \eta_x^2 \right] \frac{1}{K} \sum_{k=1}^{K} \mathbb{E}[\| m_{x,2,t}^{(k)} - \nabla_1 g^{(k)}(x_t^{(k)}, y_t^{(k)}) \|^2]$$

$$+ \left[ c_5 \frac{9\gamma_x^2 \eta_x^4}{1-\lambda} \frac{1}{\rho^2} - c_{16} \frac{1}{\rho^2} \gamma_x \eta_x^2 \right] \frac{1}{K} \sum_{k=1}^{K} \mathbb{E}[\| m_{x,3,t}^{(k)} - \nabla_1 g^{(k)}(x_t^{(k)}, z_t^{(k)}) \|^2]$$

$$+ \left[ c_6 \frac{6\gamma_y^2 \eta_y^4}{1-\lambda} - c_{17} \gamma_y \eta_y^2 \right] \frac{1}{K} \sum_{k=1}^{K} \mathbb{E}[\| m_{y,1,t}^{(k)} - \nabla_2 f^{(k)}(x_t^{(k)}, y_t^{(k)}) \|^2]$$

$$+ \left[ c_6 \frac{6\gamma_y^2 \eta_y^4}{1-\lambda} \frac{1}{\rho^2} - c_{18} \frac{1}{\rho^2} \gamma_y \eta_y^2 \right] \frac{1}{K} \sum_{k=1}^{K} \mathbb{E}[\| m_{y,2,t}^{(k)} - \nabla_2 g^{(k)}(x_t^{(k)}, y_t^{(k)}) \|^2]$$

$$+ \left[ c_7 \frac{3\gamma_z^2 \eta_z^4}{1-\lambda} \frac{1}{\rho^2} - c_{19} \frac{1}{\rho^2} \gamma_z \eta_z^2 \right] \frac{1}{K} \sum_{k=1}^{K} \mathbb{E}[\| m_{z,1,t}^{(k)} - \nabla_2 g^{(k)}(x_t^{(k)}, z_t^{(k)}) \|^2] , \tag{80}$$

where

$$C_1 = c_5 \frac{9}{1-\lambda}(L_f^2 + 2\frac{L_g^2}{\rho^2}) + c_6 \frac{6}{1-\lambda}(L_f^2 + \frac{L_g^2}{\rho^2}) + c_7 \frac{3}{1-\lambda}\frac{L_g^2}{\rho^2}$$
$$+ 2c_8 L_f^2 \frac{1}{K} + 2c_9 \frac{1}{\rho^2} L_g^2 \frac{1}{K} + 2c_{10} \frac{1}{\rho^2} L_g^2 \frac{1}{K} + 2c_{11} L_f^2 \frac{1}{K} + 2c_{12} \frac{1}{\rho^2} L_g^2 \frac{1}{K} + 2c_{13} \frac{1}{\rho^2} L_g^2 \frac{1}{K}$$
$$+ 2c_{14} L_f^2 + 2c_{15} \frac{1}{\rho^2} L_g^2 + 2c_{16} \frac{1}{\rho^2} L_g^2 + 2c_{17} L_f^2 + 2c_{18} \frac{1}{\rho^2} L_g^2 + 2c_{19} \frac{1}{\rho^2} L_g^2 ,$$

$$C_2 = c_5 \frac{9}{1-\lambda}(L_f^2 + \frac{L_g^2}{\rho^2}) + c_6 \frac{6}{1-\lambda}(L_f^2 + \frac{L_g^2}{\rho^2}) + 2c_8 L_f^2 \frac{1}{K} + 2c_9 \frac{1}{\rho^2} L_g^2 \frac{1}{K}$$
$$+ 2c_{11} L_f^2 \frac{1}{K} + 2c_{12} \frac{1}{\rho^2} L_g^2 \frac{1}{K} + 2c_{14} L_f^2 + 2c_{15} \frac{1}{\rho^2} L_g^2 + 2c_{17} L_f^2 + 2c_{18} \frac{1}{\rho^2} L_g^2 ,$$

$$C_3 = c_5 \frac{9}{1-\lambda}\frac{L_g^2}{\rho^2} + c_7 \frac{3}{1-\lambda}\frac{L_g^2}{\rho^2} + 2c_{10} \frac{1}{\rho^2} L_g^2 \frac{1}{K} + 2c_{13} \frac{1}{\rho^2} L_g^2 \frac{1}{K} + 2c_{16} \frac{1}{\rho^2} L_g^2 + 2c_{19} \frac{1}{\rho^2} L_g^2 . \tag{81}$$

In the following, we select the coefficient $\{c_i\}_{i=0}^{19}$ and the hyperparameters to eliminate all terms from the fifth line to the last line in Eq. (76).

By setting

$$c_0 = \frac{36(\rho^2 L_f^2 + L_g^2)}{\mu^2} , \quad c_1 = \frac{36L_g^2}{\mu^2}$$

$$c_5 = (1-\lambda)\eta_x , \quad c_6 = (1-\lambda)\eta_y , \quad c_7 = (1-\lambda)\eta_z ,$$

$$c_8 = \frac{9}{\gamma_x \eta_x} , \quad c_9 = \frac{9}{\gamma_x \eta_x} , \quad c_{10} = \frac{9}{\gamma_x \eta_x} ,$$

$$c_{11} = \frac{144(\rho^2 L_f^2 + L_g^2)}{\mu^2} \frac{\eta_x}{\gamma_y \eta_y^2} , \quad c_{12} = \frac{144(\rho^2 L_f^2 + L_g^2)}{\mu^2} \frac{\eta_x}{\gamma_y \eta_y^2} , \quad c_{13} = \frac{72 L_g^2}{\mu^2} \frac{\eta_x}{\gamma_z \eta_z^2} ,$$

$$c_{14} = 9\eta_x , \quad c_{15} = 9\eta_x , \quad c_{16} = 9\eta_x , \quad c_{17} = 9\eta_y , \quad c_{18} = 9\eta_y , \quad c_{19} = 9\eta_z,$$

$$\frac{c_0 \eta_x}{\eta_y} = \frac{1}{10} , \frac{c_1 \eta_x}{\eta_z} = \frac{1}{10} , \tag{82}$$

the global gradient estimation error $T_{\text{gradient-estimation-error-global}}$ and the local gradient estimation error $T_{\text{gradient-estimation-error-local}}$ can be removed, as the coefficient of all items in $T_{\text{gradient-estimation-error-global}}$ and $T_{\text{gradient-estimation-error-local}}$ are non-positive. Similarly, both $\mathbb{E}[\|\nabla_2 h_\rho(\bar{x}_t, \bar{y}_t)\|^2]$ and $\mathbb{E}[\|\nabla_2 g(\bar{x}_t, \bar{z}_t)\|^2]$ can be removed.

As a result, we have

$$P_{t+1} - P_t$$

$$\leq -\frac{\eta_x}{2} \mathbb{E}[\|\nabla \mathcal{L}(\bar{x}_t)\|^2] + \frac{3\eta_x}{2} \mathbb{E}[\|\nabla \mathcal{L}(\bar{x}_t) - \nabla \mathcal{L}_\rho(\bar{x}_t)\|^2]$$

$$+ c_5 \frac{9\gamma_x^2 \eta_x^4 \sigma^2}{1-\lambda}(1 + \frac{2}{\rho^2}) + c_6 \frac{6\gamma_y^2 \eta_y^4 \sigma^2}{1-\lambda}(1 + \frac{1}{\rho^2}) + c_7 \frac{3\gamma_z^2 \eta_z^4 \sigma^2}{1-\lambda} \frac{1}{\rho^2}$$

$$+ 2c_8 \gamma_x^2 \eta_x^4 \sigma^2 \frac{1}{K} + 2c_9 \frac{1}{\rho^2} \gamma_x^2 \eta_x^4 \sigma^2 \frac{1}{K} + 2c_{10} \frac{1}{\rho^2} \gamma_x^2 \eta_x^4 \sigma^2 \frac{1}{K} + 2c_{11} \gamma_y^2 \eta_y^4 \sigma^2 \frac{1}{K} + 2c_{12} \frac{1}{\rho^2} \gamma_y^2 \eta_y^4 \sigma^2 \frac{1}{K} + 2c_{13} \frac{1}{\rho^2} \gamma_z^2 \eta_z^4 \sigma^2 \frac{1}{K}$$

$$+ 2c_{14} \gamma_x^2 \eta_x^4 \sigma^2 + 2c_{15} \frac{1}{\rho^2} \gamma_x^2 \eta_x^4 \sigma^2 + 2c_{16} \frac{1}{\rho^2} \gamma_x^2 \eta_x^4 \sigma^2 + 2c_{17} \gamma_y^2 \eta_y^4 \sigma^2 + 2c_{18} \frac{1}{\rho^2} \gamma_y^2 \eta_y^4 \sigma^2 + 2c_{19} \frac{1}{\rho^2} \gamma_z^2 \eta_z^4 \sigma^2$$

$$+ \left[ 3\eta_y^2 C_2 - \frac{c_0 \eta_x}{\eta_y} \frac{\eta_y}{4} \right] \mathbb{E}[\|\bar{m}_{y,t}\|^2] + \left[ 3\eta_z^2 C_3 - \frac{c_1 \eta_x}{\eta_z} \frac{\eta_z}{4} \right] \mathbb{E}[\|\bar{m}_{z,t}\|^2]$$

$$+ \left[ \frac{c_0 \eta_x}{\eta_y} \left( \frac{\eta_x}{2} + \frac{1}{\rho} \frac{\eta_x^2 L_{h_\rho}}{2} + \frac{1}{\rho} \frac{3\eta_x^2 L_{h_\rho}}{2} + \frac{1}{\rho} \frac{\eta_x^2 L_{\hat{h}_\rho}}{2} \right) \right.$$

$$\left. + \frac{c_1 \eta_x}{\eta_z} \left( \frac{\eta_x}{2} + 2\eta_x^2 L_g \frac{1}{\rho} + \frac{\eta_x^2 L_{\hat{g}}}{2} \frac{1}{\rho} \right) + 3\eta_x^2 C_1 - \frac{\eta_x}{4} \right] \mathbb{E}[\|\bar{m}_{x,t}\|^2]$$

$$+ T_{\text{consensus-error-variable}} + T_{\text{consensus-error-gradient}} . \tag{83}$$

Moreover, given the coefficients in Eq. (82), we have

$$C_1 = c_5 \frac{9}{1-\lambda}(L_f^2 + 2\frac{L_g^2}{\rho^2}) + c_6 \frac{6}{1-\lambda}(L_f^2 + \frac{L_g^2}{\rho^2}) + c_7 \frac{3}{1-\lambda} \frac{L_g^2}{\rho^2}$$

$$+ 2c_8 L_f^2 \frac{1}{K} + 2c_9 \frac{1}{\rho^2} L_g^2 \frac{1}{K} + 2c_{10} \frac{1}{\rho^2} L_g^2 \frac{1}{K} + 2c_{11} L_f^2 \frac{1}{K} + 2c_{12} \frac{1}{\rho^2} L_g^2 \frac{1}{K} + 2c_{13} \frac{1}{\rho^2} L_g^2 \frac{1}{K}$$

$$+ 2c_{14} L_f^2 + 2c_{15} \frac{1}{\rho^2} L_g^2 + 2c_{16} \frac{1}{\rho^2} L_g^2 + 2c_{17} L_f^2 + 2c_{18} \frac{1}{\rho^2} L_g^2 + 2c_{19} \frac{1}{\rho^2} L_g^2$$

$$= 9\eta_x(L_f^2 + 2\frac{L_g^2}{\rho^2}) + 6\eta_y(L_f^2 + \frac{L_g^2}{\rho^2}) + 3\eta_z \frac{L_g^2}{\rho^2} + \frac{18 L_f^2}{\gamma_x \eta_x} \frac{1}{K} + \frac{18 L_g^2}{\gamma_x \eta_x} \frac{1}{\rho^2} \frac{1}{K} + \frac{18 L_g^2}{\gamma_x \eta_x} \frac{1}{\rho^2} \frac{1}{K}$$

$$+ \frac{288(\rho^2 L_f^2 + L_g^2)}{\mu^2} \frac{\eta_x L_f^2}{\gamma_y \eta_y^2} \frac{1}{K} + \frac{288(\rho^2 L_f^2 + L_g^2)}{\mu^2} \frac{\eta_x L_g^2}{\gamma_y \eta_y^2} \frac{1}{\rho^2} \frac{1}{K} + \frac{144 L_g^2}{\mu^2} \frac{\eta_x L_g^2}{\gamma_z \eta_z^2} \frac{1}{\rho^2} \frac{1}{K}$$

$$+ 18\eta_x L_f^2 + 36\eta_x L_g^2 \frac{1}{\rho^2} + 18\eta_y L_f^2 + 18\eta_y L_g^2 \frac{1}{\rho^2} + 18\eta_z L_g^2 \frac{1}{\rho^2}$$

$$\leq 27\eta_x(L_f^2 + 2\frac{L_g^2}{\rho^2}) + 27\eta_y(L_f^2 + \frac{L_g^2}{\rho^2}) + 27\eta_z \frac{L_g^2}{\rho^2}$$

$$+ \frac{1}{K}\frac{288}{\gamma_x \eta_x}(L_f^2 + 2\frac{L_g^2}{\rho^2}) + \frac{1}{K}\frac{288\eta_x}{\gamma_y \eta_y^2}\frac{(\rho^2 L_f^2 + L_g^2)}{\mu^2}(L_f^2 + \frac{L_g^2}{\rho^2}) + \frac{1}{K}\frac{288\eta_x}{\gamma_z \eta_z^2}\frac{L_g^2}{\mu^2}\frac{L_g^2}{\rho^2}$$

$$= 27\eta_x(L_f^2 + 2\frac{L_g^2}{\rho^2}) + 270c_0\eta_x(L_f^2 + \frac{L_g^2}{\rho^2}) + 270c_1\eta_x\frac{L_g^2}{\rho^2}$$

$$+ \frac{1}{K}\frac{288}{\gamma_x \eta_x}(L_f^2 + 2\frac{L_g^2}{\rho^2}) + \frac{1}{100c_0^2}\frac{1}{K}\frac{288}{\gamma_y \eta_x}\frac{(\rho^2 L_f^2 + L_g^2)}{\mu^2}(L_f^2 + \frac{L_g^2}{\rho^2}) + \frac{1}{100c_0^2}\frac{1}{K}\frac{288}{\gamma_z \eta_x}\frac{L_g^2}{\mu^2}\frac{L_g^2}{\rho^2}, \tag{84}$$

and

$$C_2 = c_5\frac{9}{1-\lambda}(L_f^2 + \frac{L_g^2}{\rho^2}) + c_6\frac{6}{1-\lambda}(L_f^2 + \frac{L_g^2}{\rho^2}) + 2c_8 L_f^2\frac{1}{K} + 2c_9\frac{1}{\rho^2}L_g^2\frac{1}{K} + 2c_{11}L_f^2\frac{1}{K} + 2c_{12}\frac{1}{\rho^2}L_g^2\frac{1}{K}$$

$$+ 2c_{14}L_f^2 + 2c_{15}\frac{1}{\rho^2}L_g^2 + 2c_{17}L_f^2 + 2c_{18}\frac{1}{\rho^2}L_g^2$$

$$= (1-\lambda)\eta_x\frac{9}{1-\lambda}(L_f^2 + \frac{L_g^2}{\rho^2}) + (1-\lambda)\eta_y\frac{6}{1-\lambda}(L_f^2 + \frac{L_g^2}{\rho^2}) + 2\frac{9}{\gamma_x \eta_x}L_f^2\frac{1}{K} + 2\frac{9}{\gamma_x \eta_x}\frac{1}{\rho^2}L_g^2\frac{1}{K} + 18\eta_y\frac{1}{\rho^2}L_g^2$$

$$+ 2\frac{144(\rho^2 L_f^2 + L_g^2)}{\mu^2}\frac{\eta_x}{\gamma_y \eta_y^2}L_f^2\frac{1}{K} + 2\frac{144(\rho^2 L_f^2 + L_g^2)}{\mu^2}\frac{\eta_x}{\gamma_y \eta_y^2}\frac{1}{\rho^2}L_g^2\frac{1}{K} + 18\eta_x L_f^2 + 18\eta_x\frac{1}{\rho^2}L_g^2 + 18\eta_y L_f^2$$

$$\leq 27\eta_x(L_f^2 + \frac{L_g^2}{\rho^2}) + 27\eta_y(L_f^2 + \frac{L_g^2}{\rho^2}) + \frac{288}{\gamma_x \eta_x}\frac{1}{K}(L_f^2 + \frac{L_g^2}{\rho^2}) + \frac{288(\rho^2 L_f^2 + L_g^2)}{\mu^2}\frac{\eta_x}{\gamma_y \eta_y^2}\frac{1}{K}(L_f^2 + \frac{L_g^2}{\rho^2})$$

$$\leq 27\frac{\eta_y}{10c_0}(L_f^2 + \frac{L_g^2}{\rho^2}) + 27\eta_y(L_f^2 + \frac{L_g^2}{\rho^2}) + \frac{2880c_0}{\gamma_x \eta_y}\frac{1}{K}(L_f^2 + \frac{L_g^2}{\rho^2}) + \frac{288(\rho^2 L_f^2 + L_g^2)}{\mu^2}\frac{1}{10c_0}\frac{1}{\gamma_y \eta_y}\frac{1}{K}(L_f^2 + \frac{L_g^2}{\rho^2}), \tag{85}$$

and

$$C_3 = c_5\frac{9}{1-\lambda}\frac{L_g^2}{\rho^2} + c_7\frac{3}{1-\lambda}\frac{L_g^2}{\rho^2} + 2c_{10}\frac{1}{\rho^2}L_g^2\frac{1}{K} + 2c_{13}\frac{1}{\rho^2}L_g^2\frac{1}{K} + 2c_{16}\frac{1}{\rho^2}L_g^2 + 2c_{19}\frac{1}{\rho^2}L_g^2$$

$$= (1-\lambda)\eta_x\frac{9}{1-\lambda}\frac{L_g^2}{\rho^2} + (1-\lambda)\eta_z\frac{3}{1-\lambda}\frac{L_g^2}{\rho^2} + 2\frac{9}{\gamma_x \eta_x}\frac{1}{\rho^2}L_g^2\frac{1}{K}$$

$$+ 2\frac{72L_g^2}{\mu^2}\frac{\eta_x}{\gamma_z \eta_z^2}\frac{1}{\rho^2}L_g^2\frac{1}{K} + 18\eta_x\frac{1}{\rho^2}L_g^2 + 18\eta_z\frac{1}{\rho^2}L_g^2$$

$$= 27\eta_x\frac{L_g^2}{\rho^2} + 27\eta_z\frac{L_g^2}{\rho^2} + \frac{18}{\gamma_x \eta_x}\frac{L_g^2}{\rho^2}\frac{1}{K} + \frac{144L_g^2}{\mu^2}\frac{\eta_x}{\gamma_z \eta_z^2}\frac{L_g^2}{\rho^2}\frac{1}{K}$$

$$\leq 27\frac{\eta_z}{10c_1}\frac{L_g^2}{\rho^2} + 27\eta_z\frac{L_g^2}{\rho^2} + \frac{2880c_1}{\gamma_x \eta_z}\frac{L_g^2}{\rho^2}\frac{1}{K} + \frac{288L_g^2}{\mu^2}\frac{1}{10c_1}\frac{1}{\gamma_z \eta_z}\frac{L_g^2}{\rho^2}\frac{1}{K}. \tag{86}$$

To eliminate the consensus error term $\mathbb{E}[\|X_t - \bar{X}_t\|_F^2]$, we enforce

$$9\eta_x(L_f^2 + 2\frac{1}{\rho^2}L_g^2) + \frac{c_0\eta_x}{\eta_y}2\eta_y\frac{L_{h_\rho}^2}{\rho^2} + \frac{c_1\eta_x}{\eta_z}2\eta_z\frac{L_g^2}{\rho^2} + 12C_1 - (1-\lambda)c_2$$

$$\leq 9\eta_x(L_f^2 + 2\frac{1}{\rho^2}L_g^2) + 2c_0\eta_x\frac{L_{h_\rho}^2}{\rho^2} + 2c_1\eta_x\frac{L_g^2}{\rho^2} - (1-\lambda)c_2$$

$$+ 12\left(27\eta_x(L_f^2 + 2\frac{L_g^2}{\rho^2}) + 270c_0\eta_x(L_f^2 + \frac{L_g^2}{\rho^2}) + 270c_1\eta_x\frac{L_g^2}{\rho^2}\right.$$

$$\left. + \frac{1}{K}\frac{288}{\gamma_x \eta_x}(L_f^2 + 2\frac{L_g^2}{\rho^2}) + \frac{1}{100c_0^2}\frac{1}{K}\frac{288}{\gamma_y \eta_x}\frac{(\rho^2 L_f^2 + L_g^2)}{\mu^2}(L_f^2 + \frac{L_g^2}{\rho^2}) + \frac{1}{100c_0^2}\frac{1}{K}\frac{288}{\gamma_z \eta_x}\frac{L_g^2}{\mu^2}\frac{L_g^2}{\rho^2}\right)$$

$$\leq 0. \tag{87}$$

Then, we can set

$$
c_2 = \frac{1}{(1-\lambda)}\left[9\eta_x(L_f^2 + 2\frac{1}{\rho^2}L_g^2) + 2c_0\eta_x\frac{L_{h_\rho}^2}{\rho^2} + 2c_1\eta_x\frac{L_g^2}{\rho^2} + 324\eta_x(L_f^2 + 2\frac{L_g^2}{\rho^2}) + 3240c_0\eta_x(L_f^2 + \frac{L_g^2}{\rho^2})\right.
$$

$$
\left. + 3240c_1\eta_x\frac{L_g^2}{\rho^2} + \frac{1}{K}\frac{3456}{\gamma_x\eta_x}(L_f^2 + 2\frac{L_g^2}{\rho^2}) + \frac{1}{100c_0^2}\frac{1}{K}\frac{3456}{\gamma_y\eta_x}\frac{(\rho^2 L_f^2 + L_g^2)}{\mu^2}(L_f^2 + \frac{L_g^2}{\rho^2}) + \frac{1}{100c_1^2}\frac{1}{K}\frac{3456}{\gamma_z\eta_x}\frac{L_g^2}{\mu^2}\frac{L_g^2}{\rho^2}\right]. \quad (88)
$$

To eliminate the consensus error term $\mathbb{E}[\|U_{x,t} - \bar{U}_{x,t}\|_F^2]$, we enforce

$$
c_2\frac{\eta_x^2}{1-\lambda} + 3\eta_x^2 C_1 - (1-\lambda)c_5
$$

$$
\leq \frac{\eta_x^2}{1-\lambda}\frac{1}{(1-\lambda)}\left[9\eta_x(L_f^2 + 2\frac{1}{\rho^2}L_g^2) + 2c_0\eta_x\frac{L_{h_\rho}^2}{\rho^2} + 2c_1\eta_x\frac{L_g^2}{\rho^2} + 324\eta_x(L_f^2 + 2\frac{L_g^2}{\rho^2}) + 3240c_0\eta_x(L_f^2 + \frac{L_g^2}{\rho^2})\right.
$$

$$
\left. + 3240c_1\eta_x\frac{L_g^2}{\rho^2} + \frac{1}{K}\frac{3456}{\gamma_x\eta_x}(L_f^2 + 2\frac{L_g^2}{\rho^2}) + \frac{1}{100c_0^2}\frac{1}{K}\frac{3456}{\gamma_y\eta_x}\frac{(\rho^2 L_f^2 + L_g^2)}{\mu^2}(L_f^2 + \frac{L_g^2}{\rho^2}) + \frac{1}{100c_1^2}\frac{1}{K}\frac{3456}{\gamma_z\eta_x}\frac{L_g^2}{\mu^2}\frac{L_g^2}{\rho^2}\right]
$$

$$
+ 3\eta_x^2\left[27\eta_x(L_f^2 + 2\frac{L_g^2}{\rho^2}) + 270c_0\eta_x(L_f^2 + \frac{L_g^2}{\rho^2}) + 270c_1\eta_x\frac{L_g^2}{\rho^2} + \frac{1}{K}\frac{288}{\gamma_x\eta_x}(L_f^2 + 2\frac{L_g^2}{\rho^2})\right.
$$

$$
\left. + \frac{1}{100c_0^2}\frac{1}{K}\frac{288}{\gamma_y\eta_x}\frac{(\rho^2 L_f^2 + L_g^2)}{\mu^2}(L_f^2 + \frac{L_g^2}{\rho^2}) + \frac{1}{100c_0^2}\frac{1}{K}\frac{288}{\gamma_z\eta_x}\frac{L_g^2}{\mu^2}\frac{L_g^2}{\rho^2}\right] - (1-\lambda)^2\eta_x
$$

$$
\leq 0. \quad (89)
$$

Here, we can solve the following four inequalities:

$$
\frac{\eta_x^2}{(1-\lambda)^2}\left[9\eta_x(L_f^2 + 2\frac{1}{\rho^2}L_g^2) + 2c_0\eta_x\frac{L_{h_\rho}^2}{\rho^2} + 2c_1\eta_x\frac{L_g^2}{\rho^2}\right.
$$

$$
\left. + 324\eta_x(L_f^2 + 2\frac{L_g^2}{\rho^2}) + 3240c_0\eta_x(L_f^2 + \frac{L_g^2}{\rho^2}) + 3240c_1\eta_x\frac{L_g^2}{\rho^2}\right] \leq \frac{(1-\lambda)^2\eta_x}{4},
$$

$$
\frac{\eta_x^2}{(1-\lambda)^2}\left[\frac{1}{K}\frac{3456}{\gamma_x\eta_x}(L_f^2 + 2\frac{L_g^2}{\rho^2}) + \frac{1}{100c_0^2}\frac{1}{K}\frac{3456}{\gamma_y\eta_x}\frac{(\rho^2 L_f^2 + L_g^2)}{\mu^2}(L_f^2 + \frac{L_g^2}{\rho^2}) + \frac{1}{100c_1^2}\frac{1}{K}\frac{3456}{\gamma_z\eta_x}\frac{L_g^2}{\mu^2}\frac{L_g^2}{\rho^2}\right] \leq \frac{(1-\lambda)^2\eta_x}{4},
$$

$$
3\eta_x^2\left[27\eta_x(L_f^2 + 2\frac{L_g^2}{\rho^2}) + 270c_0\eta_x(L_f^2 + \frac{L_g^2}{\rho^2}) + 270c_1\eta_x\frac{L_g^2}{\rho^2}\right] \leq \frac{(1-\lambda)^2\eta_x}{4},
$$

$$
3\eta_x^2\left[\frac{1}{K}\frac{288}{\gamma_x\eta_x}(L_f^2 + 2\frac{L_g^2}{\rho^2}) + \frac{1}{100c_0^2}\frac{1}{K}\frac{288}{\gamma_y\eta_x}\frac{(\rho^2 L_f^2 + L_g^2)}{\mu^2}(L_f^2 + \frac{L_g^2}{\rho^2}) + \frac{1}{100c_0^2}\frac{1}{K}\frac{288}{\gamma_z\eta_x}\frac{L_g^2}{\mu^2}\frac{L_g^2}{\rho^2}\right] \leq \frac{(1-\lambda)^2\eta_x}{4}. \quad (90)
$$

We can obtain

$$
\eta_x \leq \frac{(1-\lambda)^2\rho\mu}{2}\left/\sqrt{\left[72(\rho^2 L_f^2 + L_g^2)L_{h_\rho}^2 + 666\mu^2(\rho^2 L_f^2 + L_g^2) + 3240\times 36(\rho^2 L_f^2 + L_g^2)^2 + 3242\times 36L_g^4\right]}\right.,
$$

$$
\eta_x \leq \frac{(1-\lambda)\rho\mu}{2\sqrt{3}}\left/\sqrt{\left[27\mu^2(\rho^2 L_f^2 + L_g^2) + 270\times 36(\rho^2 L_f^2 + L_g^2)^2 + 270\times 36L_g^4\right]}\right.,
$$

$$
\gamma_x \geq \left\{\frac{12\times 6912(\rho^2 L_f^2 + L_g^2)}{(1-\lambda)^4 K\rho^2}, \frac{36\times 576(\rho^2 L_f^2 + L_g^2)}{(1-\lambda)^2 K\rho^2}\right\},
$$

$$
\gamma_y \geq \left\{\frac{12}{(1-\lambda)^4}\frac{8\mu^2}{300K\rho^2}, \frac{1}{(1-\lambda)^2}\frac{2\mu^2}{25K\rho^2}\right\},
$$

$$
\gamma_z \geq \left\{\frac{12}{(1-\lambda)^4}\frac{8\mu^2}{300K\rho^2}, \frac{1}{(1-\lambda)^2}\frac{2\mu^2 L_g^4}{25K(\rho^2 L_f^2 + L_g^2)^2\rho^2}\right\}. \quad (91)
$$

To eliminate the consensus error term $\mathbb{E}[\|Y_t - \bar{Y}_t\|_F^2]$, we enforce

$$9\eta_x(L_f^2 + \frac{1}{\rho^2}L_g^2) + \frac{c_0\eta_x}{\eta_y}2\eta_y\frac{L_{h_\rho}^2}{\rho^2} + 12C_2 - (1-\lambda)c_3 \leq 0 \,. \tag{92}$$

Then, we can set

$$c_3 = \frac{1}{1-\lambda}\left[\frac{9\eta_y}{10c_0}(L_f^2 + \frac{1}{\rho^2}L_g^2) + \frac{\eta_y}{5}\frac{L_{h_\rho}^2}{\rho^2} + 324\frac{\eta_y}{10c_0}(L_f^2 + \frac{L_g^2}{\rho^2}) + 324\eta_y(L_f^2 + \frac{L_g^2}{\rho^2})\right.$$
$$\left. + \frac{34560c_0}{\gamma_x\eta_y}\frac{1}{K}(L_f^2 + \frac{L_g^2}{\rho^2}) + \frac{3456(\rho^2L_f^2 + L_g^2)}{\mu^2}\frac{1}{10c_0}\frac{1}{\gamma_y\eta_y}\frac{1}{K}(L_f^2 + \frac{L_g^2}{\rho^2})\right] \,. \tag{93}$$

To eliminate the consensus error term $\mathbb{E}[\|U_{y,t} - \bar{U}_{y,t}\|_F^2]$, we enforce

$$c_3\frac{\eta_y^2}{1-\lambda} + 3\eta_y^2 C_2 - (1-\lambda)c_6$$
$$\leq \frac{\eta_y^2}{1-\lambda}\frac{1}{1-\lambda}\left[\frac{9\eta_y}{10c_0}(L_f^2 + \frac{1}{\rho^2}L_g^2) + \frac{\eta_y}{5}\frac{L_{h_\rho}^2}{\rho^2} + 324\frac{\eta_y}{10c_0}(L_f^2 + \frac{L_g^2}{\rho^2}) + 324\eta_y(L_f^2 + \frac{L_g^2}{\rho^2})\right.$$
$$\left. + \frac{34560c_0}{\gamma_x\eta_y}\frac{1}{K}(L_f^2 + \frac{L_g^2}{\rho^2}) + \frac{3456(\rho^2L_f^2 + L_g^2)}{\mu^2}\frac{1}{10c_0}\frac{1}{\gamma_y\eta_y}\frac{1}{K}(L_f^2 + \frac{L_g^2}{\rho^2})\right]$$
$$+ 3\eta_y^2\left[27\frac{\eta_y}{10c_0}(L_f^2 + \frac{L_g^2}{\rho^2}) + 27\eta_y(L_f^2 + \frac{L_g^2}{\rho^2})\right.$$
$$\left. + \frac{2880c_0}{\gamma_x\eta_y}\frac{1}{K}(L_f^2 + \frac{L_g^2}{\rho^2}) + \frac{288(\rho^2L_f^2 + L_g^2)}{\mu^2}\frac{1}{10c_0}\frac{1}{\gamma_y\eta_y}\frac{1}{K}(L_f^2 + \frac{L_g^2}{\rho^2})\right] - (1-\lambda)^2\eta_y$$
$$\leq 0 \,. \tag{94}$$

Here, we can solve the following four inequalities:

$$\frac{\eta_y^2}{(1-\lambda)^2}\left[\frac{9\eta_y}{10c_0}(L_f^2 + \frac{1}{\rho^2}L_g^2) + \frac{\eta_y}{5}\frac{L_{h_\rho}^2}{\rho^2} + 324\frac{\eta_y}{10c_0}(L_f^2 + \frac{L_g^2}{\rho^2}) + 324\eta_y(L_f^2 + \frac{L_g^2}{\rho^2})\right] \leq \frac{(1-\lambda)^2\eta_y}{4} \,,$$

$$\frac{\eta_y^2}{(1-\lambda)^2}\left[\frac{34560c_0}{\gamma_x\eta_y}\frac{1}{K}(L_f^2 + \frac{L_g^2}{\rho^2}) + \frac{3456(\rho^2L_f^2 + L_g^2)}{\mu^2}\frac{1}{10c_0}\frac{1}{\gamma_y\eta_y}\frac{1}{K}(L_f^2 + \frac{L_g^2}{\rho^2})\right] \leq \frac{(1-\lambda)^2\eta_y}{4} \,,$$

$$3\eta_y^2\left[27\frac{\eta_y}{10c_0}(L_f^2 + \frac{L_g^2}{\rho^2}) + 27\eta_y(L_f^2 + \frac{L_g^2}{\rho^2})\right] \leq \frac{(1-\lambda)^2\eta_y}{4} \,,$$

$$3\eta_y^2\left[\frac{2880c_0}{\gamma_x\eta_y}\frac{1}{K}(L_f^2 + \frac{L_g^2}{\rho^2}) + \frac{288(\rho^2L_f^2 + L_g^2)}{\mu^2}\frac{1}{10c_0}\frac{1}{\gamma_y\eta_y}\frac{1}{K}(L_f^2 + \frac{L_g^2}{\rho^2})\right] \leq \frac{(1-\lambda)^2\eta_y}{4} \,. \tag{95}$$

We can obtain

$$\eta_y \leq \min\left\{\frac{(1-\lambda)^2\rho}{2}\Big/\sqrt{\left[\frac{\mu^2}{40} + \frac{L_{h_\rho}^2}{5} + \frac{9\mu^2}{10} + 324(\rho^2L_f^2 + L_g^2)\right]}, \frac{(1-\lambda)\rho}{6}\Big/\sqrt{\left[\frac{\mu^2}{40} + 9(\rho^2L_f^2 + L_g^2)\right]}\right\} \,,$$

$$\gamma_x \geq \left\{\frac{34560 \times 288(\rho^2L_f^2 + L_g^2)^2}{(1-\lambda)^4\mu^2 K\rho^2}, \frac{2880 \times 36 \times 24(\rho^2L_f^2 + L_g^2)^2}{(1-\lambda)^2\mu^2 K\rho^2}\right\} \,,$$

$$\gamma_y \geq \left\{\frac{384(\rho^2L_f^2 + L_g^2)}{5(1-\lambda)^4 K\rho^2}, \frac{96(\rho^2L_f^2 + L_g^2)}{5(1-\lambda)^2 K\rho^2}\right\} \,. \tag{96}$$

To eliminate the consensus error term $\mathbb{E}[\|Z_t - \bar{Z}_t\|_F^2]$, we enforce

$$9\eta_x\frac{1}{\rho^2}L_g^2 + \frac{c_1\eta_x}{\eta_z}2\eta_z\frac{L_g^2}{\rho^2} + 12C_3 - (1-\lambda)c_4$$

$$\leq \frac{9\eta_z}{10c_1}\frac{1}{\rho^2}L_g^2 + \frac{\eta_z}{5}\frac{L_g^2}{\rho^2}$$
$$+ 324\frac{\eta_z}{10c_1}\frac{L_g^2}{\rho^2} + 324\eta_z\frac{L_g^2}{\rho^2} + \frac{34560c_1}{\gamma_x\eta_z}\frac{L_g^2}{\rho^2}\frac{1}{K} + \frac{3456L_g^2}{\mu^2}\frac{1}{10c_1}\frac{1}{\gamma_z\eta_z}\frac{L_g^2}{\rho^2}\frac{1}{K} - (1-\lambda)c_4$$
$$\leq 0 . \tag{97}$$

Then, we can set

$$c_4 = \frac{1}{1-\lambda}\left[\frac{9\eta_z}{10c_1}\frac{1}{\rho^2}L_g^2 + \frac{\eta_z}{5}\frac{L_g^2}{\rho^2} + 324\frac{\eta_z}{10c_1}\frac{L_g^2}{\rho^2} + 324\eta_z\frac{L_g^2}{\rho^2} + \frac{34560c_1}{\gamma_x\eta_z}\frac{L_g^2}{\rho^2}\frac{1}{K} + \frac{3456L_g^2}{\mu^2}\frac{1}{10c_1}\frac{1}{\gamma_z\eta_z}\frac{L_g^2}{\rho^2}\frac{1}{K}\right] . \tag{98}$$

To eliminate the consensus error term $\mathbb{E}[\|U_{z,t} - \bar{U}_{z,t}\|_F^2]$, we enforce

$$c_4\frac{\eta_z^2}{1-\lambda} + 3\eta_z^2 C_3 - (1-\lambda)c_7$$
$$\leq \frac{\eta_z^2}{(1-\lambda)^2}\left[\frac{9\eta_z}{10c_1}\frac{1}{\rho^2}L_g^2 + \frac{\eta_z}{5}\frac{L_g^2}{\rho^2} + 324\frac{\eta_z}{10c_1}\frac{L_g^2}{\rho^2} + 324\eta_z\frac{L_g^2}{\rho^2} + \frac{34560c_1}{\gamma_x\eta_z}\frac{L_g^2}{\rho^2}\frac{1}{K} + \frac{3456L_g^2}{\mu^2}\frac{1}{10c_1}\frac{1}{\gamma_z\eta_z}\frac{L_g^2}{\rho^2}\frac{1}{K}\right]$$
$$+ 3\eta_z^2\left[27\frac{\eta_z}{10c_1}\frac{L_g^2}{\rho^2} + 27\eta_z\frac{L_g^2}{\rho^2} + \frac{2880c_1}{\gamma_x\eta_z}\frac{L_g^2}{\rho^2}\frac{1}{K} + \frac{288L_g^2}{\mu^2}\frac{1}{10c_1}\frac{1}{\gamma_z\eta_z}\frac{L_g^2}{\rho^2}\frac{1}{K}\right] - (1-\lambda)^2\eta_z$$
$$\leq 0 . \tag{99}$$

Here, we can solve the following four inequalities:

$$\frac{\eta_z^2}{(1-\lambda)^2}\left[\frac{9\eta_z}{10c_1}\frac{1}{\rho^2}L_g^2 + \frac{\eta_z}{5}\frac{L_g^2}{\rho^2} + 324\frac{\eta_z}{10c_1}\frac{L_g^2}{\rho^2} + 324\eta_z\frac{L_g^2}{\rho^2}\right] \leq \frac{(1-\lambda)^2\eta_z}{4} ,$$
$$\frac{\eta_z^2}{(1-\lambda)^2}\left[\frac{34560c_1}{\gamma_x\eta_z}\frac{L_g^2}{\rho^2}\frac{1}{K} + \frac{3456L_g^2}{\mu^2}\frac{1}{10c_1}\frac{1}{\gamma_z\eta_z}\frac{L_g^2}{\rho^2}\frac{1}{K}\right] \leq \frac{(1-\lambda)^2\eta_z}{4} ,$$
$$3\eta_z^2\left[27\frac{\eta_z}{10c_1}\frac{L_g^2}{\rho^2} + 27\eta_z\frac{L_g^2}{\rho^2}\right] \leq \frac{(1-\lambda)^2\eta_z}{4} ,$$
$$3\eta_z^2\left[\frac{2880c_1}{\gamma_x\eta_z}\frac{L_g^2}{\rho^2}\frac{1}{K} + \frac{288L_g^2}{\mu^2}\frac{1}{10c_1}\frac{1}{\gamma_z\eta_z}\frac{L_g^2}{\rho^2}\frac{1}{K}\right] \leq \frac{(1-\lambda)^2\eta_z}{4} . \tag{100}$$

We can obtain

$$\eta_z \leq \min\left\{\frac{(1-\lambda)^2\rho}{2L_g}\bigg/\sqrt{\left[\frac{1621}{5} + \frac{37\mu^2}{40L_g^2}\right]}, \frac{(1-\lambda)}{36}\bigg/\sqrt{\left[\frac{\mu^2}{360L_g^2} + 1\right]}\right\} ,$$
$$\gamma_x \geq \left\{\frac{34560 \times 288L_g^4}{(1-\lambda)^4\mu^2 K\rho^2}, \frac{2880 \times 36 \times 24L_g^4}{(1-\lambda)^2\mu^2 K\rho^2}\right\} ,$$
$$\gamma_z \geq \left\{\frac{384L_g^2}{5(1-\lambda)^4 K\rho^2}, \frac{96L_g^2}{5(1-\lambda)^2 K\rho^2}\right\} . \tag{101}$$

Therefore, by setting

$$c_2 = \frac{\eta_x}{1-\lambda}\left[9(L_f^2 + 2\frac{1}{\rho^2}L_g^2) + 2c_0\frac{L_{h_\rho}^2}{\rho^2} + 2c_1\frac{L_g^2}{\rho^2} + 324(L_f^2 + 2\frac{L_g^2}{\rho^2}) + 3240c_0(L_f^2 + \frac{L_g^2}{\rho^2}) + 3240c_1\frac{L_g^2}{\rho^2}\right]$$
$$+ \frac{1}{(1-\lambda)\eta_x}\left[\frac{3456}{\gamma_x K}(L_f^2 + 2\frac{L_g^2}{\rho^2}) + \frac{1}{100c_0^2}\frac{3456}{\gamma_y K}\frac{(\rho^2 L_f^2 + L_g^2)}{\mu^2}(L_f^2 + \frac{L_g^2}{\rho^2}) + \frac{1}{100c_1^2}\frac{3456}{\gamma_z K}\frac{L_g^2}{\mu^2}\frac{L_g^2}{\rho^2}\right] ,$$

$$c_3 = \frac{\eta_y}{1-\lambda}\left[\frac{9}{10c_0}(L_f^2 + \frac{1}{\rho^2}L_g^2) + \frac{1}{5}\frac{L_{h_\rho}^2}{\rho^2} + \frac{324}{10c_0}(L_f^2 + \frac{L_g^2}{\rho^2}) + 324(L_f^2 + \frac{L_g^2}{\rho^2})\right]$$

$$+ \frac{1}{(1-\lambda)\eta_y}\left[\frac{34560c_0}{\gamma_x K}(L_f^2 + \frac{L_g^2}{\rho^2}) + \frac{3456(\rho^2 L_f^2 + L_g^2)}{\mu^2}\frac{1}{10c_0}\frac{1}{\gamma_y K}(L_f^2 + \frac{L_g^2}{\rho^2})\right],$$

$$c_4 = \frac{\eta_z}{1-\lambda}\left[\frac{9}{10c_1}\frac{1}{\rho^2}L_g^2 + \frac{1}{5}\frac{L_g^2}{\rho^2} + \frac{324}{10c_1}\frac{L_g^2}{\rho^2} + 324\frac{L_g^2}{\rho^2}\right] + \frac{1}{(1-\lambda)\eta_z}\left[\frac{34560c_1}{\gamma_x K}\frac{L_g^2}{\rho^2} + \frac{3456L_g^2}{\mu^2}\frac{1}{10c_1}\frac{1}{\gamma_z K}\frac{L_g^2}{\rho^2}\right], \quad (102)$$

we have

$$P_{t+1} - P_t$$

$$\leq -\frac{\eta_x}{2}\mathbb{E}[\|\nabla\mathcal{L}(\bar{x}_t)\|^2] + \frac{3\eta_x}{2}\mathbb{E}[\|\nabla\mathcal{L}(\bar{x}_t) - \nabla\mathcal{L}_\rho(\bar{x}_t)\|^2]$$

$$+ c_5\frac{9\gamma_x^2\eta_x^4\sigma^2}{1-\lambda}(1 + \frac{2}{\rho^2}) + c_6\frac{6\gamma_y^2\eta_y^4\sigma^2}{1-\lambda}(1 + \frac{1}{\rho^2}) + c_7\frac{3\gamma_z^2\eta_z^4\sigma^2}{1-\lambda}\frac{1}{\rho^2}$$

$$+ 2c_8\gamma_x^2\eta_x^4\sigma^2\frac{1}{K} + 2c_9\frac{1}{\rho^2}\gamma_x^2\eta_x^4\sigma^2\frac{1}{K} + 2c_{10}\frac{1}{\rho^2}\gamma_x^2\eta_x^4\sigma^2\frac{1}{K} + 2c_{11}\gamma_y^2\eta_y^4\sigma^2\frac{1}{K} + 2c_{12}\frac{1}{\rho^2}\gamma_y^2\eta_y^4\sigma^2\frac{1}{K} + 2c_{13}\frac{1}{\rho^2}\gamma_z^2\eta_z^4\sigma^2\frac{1}{K}$$

$$+ 2c_{14}\gamma_x^2\eta_x^4\sigma^2 + 2c_{15}\frac{1}{\rho^2}\gamma_x^2\eta_x^4\sigma^2 + 2c_{16}\frac{1}{\rho^2}\gamma_x^2\eta_x^4\sigma^2 + 2c_{17}\gamma_y^2\eta_y^4\sigma^2 + 2c_{18}\frac{1}{\rho^2}\gamma_y^2\eta_y^4\sigma^2 + 2c_{19}\frac{1}{\rho^2}\gamma_z^2\eta_z^4\sigma^2$$

$$+ \left[3\eta_y^2 C_2 - \frac{c_0\eta_x}{\eta_y}\frac{\eta_y}{4}\right]\mathbb{E}[\|\bar{m}_{y,t}\|^2] + \left[3\eta_z^2 C_3 - \frac{c_1\eta_x}{\eta_z}\frac{\eta_z}{4}\right]\mathbb{E}[\|\bar{m}_{z,t}\|^2]$$

$$+ \left[\frac{c_0\eta_x}{\eta_y}\left(\frac{\eta_x}{2} + \frac{1}{\rho}\frac{\eta_x^2 L_{h_\rho}}{2} + \frac{1}{\rho}\frac{3\eta_x^2 L_{h_\rho}}{2} + \frac{1}{\rho}\frac{\eta_x^2 L_{\hat{h}_\rho}}{2}\right)\right.$$

$$\left. + \frac{c_1\eta_x}{\eta_z}\left(\frac{\eta_x}{2} + 2\eta_x^2 L_g\frac{1}{\rho} + \frac{\eta_x^2 L_{\hat{g}}}{2}\frac{1}{\rho}\right) + 3\eta_x^2 C_1 - \frac{\eta_x}{4}\right]\mathbb{E}[\|\bar{m}_{x,t}\|^2]. \quad (103)$$

To eliminate $\mathbb{E}[\|\bar{m}_{y,t}\|^2]$, we solve the following inequality:

$$3\eta_y^2 C_2 - \frac{c_0\eta_x}{\eta_y}\frac{\eta_y}{4}$$

$$= 3\eta_y^2 C_2 - \frac{\eta_y}{40}$$

$$\leq 3\eta_y^2\left[27\frac{\eta_y}{10c_0}(L_f^2 + \frac{L_g^2}{\rho^2}) + 27\eta_y(L_f^2 + \frac{L_g^2}{\rho^2}) + \frac{2880c_0}{\gamma_x\eta_y}\frac{1}{K}(L_f^2 + \frac{L_g^2}{\rho^2})\right.$$

$$\left. + \frac{288(\rho^2 L_f^2 + L_g^2)}{\mu^2}\frac{1}{10c_0}\frac{1}{\gamma_y\eta_y}\frac{1}{K}(L_f^2 + \frac{L_g^2}{\rho^2})\right] - \frac{\eta_y}{40}$$

$$\leq 0. \quad (104)$$

Specifically, we enforce

$$3\eta_y^2 \times 27\frac{\eta_y}{10c_0}(L_f^2 + \frac{L_g^2}{\rho^2}) \leq \frac{\eta_y}{160},$$

$$3\eta_y^2 \times 27\eta_y(L_f^2 + \frac{L_g^2}{\rho^2}) \leq \frac{\eta_y}{160},$$

$$3\eta_y^2 \times \frac{2880c_0}{\gamma_x\eta_y}\frac{1}{K}(L_f^2 + \frac{L_g^2}{\rho^2}) \leq \frac{\eta_y}{160},$$

$$3\eta_y^2\frac{288(\rho^2 L_f^2 + L_g^2)}{\mu^2}\frac{1}{10c_0}\frac{1}{\gamma_y\eta_y}\frac{1}{K}(L_f^2 + \frac{L_g^2}{\rho^2}) \leq \frac{\eta_y}{160}. \quad (105)$$

We can obtain

$$\eta_y \leq \min\left\{\frac{\rho}{6\mu} , \frac{\rho}{36\sqrt{10(\rho^2 L_f^2 + L_g^2)}}\right\} ,$$

$$\gamma_x \geq \frac{2880 \times 36 \times 480(\rho^2 L_f^2 + L_g^2)^2}{K\mu^2\rho^2} ,$$

$$\gamma_y \geq \frac{1920(\rho^2 L_f^2 + L_g^2)}{5K\rho^2} . \tag{106}$$

To eliminate $\mathbb{E}[\|\bar{m}_{z,t}\|^2]$, we solve the following inequality:

$$3\eta_z^2 C_3 - \frac{c_1\eta_x}{\eta_z}\frac{\eta_z}{4}$$

$$= 3\eta_z^2 C_3 - \frac{\eta_z}{40}$$

$$\leq 3\eta_z^2\left[27\frac{\eta_z}{10c_1}\frac{L_g^2}{\rho^2} + 27\eta_z\frac{L_g^2}{\rho^2} + \frac{2880c_1}{\gamma_x\eta_z}\frac{L_g^2}{\rho^2}\frac{1}{K} + \frac{288L_g^2}{\mu^2}\frac{1}{10c_1}\frac{1}{\gamma_z\eta_z}\frac{L_g^2}{\rho^2}\frac{1}{K}\right] - \frac{\eta_z}{40}$$

$$\leq 0 . \tag{107}$$

Specifically, we enforce

$$3\eta_z^2 \times 27\frac{\eta_z}{10c_1}\frac{L_g^2}{\rho^2} \leq \frac{\eta_z}{160} ,$$

$$3\eta_z^2 \times 27\eta_z\frac{L_g^2}{\rho^2} \leq \frac{\eta_z}{160} ,$$

$$3\eta_z^2\frac{2880c_1}{\gamma_x\eta_z}\frac{L_g^2}{\rho^2}\frac{1}{K} \leq \frac{\eta_z}{160} ,$$

$$3\eta_z^2\frac{288L_g^2}{\mu^2}\frac{1}{10c_1}\frac{1}{\gamma_z\eta_z}\frac{L_g^2}{\rho^2}\frac{1}{K} \leq \frac{\eta_z}{160} . \tag{108}$$

We can obtain

$$\eta_z \leq \min\left\{\frac{\rho}{6\mu} , \frac{\rho}{36\sqrt{10}L_g}\right\} ,$$

$$\gamma_x \geq \frac{2880 \times 36 \times 480L_g^4}{\mu^2\rho^2 K} ,$$

$$\gamma_z \geq \frac{1920L_g^2}{5\rho^2 K} . \tag{109}$$

To eliminate $\mathbb{E}[\|\bar{m}_{x,t}\|^2]$, we solve the following inequality:

$$\frac{c_0\eta_x}{\eta_y}\left(\frac{\eta_x}{2} + \frac{1}{\rho}\frac{\eta_x^2 L_{h_\rho}}{2} + \frac{1}{\rho}\frac{3\eta_x^2 L_{h_\rho}}{2} + \frac{1}{\rho}\frac{\eta_x^2 L_{\hat{h}_\rho}}{2}\right) + \frac{c_1\eta_x}{\eta_z}\left(\frac{\eta_x}{2} + 2\eta_x^2 L_g\frac{1}{\rho} + \frac{\eta_x^2 L_{\hat{g}}}{2}\frac{1}{\rho}\right) + 3\eta_x^2 C_1 - \frac{\eta_x}{4}$$

$$= \frac{1}{10}\left(\frac{\eta_x}{2} + \frac{1}{\rho}\frac{\eta_x^2 L_{h_\rho}}{2} + \frac{1}{\rho}\frac{3\eta_x^2 L_{h_\rho}}{2} + \frac{1}{\rho}\frac{\eta_x^2 L_{\hat{h}_\rho}}{2}\right) + \frac{1}{10}\left(\frac{\eta_x}{2} + 2\eta_x^2 L_g\frac{1}{\rho} + \frac{\eta_x^2 L_{\hat{g}}}{2}\frac{1}{\rho}\right) + 3\eta_x^2 C_1 - \frac{\eta_x}{4}$$

$$\leq \frac{1}{10}\left(\frac{\eta_x}{2} + \frac{1}{\rho}\frac{\eta_x^2 L_{h_\rho}}{2} + \frac{1}{\rho}\frac{3\eta_x^2 L_{h_\rho}}{2} + \frac{1}{\rho}\frac{\eta_x^2 L_{\hat{h}_\rho}}{2}\right) + \frac{1}{10}\left(\frac{\eta_x}{2} + 2\eta_x^2 L_g\frac{1}{\rho} + \frac{\eta_x^2 L_{\hat{g}}}{2}\frac{1}{\rho}\right) - \frac{\eta_x}{4}$$

$$+ 3\eta_x^2\left[27\eta_x(L_f^2 + 2\frac{L_g^2}{\rho^2}) + 270c_0\eta_x(L_f^2 + \frac{L_g^2}{\rho^2}) + 270c_1\eta_x\frac{L_g^2}{\rho^2}\right.$$

$$+ \frac{1}{K}\frac{288}{\gamma_x \eta_x}(L_f^2 + 2\frac{L_g^2}{\rho^2}) + \frac{1}{100c_0^2}\frac{1}{K}\frac{288}{\gamma_y \eta_x}\frac{(\rho^2 L_f^2 + L_g^2)}{\mu^2}(L_f^2 + \frac{L_g^2}{\rho^2}) + \frac{1}{100c_0^2}\frac{1}{K}\frac{288}{\gamma_z \eta_x}\frac{L_g^2}{\mu^2}\frac{L_g^2}{\rho^2}\Bigg]$$

$$\le 0 \,. \tag{110}$$

Specifically, we enforce

$$\frac{1}{10}\left(\frac{\eta_x}{2} + \frac{1}{\rho}\frac{\eta_x^2 L_{h_\rho}}{2} + \frac{1}{\rho}\frac{3\eta_x^2 L_{h_\rho}}{2} + \frac{1}{\rho}\frac{\eta_x^2 L_{\hat{h}_\rho}}{2}\right) \le \frac{\eta_x}{16} \,,$$

$$\frac{1}{10}\left(\frac{\eta_x}{2} + 2\eta_x^2 L_g \frac{1}{\rho} + \frac{\eta_x^2 L_{\hat{g}}}{2}\frac{1}{\rho}\right) \le \frac{\eta_x}{16} \,,$$

$$3\eta_x^2\left[27\eta_x(L_f^2 + 2\frac{L_g^2}{\rho^2}) + 270c_0\eta_x(L_f^2 + \frac{L_g^2}{\rho^2}) + 270c_1\eta_x\frac{L_g^2}{\rho^2}\right] \le \frac{\eta_x}{16} \,,$$

$$3\eta_x^2\left[\frac{1}{K}\frac{288}{\gamma_x \eta_x}(L_f^2 + 2\frac{L_g^2}{\rho^2}) + \frac{1}{100c_0^2}\frac{1}{K}\frac{288}{\gamma_y \eta_x}\frac{(\rho^2 L_f^2 + L_g^2)}{\mu^2}(L_f^2 + \frac{L_g^2}{\rho^2}) + \frac{1}{100c_0^2}\frac{1}{K}\frac{288}{\gamma_z \eta_x}\frac{L_g^2}{\mu^2}\frac{L_g^2}{\rho^2}\right] \le \frac{\eta_x}{16} \,. \tag{111}$$

We can obtain

$$\eta_x \le \min\left\{\frac{\rho}{4(4L_{h_\rho} + L_{\hat{h}_\rho})}, \frac{\rho}{4(4L_g + L_{\hat{g}})}\right\},$$

$$\gamma_x \ge \frac{288 \times 144(\rho^2 L_f^2 + L_g^2)}{K\rho^2},$$

$$\gamma_y \ge \frac{8\mu^2}{25K\rho^2},$$

$$\gamma_z \ge \frac{8\mu^2 L_g^4}{25(\rho^2 L_f^2 + L_g^2)^2\rho^2 K} \,. \tag{112}$$

In summary, by setting

$$\eta_x \le \min\Bigg\{\frac{(1-\lambda)^2\rho\mu}{2}\Big/\sqrt{\left[72(\rho^2 L_f^2 + L_g^2)L_{h_\rho}^2 + 666\mu^2(\rho^2 L_f^2 + L_g^2) + 3240 \times 36(\rho^2 L_f^2 + L_g^2)^2 + 3242 \times 36L_g^4\right]},$$

$$\frac{(1-\lambda)\rho\mu}{2\sqrt{3}}\Big/\sqrt{\left[27\mu^2(\rho^2 L_f^2 + L_g^2) + 270 \times 36(\rho^2 L_f^2 + L_g^2)^2 + 270 \times 36L_g^4\right]},$$

$$\frac{\rho}{4(4L_{h_\rho} + L_{\hat{h}_\rho})}, \frac{\rho}{4(4L_g + L_{\hat{g}})}, \frac{1}{2L_{\mathcal{L}}}, \eta_y\frac{\mu^2}{4L_{h_\rho}^2}, \frac{\mu^2}{4L_g^2}\eta_z, \frac{1}{\sqrt{\gamma_x}}\Bigg\}, \tag{113}$$

$$\eta_y \le \min\Bigg\{\frac{(1-\lambda)^2\rho}{2}\Big/\sqrt{\left[\frac{\mu^2}{40} + \frac{L_{h_\rho}^2}{5} + \frac{9\mu^2}{10} + 324(\rho^2 L_f^2 + L_g^2)\right]}, \frac{(1-\lambda)\rho}{6}\Big/\sqrt{\left[\frac{\mu^2}{40} + 9(\rho^2 L_f^2 + L_g^2)\right]},$$

$$\frac{\rho}{6\mu}, \frac{\rho}{36\sqrt{10(\rho^2 L_f^2 + L_g^2)}}, \frac{\rho}{2L_{h_\rho}}, \frac{1}{\sqrt{\gamma_y}}\Bigg\}, \tag{114}$$

$$\eta_z \le \min\Bigg\{\frac{(1-\lambda)^2\rho}{2L_g}\Big/\sqrt{\left[\frac{1621}{5} + \frac{37\mu^2}{40L_g^2}\right]}, \frac{(1-\lambda)}{36}\Big/\sqrt{\left[\frac{\mu^2}{360L_g^2} + 1\right]}, \frac{\rho}{6\mu}, \frac{\rho}{36\sqrt{10}L_g}, \frac{\rho}{2L_g}, \frac{1}{\sqrt{\gamma_z}}\Bigg\}, \tag{115}$$

$$\gamma_x \ge \min\Bigg\{\frac{12 \times 6912(\rho^2 L_f^2 + L_g^2)}{(1-\lambda)^4 K\rho^2}, \frac{36 \times 576(\rho^2 L_f^2 + L_g^2)}{(1-\lambda)^2 K\rho^2}, \frac{34560 \times 288(\rho^2 L_f^2 + L_g^2)^2}{(1-\lambda)^4\mu^2 K\rho^2},$$

$$\frac{2880 \times 36 \times 24(\rho^2 L_f^2 + L_g^2)^2}{(1-\lambda)^2\mu^2 K\rho^2}, \frac{34560 \times 288L_g^4}{(1-\lambda)^4\mu^2 K\rho^2}, \frac{2880 \times 36 \times 24L_g^4}{(1-\lambda)^2\mu^2 K\rho^2},$$

$$\frac{2880 \times 36 \times 480L_g^4}{\mu^2\rho^2 K}, \frac{2880 \times 36 \times 480(\rho^2 L_f^2 + L_g^2)^2}{K\mu^2\rho^2}, \frac{288 \times 144(\rho^2 L_f^2 + L_g^2)}{K\rho^2}\Bigg\}, \tag{116}$$

$$\gamma_y \geq \min\left\{ \frac{12}{(1-\lambda)^4}\frac{8\mu^2}{300K\rho^2}, \frac{1}{(1-\lambda)^2}\frac{2\mu^2}{25K\rho^2}, \frac{384(\rho^2 L_f^2 + L_g^2)}{5(1-\lambda)^4 K\rho^2}, \frac{96(\rho^2 L_f^2 + L_g^2)}{5(1-\lambda)^2 K\rho^2}, \frac{1920(\rho^2 L_f^2 + L_g^2)}{5K\rho^2}, \frac{8\mu^2}{25K\rho^2} \right\},$$

(117)

$$\gamma_z \geq \min\left\{ \frac{12}{(1-\lambda)^4}\frac{8\mu^2}{300K\rho^2}, \frac{1}{(1-\lambda)^2}\frac{2\mu^2 L_g^4}{25K(\rho^2 L_f^2 + L_g^2)^2\rho^2}, \frac{384 L_g^2}{5(1-\lambda)^4 K\rho^2}, \frac{96 L_g^2}{5(1-\lambda)^2 K\rho^2}, \right.$$
$$\left. \frac{1920 L_g^2}{5\rho^2 K}, \frac{8\mu^2 L_g^4}{25(\rho^2 L_f^2 + L_g^2)^2\rho^2 K} \right\},$$

(118)

we have

$$P_{t+1} - P_t$$
$$\leq -\frac{\eta_x}{2}\mathbb{E}[\|\nabla\mathcal{L}(\bar{x}_t)\|^2] + \frac{3\eta_x}{2}\mathbb{E}[\|\nabla\mathcal{L}(\bar{x}_t) - \nabla\mathcal{L}_\rho(\bar{x}_t)\|^2]$$
$$+ (1-\lambda)\eta_x\frac{9\gamma_x^2\eta_x^4\sigma^2}{1-\lambda}(1+\frac{2}{\rho^2}) + (1-\lambda)\eta_y\frac{6\gamma_y^2\eta_y^4\sigma^2}{1-\lambda}(1+\frac{1}{\rho^2}) + (1-\lambda)\eta_z\frac{3\gamma_z^2\eta_z^4\sigma^2}{1-\lambda}\frac{1}{\rho^2}$$
$$+ 2\frac{9}{\gamma_x\eta_x}\gamma_x^2\eta_x^4\sigma^2\frac{1}{K} + 2\frac{9}{\gamma_x\eta_x}\frac{1}{\rho^2}\gamma_x^2\eta_x^4\sigma^2\frac{1}{K} + 2\frac{9}{\gamma_x\eta_x}\frac{1}{\rho^2}\gamma_x^2\eta_x^4\sigma^2\frac{1}{K}$$
$$+ 2\frac{144(\rho^2 L_f^2 + L_g^2)}{\mu^2}\frac{\eta_x}{\gamma_y\eta_y^2}\gamma_y^2\eta_y^4\sigma^2\frac{1}{K} + 2\frac{144(\rho^2 L_f^2 + L_g^2)}{\mu^2}\frac{\eta_x}{\gamma_y\eta_y^2}\frac{1}{\rho^2}\gamma_y^2\eta_y^4\sigma^2\frac{1}{K}$$
$$+ 2\frac{72 L_g^2}{\mu^2}\frac{\eta_x}{\gamma_z\eta_z^2}\frac{1}{\rho^2}\gamma_z^2\eta_z^4\sigma^2\frac{1}{K} + 18\eta_x\gamma_x^2\eta_x^4\sigma^2 + 18\eta_x\frac{1}{\rho^2}\gamma_x^2\eta_x^4\sigma^2 + 18\eta_x\frac{1}{\rho^2}\gamma_x^2\eta_x^4\sigma^2$$
$$+ 18\eta_y\gamma_y^2\eta_y^4\sigma^2 + 18\eta_y\frac{1}{\rho^2}\gamma_y^2\eta_y^4\sigma^2 + 18\eta_z\frac{1}{\rho^2}\gamma_z^2\eta_z^4\sigma^2$$
$$\leq -\frac{\eta_x}{2}\mathbb{E}[\|\nabla\mathcal{L}(\bar{x}_t)\|^2] + \frac{3\eta_x}{2}\mathbb{E}[\|\nabla\mathcal{L}(\bar{x}_t) - \nabla\mathcal{L}_\rho(\bar{x}_t)\|^2]$$
$$+ \eta_x 9\gamma_x^2\eta_x^4\sigma^2(1+\frac{2}{\rho^2}) + \eta_y 6\gamma_y^2\eta_y^4\sigma^2(1+\frac{1}{\rho^2}) + \eta_z 3\gamma_z^2\eta_z^4\sigma^2\frac{1}{\rho^2}$$
$$+ 18\gamma_x\eta_x^3\sigma^2\frac{1}{K} + 18\frac{1}{\rho^2}\gamma_x\eta_x^3\sigma^2\frac{1}{K} + 18\frac{1}{\rho^2}\gamma_x\eta_x^3\sigma^2\frac{1}{K}$$
$$+ 2\eta_x\frac{144(\rho^2 L_f^2 + L_g^2)}{\mu^2}\gamma_y\eta_y^2\sigma^2\frac{1}{K} + 2\eta_x\frac{144(\rho^2 L_f^2 + L_g^2)}{\mu^2}\frac{1}{\rho^2}\gamma_y\eta_y^2\sigma^2\frac{1}{K}$$
$$+ 2\eta_x\frac{72 L_g^2}{\mu^2}\frac{1}{\rho^2}\gamma_z\eta_z^2\sigma^2\frac{1}{K} + 18\eta_x\gamma_x^2\eta_x^4\sigma^2 + 18\eta_x\frac{1}{\rho^2}\gamma_x^2\eta_x^4\sigma^2 + 18\eta_x\frac{1}{\rho^2}\gamma_x^2\eta_x^4\sigma^2$$
$$+ 18\eta_y\gamma_y^2\eta_y^4\sigma^2 + 18\eta_y\frac{1}{\rho^2}\gamma_y^2\eta_y^4\sigma^2 + 18\eta_z\frac{1}{\rho^2}\gamma_z^2\eta_z^4\sigma^2 \, .$$

(119)

By summing over $t$ from 0 to $T - 1$, we have

$$\frac{1}{T}\sum_{t=0}^{T-1}\mathbb{E}[\|\nabla\mathcal{L}(\bar{x}_t)\|^2]$$
$$\leq \frac{2}{\eta_x T}(P_0 - P_T) + 3\frac{1}{T}\sum_{t=0}^{T-1}\mathbb{E}[\|\nabla\mathcal{L}(\bar{x}_t) - \nabla\mathcal{L}_\rho(\bar{x}_t)\|^2]$$
$$+ 18(1+\frac{2}{\rho^2})\gamma_x^2\eta_x^4\sigma^2 + 12(1+\frac{1}{\rho^2})\frac{\eta_y}{\eta_x}\gamma_y^2\eta_y^4\sigma^2 + 6\frac{1}{\rho^2}\frac{\eta_z}{\eta_x}\gamma_z^2\eta_z^4\sigma^2$$
$$+ 36(1+\frac{2}{\rho^2})\gamma_x\eta_x^2\sigma^2\frac{1}{K} + \frac{576(\rho^2 L_f^2 + L_g^2)}{\mu^2}(1+\frac{1}{\rho^2})\gamma_y\eta_y^2\sigma^2\frac{1}{K} + \frac{288 L_g^2}{\mu^2}\frac{1}{\rho^2}\gamma_z\eta_z^2\sigma^2\frac{1}{K}$$
$$+ 36(1+\frac{2}{\rho^2})\gamma_x^2\eta_x^4\sigma^2 + 36(1+\frac{1}{\rho^2})\frac{\eta_y}{\eta_x}\gamma_y^2\eta_y^4\sigma^2 + 36\frac{1}{\rho^2}\frac{\eta_z}{\eta_x}\gamma_z^2\eta_z^4\sigma^2 \, .$$

(120)

When $t = 0$, the initial batch size is $S$, then we have

$$
\frac{1}{K}\mathbb{E}[\|U_{x,0} - \bar{U}_{x,0}\|_F^2]
$$

$$
= \frac{1}{K}\sum_{k=1}^{K}\mathbb{E}[\|\nabla_1 f^{(k)}(x_0^{(k)}, y_0^{(k)}; \xi_0^{(k)}) + \frac{1}{\rho}\nabla_1 g^{(k)}(x_0^{(k)}, y_0^{(k)}; \zeta_0^{(k)}) - \frac{1}{\rho}\nabla_1 g^{(k)}(x_0^{(k)}, z_0^{(k)}; \zeta_0^{(k)})
$$

$$
- \frac{1}{K}\sum_{k=1}^{K}\left(\nabla_1 f^{(k)}(x_0^{(k)}, y_0^{(k)}; \xi_0^{(k)}) + \frac{1}{\rho}\nabla_1 g^{(k)}(x_0^{(k)}, y_0^{(k)}; \zeta_0^{(k)}) - \frac{1}{\rho}\nabla_1 g^{(k)}(x_0^{(k)}, z_0^{(k)}; \zeta_0^{(k)})\right)\|^2]
$$

$$
\leq 18\frac{1}{K}\sum_{k=1}^{K}\mathbb{E}[\|\nabla_1 f^{(k)}(x_0, y_0)\|^2] + 18\frac{1}{\rho^2}\frac{1}{K}\sum_{k=1}^{K}\mathbb{E}[\|\nabla_1 g^{(k)}(x_0, y_0)\|^2]
$$

$$
+ 18\frac{1}{\rho^2}\frac{1}{K}\sum_{k=1}^{K}\mathbb{E}[\|\nabla_1 g^{(k)}(x_0, z_0)\|^2] + 18(1 + \frac{2}{\rho^2})\frac{\sigma^2}{S}\ ,
\tag{121}
$$

and

$$
\frac{1}{K}\mathbb{E}[\|U_{y,0} - \bar{U}_{y,0}\|_F^2] \leq 18\frac{1}{K}\sum_{k=1}^{K}\mathbb{E}[\|\nabla_2 f^{(k)}(x_0, y_0)\|^2] + 18\frac{1}{\rho^2}\frac{1}{K}\sum_{k=1}^{K}\mathbb{E}[\|\nabla_2 g^{(k)}(x_0, y_0)\|^2]
$$

$$
+ 18(1 + \frac{1}{\rho^2})\frac{\sigma^2}{S}\ ,
\tag{122}
$$

and

$$
\frac{1}{K}\mathbb{E}[\|U_{z,0} - \bar{U}_{z,0}\|_F^2] \leq 18\frac{1}{\rho^2}\frac{1}{K}\sum_{k=1}^{K}\mathbb{E}[\|\nabla_2 g^{(k)}(x_0, z_0)\|^2] + 18\frac{1}{\rho^2}\frac{\sigma^2}{S}\ .
\tag{123}
$$

Then, we have

$$
P_0 \leq \mathbb{E}[\mathcal{L}(x_0)] + \frac{1}{10}\frac{1}{\rho}\mathbb{E}[h_\rho(x_0, y_0) - \hat{h}_\rho(x_0)] + \frac{1}{10}\frac{1}{\rho}\mathbb{E}[g(x_0, z_0) - \hat{g}(x_0)]
$$

$$
+ 18\eta_x\frac{1}{K}\sum_{k=1}^{K}\mathbb{E}[\|\nabla_1 f^{(k)}(x_0, y_0)\|^2] + 18\eta_x\frac{1}{\rho^2}\frac{1}{K}\sum_{k=1}^{K}\mathbb{E}[\|\nabla_1 g^{(k)}(x_0, y_0)\|^2]
$$

$$
+ 18\eta_x\frac{1}{\rho^2}\frac{1}{K}\sum_{k=1}^{K}\mathbb{E}[\|\nabla_1 g^{(k)}(x_0, z_0)\|^2] + 18\eta_y\frac{1}{K}\sum_{k=1}^{K}\mathbb{E}[\|\nabla_2 f^{(k)}(x_0, y_0)\|^2]
$$

$$
+ 18\eta_y\frac{1}{\rho^2}\frac{1}{K}\sum_{k=1}^{K}\mathbb{E}[\|\nabla_2 g^{(k)}(x_0, y_0)\|^2] + 18\eta_z\frac{1}{\rho^2}\frac{1}{K}\sum_{k=1}^{K}\mathbb{E}[\|\nabla_2 g^{(k)}(x_0, z_0)\|^2]
$$

$$
+ \frac{9}{\gamma_x\eta_x}\frac{\sigma^2}{S}(1 + \frac{2}{\rho^2}) + \frac{144(\rho^2 L_f^2 + L_g^2)}{\mu^2}\frac{\eta_x}{\gamma_y\eta_y^2}(1 + \frac{1}{\rho^2})\frac{\sigma^2}{S} + \frac{72L_g^2}{\mu^2}\frac{\eta_x}{\gamma_z\eta_z^2}\frac{1}{\rho^2}\frac{\sigma^2}{S}
$$

$$
+ 27\eta_x\frac{\sigma^2}{S}(1 + \frac{2}{\rho^2}) + 27\eta_y\frac{\sigma^2}{S}(1 + \frac{1}{\rho^2}) + 27\eta_z\frac{1}{\rho^2}\frac{\sigma^2}{S}\ .
\tag{124}
$$

As a result, we can obtain

$$
\frac{1}{T}\sum_{t=0}^{T-1}\mathbb{E}[\|\nabla\mathcal{L}(\bar{x}_t)\|^2]
$$

$$
\leq \frac{2\mathcal{L}(x_0)}{\eta_x T} + \frac{2}{\eta_x T}\frac{1}{10}\frac{1}{\rho}\mathbb{E}[h_\rho(x_0, y_0) - \hat{h}_\rho(x_0)] + \frac{2}{\eta_x T}\frac{1}{10}\frac{1}{\rho}\mathbb{E}[g(x_0, z_0) - \hat{g}(x_0)]
$$

$$
+ 3\frac{1}{T}\sum_{t=0}^{T-1}\mathbb{E}[\|\nabla\mathcal{L}(\bar{x}_t) - \nabla\mathcal{L}_\rho(\bar{x}_t)\|^2] + \frac{36}{T}\frac{1}{K}\sum_{k=1}^{K}\mathbb{E}[\|\nabla_1 f^{(k)}(x_0, y_0)\|^2]
$$

$$+ \frac{36}{T} \frac{1}{\rho^2} \frac{1}{K} \sum_{k=1}^{K} \mathbb{E}[\|\nabla_1 g^{(k)}(x_0, y_0)\|^2] + \frac{36}{T} \frac{1}{\rho^2} \frac{1}{K} \sum_{k=1}^{K} \mathbb{E}[\|\nabla_1 g^{(k)}(x_0, z_0)\|^2]$$

$$+ \frac{36\eta_y}{\eta_x T} \frac{1}{K} \sum_{k=1}^{K} \mathbb{E}[\|\nabla_2 f^{(k)}(x_0, y_0)\|^2] + \frac{36\eta_y}{\eta_x T} \frac{1}{\rho^2} \frac{1}{K} \sum_{k=1}^{K} \mathbb{E}[\|\nabla_2 g^{(k)}(x_0, y_0)\|^2]$$

$$+ \frac{36\eta_z}{\eta_x T} \frac{1}{\rho^2} \frac{1}{K} \sum_{k=1}^{K} \mathbb{E}[\|\nabla_2 g^{(k)}(x_0, z_0)\|^2]$$

$$+ \frac{18}{\gamma_x \eta_x^2 T} \frac{\sigma^2}{S} (1 + \frac{2}{\rho^2}) + \frac{288(\rho^2 L_f^2 + L_g^2)}{\mu^2} \frac{1}{\gamma_y \eta_y^2 T} (1 + \frac{1}{\rho^2}) \frac{\sigma^2}{S} + \frac{144 L_g^2}{\mu^2} \frac{1}{\gamma_z \eta_z^2 T} \frac{1}{\rho^2} \frac{\sigma^2}{S}$$

$$+ \frac{54}{T} \frac{\sigma^2}{S} (1 + \frac{2}{\rho^2}) + \frac{54\eta_y}{\eta_x T} \frac{\sigma^2}{S} (1 + \frac{1}{\rho^2}) + \frac{54\eta_z}{\eta_x T} \frac{1}{\rho^2} \frac{\sigma^2}{S}$$

$$+ 18(1 + \frac{2}{\rho^2}) \gamma_x^2 \eta_x^4 \sigma^2 + 12(1 + \frac{1}{\rho^2}) \frac{\eta_y}{\eta_x} \gamma_y^2 \eta_y^4 \sigma^2 + 6 \frac{1}{\rho^2} \frac{\eta_z}{\eta_x} \gamma_z^2 \eta_z^4 \sigma^2$$

$$+ 36(1 + \frac{2}{\rho^2}) \gamma_x \eta_x^2 \sigma^2 \frac{1}{K} + \frac{576(\rho^2 L_f^2 + L_g^2)}{\mu^2} (1 + \frac{1}{\rho^2}) \gamma_y \eta_y^2 \sigma^2 \frac{1}{K} + \frac{288 L_g^2}{\mu^2} \frac{1}{\rho^2} \gamma_z \eta_z^2 \sigma^2 \frac{1}{K}$$

$$+ 36(1 + \frac{2}{\rho^2}) \gamma_x^2 \eta_x^4 \sigma^2 + 36(1 + \frac{1}{\rho^2}) \frac{\eta_y}{\eta_x} \gamma_y^2 \eta_y^4 \sigma^2 + 36 \frac{1}{\rho^2} \frac{\eta_z}{\eta_x} \gamma_z^2 \eta_z^4 \sigma^2 . \tag{125}$$

Then, due to

$$c_0 = \frac{36(\rho^2 L_f^2 + L_g^2)}{\mu^2} = O(\kappa^2) , \quad c_1 = \frac{36 L_g^2}{\mu^2} = O(\kappa^2) ,$$

$$\frac{c_0 \eta_x}{\eta_y} = \frac{1}{10} , \frac{c_1 \eta_x}{\eta_z} = \frac{1}{10} , \tag{126}$$

we can know that $\eta_y = O(\kappa^2)\eta_x$ and $\eta_z = O(\kappa^2)\eta_x$. Then, we can denote the convergence upper bound as follows:

$$\frac{1}{T} \sum_{t=0}^{T-1} \mathbb{E}[\|\nabla \mathcal{L}(\bar{x}_t)\|^2]$$

$$\leq O\left(\frac{\mathcal{L}(x_0)}{\eta_x T}\right) + 3 \frac{1}{T} \sum_{t=0}^{T-1} \mathbb{E}[\|\nabla \mathcal{L}(\bar{x}_t) - \nabla \mathcal{L}_\rho(\bar{x}_t)\|^2]$$

$$+ O\left(\frac{1}{\eta_x T} \frac{1}{\rho}\right) \mathbb{E}[h_\rho(x_0, y_0) - \hat{h}_\rho(x_0)] + O\left(\frac{1}{\eta_x T} \frac{1}{\rho}\right) \mathbb{E}[g(x_0, z_0) - \hat{g}(x_0)]$$

$$+ O\left(\frac{1}{T}\right) \frac{1}{K} \sum_{k=1}^{K} \mathbb{E}[\|\nabla_1 f^{(k)}(x_0, y_0)\|^2] + O\left(\frac{1}{T} \frac{1}{\rho^2}\right) \frac{1}{K} \sum_{k=1}^{K} \mathbb{E}[\|\nabla_1 g^{(k)}(x_0, y_0)\|^2]$$

$$+ O\left(\frac{1}{T} \frac{1}{\rho^2}\right) \frac{1}{K} \sum_{k=1}^{K} \mathbb{E}[\|\nabla_1 g^{(k)}(x_0, z_0)\|^2] + O\left(\frac{\eta_y}{\eta_x T}\right) \frac{1}{K} \sum_{k=1}^{K} \mathbb{E}[\|\nabla_2 f^{(k)}(x_0, y_0)\|^2]$$

$$+ O\left(\frac{\eta_y}{\eta_x T} \frac{1}{\rho^2}\right) \frac{1}{K} \sum_{k=1}^{K} \mathbb{E}[\|\nabla_2 g^{(k)}(x_0, y_0)\|^2] + O\left(\frac{\eta_z}{\eta_x T} \frac{1}{\rho^2}\right) \frac{1}{K} \sum_{k=1}^{K} \mathbb{E}[\|\nabla_2 g^{(k)}(x_0, z_0)\|^2]$$

$$+ O\left(\frac{1}{\gamma_x \eta_x^2 T} \frac{1}{\rho^2} \frac{\sigma^2}{S}\right) + O\left(\frac{1}{\gamma_y \eta_y^2 T} \frac{\kappa^2}{\rho^2} \frac{\sigma^2}{S}\right) + O\left(\frac{1}{\gamma_z \eta_z^2 T} \frac{\kappa^2}{\rho^2} \frac{\sigma^2}{S}\right)$$

$$+ O\left(\frac{\sigma^2}{ST} \frac{1}{\rho^2}\right) + O\left(\frac{\eta_y}{\eta_x} \frac{\sigma^2}{ST} \frac{1}{\rho^2}\right) + O\left(\frac{\eta_z}{\eta_x} \frac{\sigma^2}{ST} \frac{1}{\rho^2}\right)$$

$$+ O\left(\frac{\gamma_x^2 \eta_x^4 \sigma^2}{\rho^2}\right) + O\left(\frac{1}{\rho^2} \frac{\eta_y}{\eta_x} \gamma_y^2 \eta_y^4 \sigma^2\right) + O\left(\frac{1}{\rho^2} \frac{\eta_z}{\eta_x} \gamma_z^2 \eta_z^4 \sigma^2\right)$$

$$+ O\left(\frac{\gamma_x \eta_x^2 \sigma^2}{\rho^2} \frac{1}{K}\right) + O\left(\frac{\kappa^2}{\rho^2} \gamma_y \eta_y^2 \sigma^2 \frac{1}{K}\right) + O\left(\frac{\kappa^2}{\rho^2} \gamma_z \eta_z^2 \sigma^2 \frac{1}{K}\right)$$

$$\leq O\left(\frac{\mathcal{L}(x_0)}{\eta_x T}\right) + 3\frac{1}{T}\sum_{t=0}^{T-1} \mathbb{E}[\|\nabla\mathcal{L}(\bar{x}_t) - \nabla\mathcal{L}_\rho(\bar{x}_t)\|^2]$$

$$+ O\left(\frac{1}{\eta_x T}\frac{1}{\rho}\right)\mathbb{E}[h_\rho(x_0, y_0) - \hat{h}_\rho(x_0)] + O\left(\frac{1}{\eta_x T}\frac{1}{\rho}\right)\mathbb{E}[g(x_0, z_0) - \hat{g}(x_0)]$$

$$+ O\left(\frac{1}{T}\right)\frac{1}{K}\sum_{k=1}^{K}\mathbb{E}[\|\nabla_1 f^{(k)}(x_0, y_0)\|^2] + O\left(\frac{1}{T}\frac{1}{\rho^2}\right)\frac{1}{K}\sum_{k=1}^{K}\mathbb{E}[\|\nabla_1 g^{(k)}(x_0, y_0)\|^2]$$

$$+ O\left(\frac{1}{T}\frac{1}{\rho^2}\right)\frac{1}{K}\sum_{k=1}^{K}\mathbb{E}[\|\nabla_1 g^{(k)}(x_0, z_0)\|^2] + O\left(\frac{\kappa^2}{T}\right)\frac{1}{K}\sum_{k=1}^{K}\mathbb{E}[\|\nabla_2 f^{(k)}(x_0, y_0)\|^2]$$

$$+ O\left(\frac{\kappa^2}{T}\frac{1}{\rho^2}\right)\frac{1}{K}\sum_{k=1}^{K}\mathbb{E}[\|\nabla_2 g^{(k)}(x_0, y_0)\|^2] + O\left(\frac{\kappa^2}{T}\frac{1}{\rho^2}\right)\frac{1}{K}\sum_{k=1}^{K}\mathbb{E}[\|\nabla_2 g^{(k)}(x_0, z_0)\|^2]$$

$$+ O\left(\frac{1}{\gamma_x \eta_x^2 T}\frac{1}{\rho^2}\frac{\sigma^2}{S}\right) + O\left(\frac{1}{\gamma_y \kappa^2 \eta_x^2 T}\frac{1}{\rho^2}\frac{\sigma^2}{S}\right) + O\left(\frac{1}{\gamma_z \kappa^2 \eta_x^2 T}\frac{1}{\rho^2}\frac{\sigma^2}{S}\right)$$

$$+ O\left(\frac{\sigma^2}{ST}\frac{1}{\rho^2}\right) + O\left(\frac{\sigma^2}{ST}\frac{\kappa^2}{\rho^2}\right) + O\left(\frac{\sigma^2}{ST}\frac{\kappa^2}{\rho^2}\right)$$

$$+ O\left(\frac{\gamma_x^2 \eta_x^4 \sigma^2}{\rho^2}\right) + O\left(\frac{\kappa^{10}}{\rho^2}\gamma_y^2 \eta_x^4 \sigma^2\right) + O\left(\frac{\kappa^{10}}{\rho^2}\gamma_z^2 \eta_x^4 \sigma^2\right)$$

$$+ O\left(\frac{\gamma_x \eta_x^2 \sigma^2}{\rho^2}\frac{1}{K}\right) + O\left(\frac{\kappa^6}{\rho^2}\gamma_y \eta_x^2 \sigma^2 \frac{1}{K}\right) + O\left(\frac{\kappa^6}{\rho^2}\gamma_z \eta_x^2 \sigma^2 \frac{1}{K}\right) . \tag{127}$$

From Eq. (113), we can set

$$\gamma_x = O\left(\frac{\kappa^2}{(1-\lambda)^4 \rho^2 K}\right), \quad \gamma_y = O\left(\frac{1}{(1-\lambda)^4 \rho^2 K}\right), \quad \gamma_z = O\left(\frac{1}{(1-\lambda)^4 \rho^2 K}\right) . \tag{128}$$

To achieve the $\epsilon$-accuracy solution: $\frac{1}{T}\sum_{t=0}^{T-1}\mathbb{E}[\|\nabla\mathcal{L}(\bar{x}_t)\|^2] \leq \epsilon^2$, we need to set

$$\rho = O\left(\frac{\epsilon}{\kappa^3}\right) \tag{129}$$

such that $\mathbb{E}[\|\nabla\mathcal{L}(\bar{x}_t) - \nabla\mathcal{L}_\rho(\bar{x}_t)\|^2] \leq O(\epsilon^2)$ in terms of Lemma A.16. In addition, by initializing $y_0$ and $z_0$ such that $\mathbb{E}[h_\rho(x_0, y_0) - \hat{h}_\rho(x_0)] \leq O(\epsilon/\kappa^3)$ and $\mathbb{E}[g(x_0, z_0) - \hat{g}(x_0)] \leq O(\epsilon/\kappa^3)$. Note that both $h_\rho(x, y)$ and $g(x, y)$ are $\mu$-PL in $y$ and their smoothness constants are not dominated by $\rho$. Therefore, we can use a stochastic gradient descent algorithm to complete the initialization. The complexity for initialization is of the order $O(\frac{\kappa^5}{\epsilon}\log\frac{\kappa^3}{\epsilon})$ (Garrigos & Gower, 2023), which does not dominate the following iteration complexity $T$.

Moreover, we set

$$\eta_x = O\left(\frac{K(1-\lambda)^2 \epsilon^3}{\kappa^9}\right), \quad \eta_y = O\left(\frac{K(1-\lambda)^2 \epsilon^3}{\kappa^7}\right), \quad \eta_z = O\left(\frac{K(1-\lambda)^2 \epsilon^3}{\kappa^7}\right),$$

$$S = O\left(\frac{\kappa^7}{\epsilon^3}\right), \quad T = O\left(\frac{\kappa^9}{K(1-\lambda)^2 \epsilon^5}\right) . \tag{130}$$

Then, we have

$$\frac{1}{T}\sum_{t=0}^{T-1}\mathbb{E}[\|\nabla\mathcal{L}(\bar{x}_t)\|^2] \leq O(\epsilon^2) . \tag{131}$$

$\square$

## A.9. Foundations of Problem Reformulation

**Lemma A.14.** *(Chen et al., 2024) Suppose Assumptions 4.1-4.5 hold, denoting $Y_\rho^*(x) = \arg\min_{y \in \mathbb{R}^{d_y}} h_\rho(x, y)$, we have that*

$$dist(Y_\rho^*(x_1), Y_\rho^*(x_2)) \leq C_{y_\rho^*} \|x_1 - x_2\| , \tag{132}$$

*for any $x_1 \in \mathbb{R}^{d_x}$ and $x_2 \in \mathbb{R}^{d_x}$, where $C_{y_\rho^*} = \frac{\rho L_f + L_g}{\mu} = O(\kappa)$.*

**Lemma A.15.** *(Chen et al., 2024; Shen & Chen, 2023) Suppose Assumptions 4.1-4.5 hold, we have that*

$$\nabla \mathcal{L}_\rho(x) = \nabla_1 f(x, y_\rho^*(x)) + \frac{1}{\rho}(\nabla_1 g(x, y_\rho^*(x)) - \nabla_1 g(x, y^*(x))) , \tag{133}$$

*where $y_\rho^*(x) \in Y_\rho^*(x)$ and $y^*(x) \in Y^*(x)$.*

**Lemma A.16.** *(Chen et al., 2024; Kwon et al., 2024a) Suppose Assumptions 4.1-4.5 hold, $\nabla \mathcal{L}(x)$ exists and can be obtained by $\lim_{\rho \to 0+} \nabla \mathcal{L}_\rho(x)$. Moreover, for any $0 \leq \rho \min\{\ell_g/\ell_f, \bar{\rho}\}$, where $\bar{\rho}$ is a given constant, we have that*

$$|\mathcal{L}_\rho(x) - \mathcal{L}(x)| \leq O(\rho \ell \kappa) ,$$
$$\|\nabla \mathcal{L}_\rho(x) - \nabla \mathcal{L}(x)\| \leq O(\rho \ell \kappa^3) . \tag{134}$$

