# OpenReview forum: "Convergence Analysis of Decentralized Hessian-/Jacobian-Free Algorithm for Nonconvex Stochastic Bilevel Optimization"
_ICML.cc/2026/Conference — ICML 2026 regular_

### Official Review · Reviewer_YGNZ · 2026-02-15

**Soundness:** 3
**Presentation:** 3
**Significance:** 2
**Originality:** 2
**Overall Recommendation:** 4
**Confidence:** 4

**Summary:**

The work studies stochastic bi-level optimization in the decentralized setting, assuming that lower-level problems individually satisfy the PL condition. The algorithm introduced and analytical tools can be viewed as a combination of 1) the penalty method introduced in Kwon et al. [2023] for estimating hyper-gradients without Hessian/Jacobian queries, 2) STOchastic Recursive
Momentum (STORM) for gradient estimator variance reduction, and 3) standard tools for decentralized optimization. The authors establish a sample complexity of $\Ocal(\epsilon^{-5})$, which matches the result in Kwon et al. [2023] in the centralized setting. Illustrative experiments are conducted that validate the empirical performance of the proposed algorithm.

**Compliance With Llm Reviewing Policy:**

Affirmed.

**Final Justification:**

The authors’ clarification regarding the distinction from prior work addresses my biggest concern, and I would therefore like to raise my evaluation to an acceptance score.

**Key Questions For Authors:**

Besides main concern above on the lack of technical novelty, I find certain statements in the introduction section to be unclear, ungrounded, or exaggerated.

1) Page 2 left column "the PL condition is easy to be satisfied by many machine learning models".

While the PL condition is weaker than strong convexity, it is certainly not "easy to be satisfied" in practice. It is unrealistic to expect the PL condition to rise naturally, unless regularization is added (e.g. in the context of RL) or inconvenient assumptions are imposed when solving specific problems (e.g. a sufficiently overparameterized neural network is used to approximate the solution to a least squares problem, in the paper referenced by the authors).

2) Page 2 right column "In other words, it remains unclear whether the stochastic first-order algorithm can converge under the pure PL condition."

As said above in Weaknesses section, stochastic first-order algorithms under lower-level PL condition has been studied in many prior works. While we do not know if the complexities established so far are optimal, it feels misleading to say it is unclear whether such algorithms converge at all.

3) Page 2 right column "unlike the existing single-agent method (Kwon et al., 2023), our algorithm does not introduce a regularization term to enforce artificial strong convexity".

 I do not think this is a very fair statement. It boils down to what assumptions are being made. The point of the paper is to study bi-level optimization under the PL condition.

4) Page 2 right column "may leak training stage information to malicious workers in the decentralized setting".

Can the authors explain why training stage information is leaked just because of the choice of the step size?

5) Page 3 left column "we introduce a novel approach to quantify how close the current iterate of the lower-level variable is to its optimal value at the t-th iteration."

Can the authors elaborate what exactly is novel, especially in comparison to prior works like Chen et al. [2024], Gaur et al. [2025], Zeng et al. [2025]?

Minor: The paper seems to cite Kwon et al. [2023] as the first work developing penalty methods for bi-level optimization. However, this particular paper extends the techniques developed by the same group of authors in an earlier work (reference below). While focused on problems with lower-level strong convexity, this earlier work should be the more appropriate reference for the original development of the penalty methods.

Kwon, J., Kwon, D., Wright, S. and Nowak, R.D., 2023, July. A fully first-order method for stochastic bilevel optimization. In International Conference on Machine Learning (pp. 18083-18113). PMLR.

**Limitations:**

I do not have concerns regarding potential negative societal impact.

**Strengths And Weaknesses:**

In general I find the paper well written, though the mathematical density and lack of step-by-step explanation on algorithmic development can make the paper challenging to read for an audience not already familiar with the penalty method for bi-level optimization and STORM. The results established in the work are new to my knowledge, at the particular intersection of stochastic bi-level Hessian/Jacobian-free optimization and decentralized optimization.

On the negative side, I find the paper lacking in actual technical innovation. As said above, the algorithm and analysis are heavily built on penalty method for bi-level optimization, STORM, and standard decentralized optimization tools. While this paper may be the first to combine the three tool sets and that may require non-trivial efforts, I believe there have been many works that have combined two out of the three (e.g. Kwon et al. [2023] in the development of the single-loop momentum algorithm), making the paper less interesting from a technical perspective.

I also do not see the claimed innovation from handling the PL condition. In the centralized setting, Chen et al. [2024], which the authors have referenced, and Gaur et al. [2025] both have studied how to deal with the PL condition at the lower level. The takeaway from these papers is that the complexity of penalty method under lower-level PL condition can be shown to match that under lower-level strong convexity. I understand that the present work focuses on a single-loop algorithm, which may complicate the treat of lower-level variable under the PL condition. However, Zeng et al. [2025] also analyzes a single-loop algorithm under lower-level PL condition, and they also establish a complexity bound on the same order as if the lower-level problem is strongly convex.

References

Gaur, M., Singh, U., Bedi, A.S., Pasupathu, R. and Aggarwal, V., 2025. On The Sample Complexity Bounds In Bilevel Reinforcement Learning. arXiv preprint arXiv:2503.17644.

Zeng, S., Bhatt, S., Ganesh, S. and Koppel, A., A Regularized Actor-Critic Algorithm for Bi-Level Reinforcement Learning. In NeurIPS 2025 Workshop: Second Workshop on Aligning Reinforcement Learning Experimentalists and Theorists.


**Post-Rebuttal:**

Adjusted score from weak reject to weak accept, as the authors’ clarification regarding the distinction from prior work addressed my main concern.

---

> ### Author Rebuttal · Authors · 2026-03-27
>
> Thank you for your comments. We provide a point-by-point response below.
>
> < To Weaknesses > :   The reviewer's concern mainly lies in our novelty compared to Kwon et al. [2023], Chen et al. [2024], Gaur et al. [2025], and Zeng et al. [2025].  The main difference from these existing works are shown below.
>
> * _Comparison with Kwon et al. [2023]. Even though this existing work assumes the PL condition for the lower-level loss function, it introduces a regularization term such that the loss function becomes strongly convex. Then, in their proof (See their  Lemma D.3), **they actually use the strong convexity to complete the proof**. In addition, they bound the optimization error $||y\_{t} -y^{\*}\_{\rho}(x)||^2$ and $||z\_{t} - y^{\*}(x)||^2$. On the contrary, we just use the pure PL condition and bound the optimization error in our Eq. (8).
>
> * _Comparison with Chen et al. [2024]_.  This existing work uses the standard **stochastic gradient** to update lower-level variables. Therefore, it is easy to bound the optimization error $h(x)-h^{\*}$ (See their Lemma H.1) for the **double-loop** update,  as **the gradient variance $\sigma$ is independent of the update**.  On the contrary, we use the **stochastic variance-reduced gradient**, which cannot naturally be applied to the double-loop scheme. Specifically, when bounding the optimization error $g(x, z)- \hat{g}^{\*}(x)$, our method needs to handle the gradient estimation error, i.e., $\|\nabla g - m\|^2$, which is not a constant as $\sigma$. Therefore, we proposed two novel Lemmas 4.10 and 4.11 to bound the optimization error in Eq. (8).
>
> * _Comparison with Gaur et al. [2025]_.  This existing work assumes **bounded gradient** (See their Assumptions 1 and 3) and uses the **double-loop** update. This simplified the setting too much.
>
> * _Comparison with Zeng et al. [2025]_. This existing work assumes **bounded $\textcolor{blue}{stochastic}$ gradients** (See their Assumption 3), which is not used in Kwon et al. [2023], Chen et al. [2024], and our work.  This is a quite strong assumption. As a result, **Zeng et al. [2025] does NOT need to handle the complicated variance issue**. On the contrary, **our method proposed novel strategies to handle the complicated variance when designing the potential function**. Specifically, in our potential function in Page 13, the optimization error in the second and third terms explicitly involves $1/\rho$, the gradient estimation error regarding the lower-level function in the 9-th and 10-th terms are also  explicitly involves $1/\rho^2$, while that regarding the upper-level function in the 8-th term are NOT involve $1/\rho^2$. These novel designs guarantee that our algorithm can achieve the $1/\epsilon^5$ convergence rate.
>
> **In summary, compared to existing methods, our proof is novel.** We will provide detailed discussion about these existing works to better demonstrate the novelty and contribution of our paper in the next version.
>
> < To Questions 1 > : We will revise the tone of our writing. However, we would like to emphasize that existing studies, such as Liu et al. (2022a), show that the optimization of overparameterized deep neural networks satisfies the PL condition. Therefore, in our experiments—such as hyperparameter optimization in Figure 1 and model pruning in Figure 3—the lower-level optimization involves a deep neural network, which serves as a representative example of the PL condition.
>
> < To Questions 2 > : As discussed in our response to Weaknesses, under the same assumptions as our work and Kwon et al. [2023], Chen et al. [2024], no existing works have shown that the single-loop algorithm can guarantee convergence. We will cite Zeng et al. [2025]  and revise the tone in the next version.
>
> < To Questions 3 > : Kwon et al. [2023] uses the PL assumption, but their proof relies on strong convexity by introducing the regularization term. Please see their  Lemma D.3.
>
> < To Questions 4 > : For example, if the learning rate is set $1/t^{2/5}$. It is easy to infer the current iteration $t$ for a newly joined worker. The constant learning rate does not have this issue.
>
> < To Questions 5 > :  Please see our response to Weaknesses.

---

> > ### Author Rebuttal · Reviewer_YGNZ · 2026-04-01
> >
> > The authors’ clarification regarding the distinction from prior work addresses my biggest concern, and I would therefore like to raise my evaluation to an acceptance score.
> >
> > Regarding the response to my first question, I still believe the claim should be somewhat toned down. While the PL condition is much milder than strong convexity, it almost never hold exactly in practical problems. The works which show that certain learning objectives under an over-parameterized NN satisfy the PL condition assume that the NN is unpractically large. That represents a pretty large gap to reality.

---

> > > ### Author Response · Authors · 2026-04-01
> > >
> > > We sincerely appreciate you increasing the evaluation to an acceptance level. We will revise and improve our writing based on your suggestions.

---

### Official Review · Reviewer_U2v2 · 2026-03-05

**Soundness:** 1
**Presentation:** 1
**Significance:** 2
**Originality:** 1
**Overall Recommendation:** 2
**Confidence:** 3

**Summary:**

This work developed a novel decentralized stochastic first-order optimization algorithm to solve the issue in decentralized stochastic bi-level optimization. Particularly, the resulting algorithm did not require second-order Hessian or Jacobian matrices, for the setting where the lower-level loss function was nonconvex but satisfied the Polyak-{\L}ojasiewicz (P{\L}) condition. Moreover, the authors proved the convergence behavior of the resulting algorithms. Experimental results showed the efficacy of the resulting algorithm.

**Compliance With Llm Reviewing Policy:**

Affirmed.

**Key Questions For Authors:**

See Weakness

**Limitations:**

More theoretical results about the existing studies should be discussed.

**Strengths And Weaknesses:**

Strength:
This article developed and analyzed a novel decentralized stochastic first-order optimization algorithm that did not compute second-order Hessian or Jacobian matrices. The author analyzed the convergence behavior of the resulting algorithm for the P{\L} assumption. The article performed numerical experiments to confirm the efficacy of the resulting algorithm.

Weaknesses:

The contribution of this article is quite limited. First, the P{\L} assumption can be as a special case of convex optimization and is not difficult to satisfy in practice. Therefore, the author claimed that they proved the convergence behavior of the resulting algorithm under nonconvex, which is not very important. Therefore, the theoretical results obtained by this work can be viewed as the extension of the existing studies.

The article cited large numbers of studies, but the author did not discuss these works in detail. Therefore, it is hard to understand the results provided by this work. More specifically, many studies have been accepted by conferences or journals. However, the author only cited the archive version, which is not appropriate.

The resulting algorithm is just an extension of STORM, leading to the contribution of this article being quite limited. In addition, the numerical experiments are far away enough to support the theoretical findings of the authors’ claim.

---

> ### Author Rebuttal · Authors · 2026-03-27
>
> Thank you for your comments. We provide a point-by-point response below.
>
> < To Weaknesses  1> :  We respectfully disagree with the reviewer's argument.
>
> * First, the PL condition is weaker than strong convexity. Therefore, it requires new efforts to establish convergence rates under the PL condition rather than strong convexity. This is why many recent works have focused on addressing this challenge over the past few years, leading to numerous state-of-the-art methods published at top conferences such as ICML [1], ICLR [2], COLT [3], and AISTATS [4]. If it were not important, there would not be so many high-quality publications.
>
> * Second, our paper is not a straightforward extension of existing works. In fact, we propose a new proof strategy to establish our convergence rate, as clearly stated in Section 4.3. We sincerely hope the reviewer will carefully examine that section to understand our key contributions.
>
>
> [1] Han Shen et al., On Penalty-based Bilevel Gradient Descent Method, **ICML 2023**.
>
> [2] Jeongyeol Kwon et al., On Penalty Methods for Nonconvex Bilevel Optimization and First-Order Stochastic Approximation, **ICLR 2024**.
>
> [3] Lesi Chen et al., On Finding Small Hyper-Gradients in Bilevel Optimization: Hardness Results and Improved Analysis, **COLT 2024**.
>
> [4] Sihan Zeng et al., A Hessian-Free Actor-Critic Algorithm for Bi-Level Reinforcement Learning with Applications to LLM Fine-Tuning, **AISTATS 2026**.
>
> < To Weaknesses  2> :  We will provide more detailed discussions about related works in the next version. However, we would like to emphasize that we have already provided detailed discussions about the more related works, such as [2]. This discussion is sufficient to demonstrate the current development of this area and the contributions of our work.
>
> < To Weaknesses  3> :  We respectfully disagree with the reviewer's argument.  Our method is not an extension of STORM. Instead, **we developed novel proof techniques, which has been stated in Section 4.3**.
>
> In addition, we have conducted extensive experiments to evaluate the performance of our algorithm. In particular, we test our algorithm on various applications, including hyperparameter optimization (Figure 1), model pruning (Figure 3), and toy examples (Figure 4). All experiments confirm its efficacy. We also provide ablation studies on different hyperparameters (Figure 2), which offer guidance for tuning them in practice.

---

> > ### Author Rebuttal · Reviewer_U2v2 · 2026-04-08
> >
> > Weaknesses
> > 2.5. The O(κ^9) condition number dependency is significantly higher than related single-agent works, lacking discussion on complexity optimization. (Thm. 4.6)
> > 2.6. Only compares with single-agent F²BSA variants; no comparison with recent decentralized nonconvex bi-level methods. (Sec. 5.1)
> > 2.7. No code repository, incomplete hyperparameter sensitivity details, and unspecified network architectures for model pruning experiments. (Sec. 5)
> > 2.8. The lengthy proof lacks step-by-step intuition, making key design choices of the potential function hard to follow. (Appendix A)
> >
> > 3. Key Questions for Authors
> > Mandatory Core Questions
> > 3.1. Please analyze the source of the O(κ^9) complexity dependency and discuss whether it can be reduced in future work (Sec. 4.2, Thm. 4.6).
> > 3.2. Add comparisons with the decentralized heavy-tailed nonconvex bi-level method on standard benchmarks (Sec. 5.1).
> > 3.3. Release full experimental code, including network architectures, initialization strategies, and complete hyperparameter settings to ensure reproducibility (Sec. 5).
> > 3.4. Clarify how the constant single-timescale learning rate ensures convergence in decentralized settings with consensus errors (Sec. 3.3, 4).

---

### Official Review · Reviewer_pSgN · 2026-03-12

**Soundness:** 3
**Presentation:** 2
**Significance:** 3
**Originality:** 2
**Overall Recommendation:** 3
**Confidence:** 3

**Summary:**

The paper studies decentralized stochastic bilevel optimization.  Unlike most decentralized bilevel works that assume the lower-level objective is strongly convex in the lower-level variable, this work targets the case where the lower-level objective is nonconvex but satisfies a Polyak–Łojasiewicz (PL) condition. The authors reformulate the bilevel problem into a single-level minimax problem via a penalty construction, and propose a decentralized first-order algorithm (Algorithm 1; FO-DSVRBGD) that combines (i) STORM-style variance-reduced gradient estimation and (ii) gradient tracking/consensus steps. The paper claims constant single-timescale learning rates (chosen as constants depending on target accuracy) and provides convergence guarantees under PL-type assumptions. Experiments on hyperparameter optimization (main body) and model pruning (appendix) compare against F2BSA variants and report faster decrease of the upper-level loss / gradient norm.

**Compliance With Llm Reviewing Policy:**

Affirmed.

**Key Questions For Authors:**

1. Under what verifiable conditions does hρ(x,·)=ρ f(x,·)+g(x,·) satisfy the PL condition if g(x,·) is PL? Is there a lemma establishing PL stability for sufficiently small ρ, and how does that interact with your chosen ρ=O(ε/κ³)?

2. In practice, κ and (1−λ) are unknown. What tuning strategy is used in experiments for ηx,ηy,ηz,γx,γy,γz,ρ? How sensitive is performance to these choices beyond the limited sweeps shown?

3. Are all methods run under the same communication topology and communication budget (ring vs all-reduce)？

**Limitations:**

The paper should explicitly discuss:

 (i) under what circumstances the PL condition holds.

(ii) under what conditions $h_{\rho}$ in Assumption 4.3 satisfies the PL condition.

(iii) Could you please write out the explicit constants hidden in the $O(\cdot)$ notation for $\eta$ given in Theorem 4.6?

**Strengths And Weaknesses:**

Strengths

1. Removing the strong-convexity requirement on the lower-level problem (replacing it with PL) is a meaningful broadening of applicability for decentralized bilevel optimization, especially for over-parameterized models where PL-type properties are sometimes argued.

2. The penalty-based reformulation avoids explicit Hessian-/Jacobian-based hypergradient computations, aligning with the “Hessian-/Jacobian-free” claim at the algorithmic level. The resulting algorithm is clean: local gradient estimation + gradient tracking + consensus mixing.

3.  The main plots suggest the proposed method improves optimization speed vs. the chosen baselines on several datasets.

Weaknesses

1. The practical relevance of PL in the stated applications is plausible but not fully substantiated in the paper (beyond citation). For a theory paper, this is acceptable if stated as a modeling assumption, but it does affect how broadly the result can be applied.

2.  Practicality of step-size/parameter prescriptions: Although the algorithm uses “constant” steps, the theorem sets ηx,ηy,ηz as functions of ε, κ, (1−λ), and K. This is standard in theory, but it limits direct usability

---

> ### Author Rebuttal · Authors · 2026-03-27
>
> Thank you for your comments. We provide a point-by-point response below.
>
> <To Weaknesses 1>: Existing studies, such as Liu et al. (2022a), show that the optimization of overparameterized deep neural networks satisfies the PL condition. Therefore, in our experiments, such as hyperparameter optimization in Figure 1 and model pruning in Figure 3, the lower-level optimization involves a deep neural network, which serves as a representative example of the PL condition.
>
> <To Weaknesses 2>: It is worth noting that almost all optimization algorithms (except for the recently developing parameter-free direction) rely on problem-specific hyperparameters, and these hyperparameters are typically not available in practice. This is quite common. Thus, we do not consider this a weakness of our paper. If this were the case, the entire optimization field and community would be problematic. Therefore, we respectfully disagree with the reviewer’s criticism. In addition, in Figure 2, we clearly demonstrate how different hyperparameters affect convergence. This is sufficient for tuning hyperparameters in practical applications.
>
> <To Questions 1>: The assumption regarding $h\_{\rho}$ follows existing single-agent methods (Kwon et al., 2023; Chen et al., 2024a), which have been accepted at ICLR 2024 and COLT 2025. We do not introduce any stronger assumptions compared to these two state-of-the-art works. We agree that developing theoretical tools to verify this assumption remains an open problem. However, we believe a feasible approach is to verify it empirically by checking the eigenvalues of its gradient in practical applications.
>
> <To Questions 2>: The condition number and spectral gap are problem-specific and system-specific parameters, which are not available for practical applications. Therefore, in Figure 2, we demonstrate different hyperparameters affect convergence. We believe this is sufficient for tuning hyperparameters in practical applications.
>
> <To Questions 3>: Yes, we use ring graph for decentralized algorithms and complete graph for sing-agent methods.
>
> <To Limitations (i), (ii)>: Please see the above responses.
>
> <To Limitations (iii)>:  The detailed upper bound regarding $\eta_x$, $\eta_y$, and $\eta_z$ can be found in Eq.(113-115) in Appendix.

---

> > ### Author Rebuttal · Reviewer_pSgN · 2026-04-04
> >
> > Thank you for the detailed rebuttal. The response is helpful and clarifies several points. In particular, I appreciate the added discussion on why PL-type behavior may be plausible in overparameterized neuralnetwork settings, as well as the pointer to the appendix for more detailed parameter bounds.
> >
> > That said, my main technical concern remains only partially addressed. The theory relies not only on $g(x, \cdot)$ being PL, but also on the additional assumption that $h_\rho(x, \cdot)=\rho f(x, \cdot)+g(x, \cdot)$ is itself $\mu$-PL. This assumption is central to the analysis, and the rebuttal essentially acknowledges that general verifiable sufficient conditions are still open. The fact that related single-agent works make similar assumptions is useful context, but by itself it does not fully resolve the concern here, especially in the decentralized setting.
> >
> > I am somewhat less concerned than before about the fact that the theorem uses constants depending on $\kappa$ and the spectral gap, since this is common in theoretical optimization work. However, the rebuttal still does not provide a concrete tuning protocol beyond limited sensitivity plots, so I still view the practical-usability claim as somewhat weaker than the presentation suggests.
> >
> > I also continue to have a concern about the empirical comparison. The clarification that decentralized methods use a ring graph while the single-agent baselines use a complete-graph / all-reduce setup means the communication setting is not fully matched, which weakens the strength of the comparison. At minimum, this should be made fully explicit and discussed as a limitation.
> >
> > Overall, I still think the paper is interesting and technically ambitious, and I agree that the setting is relevant. However, the rebuttal does not fully resolve the central assumption issue or the experimental comparability issue. Therefore, I am maintaining my recommendation at weak reject. In a revision, a clearer discussion of when $h_\rho(x, \cdot)$ is expected to satisfy PL, together with a fully controlled communication setup in the experiments, would substantially strengthen the paper.

---

> > > ### Author Response · Authors · 2026-04-08
> > >
> > > We appreciate you acknowledge our response is helpful and clear some of your concerns.  Below, we will provide a point-by-point response for the remaining concerns.
> > >
> > > < To PL condition >: Our contribution does not lie in relaxing the PL assumption commonly adopted in existing works. Instead, we employ the same PL assumption as in (Kwon et al., 2023; Chen et al., 2024a) and develop new proof techniques to establish the convergence rate under the standard PL condition. In other words, the novelty of our work lies in these proof techniques.  We hope the reviewer will evaluate the contributions of our paper objectively and fairly.
> > >
> > > < Tuning protocol>: For most optimization algorithms, hyperparameters depend on problem-specific or system-specific parameters. This dependence is generally unavoidable unless one adopts parameter-free techniques to eliminate it. In this respect, our algorithm follows the same principles as traditional optimization methods. Therefore, it is not possible to determine those values for these hyperparameters in practical applications.
> > >
> > > In our experiments, we provide appropriate ranges for these hyperparameters and demonstrate their impact on convergence. Such empirical evaluations are standard in the optimization community and are sufficient to illustrate the performance of our algorithm.
> > >
> > > < Single-agent >: For the single-agent baselines, a complete graph should be used rather than other communication graphs. Specifically, using a non-complete graph may degrade the performance of single-agent baselines due to sparse communication, leading to an unfair comparison. Therefore, to ensure fairness, we use a complete graph for the single-agent baselines.

---

### Official Review · Reviewer_ksL7 · 2026-03-12

**Soundness:** 3
**Presentation:** 2
**Significance:** 2
**Originality:** 3
**Overall Recommendation:** 4
**Confidence:** 3

**Summary:**

his paper proposes a decentralized stochastic bi-level optimization algorithm. The key contribution here is the assumption that the lower-level function is assumed to satisfy the PL condition in $y$.
While I like this contribution, I am not sure the paper is ready to be published in ICML 2026. Here are some detailed comments and questions.  I will admit however my experience with bi-level optimization with penalty method is limited. And in general, the penaly parameter $\rho\to\infty$ if you want true convergence. This is an understandable comprise but an issue nevertheless.

**Compliance With Llm Reviewing Policy:**

Affirmed.

**Final Justification:**

The author addressed my concerns well. Although I am not capable of fully evaluating the significance of their contribution, I think it's a 4 based on my knowledge.

**Key Questions For Authors:**

Please see my comments above.

**Limitations:**

Yes I understand the limitations.

**Strengths And Weaknesses:**

1. It appears the PL assumption for lower-level problem and penalty method have been studied in (Shen and Chen, 2023), as the authors stated themselves. The key difference here is the use of stocahstic gradient, which introduces additional difficulty. But this does not seem to be a fundamentally challenging problem in terms of algorithm design as the authors presented it to be. I am also not sure how much the decentralized part add to the table.  However, I recognize the hard work the authors put in to make the analysis work, including introducing the potential function, which is often used when assuming PL/KL conditions.

2. The two contributions for algorithm design are small to me, but they are changes nevertheless. One reason is that I am not convinced of the superiortiy of single-timescale constant. Can you explain why single-timescale constant will not leak to malicious workers? I am not familiar with this aspect.

3. It will be nice to have the complexity results listed in a table and include as many bi-level theories as possible so it is easier to compare. But this is not a requirement.


4. You mentioned at line 215 the key element in your algorithm is the STORM gradient estimator, but that seems to not be your key innovations in your algorithm, right? Again, you repeated your contributions but the benefits of those two are not obvious.

5. The author use really a lot of "novel". What I think they actually meant to say is ``novel'' in the sense that it is the first time some of these techniques are applied to stochastic decentralized bi-level optimization problem. For instace, is the intermediate error measure in (8) completely new and never been adopted before? If so, please do let me know because it is contrary to my limited knowledge.


6. I don't think the summary at line 334 is needed and it is quite repetitive.

7. The quality in writing for the proofs in the appendix drop dramatically as it goes on. For instance, from (30) on, the explanations of steps are very selective and sparse, making following the proofs very difficult.

Overall the acceptance of the paper depends on whether the theoretical novelty, i.e., convergence using PL condtions on lower-level problem and stochastic gradient estimate (seems to be adapted from previous algorithms) are considered significant.  I think other reviewers might be more qualified to evaluate this specific point.

---

> ### Author Rebuttal · Authors · 2026-03-26
>
> Thank you for the comments. We provide the point-by-point response below.
>
> <To 1>: The key difference from existing works does not lie in the stochastic gradient. Instead, as clearly shown in Section 4.3, **the key difference and novelty lie in the new proof techniques**.
> * First, Shen and Chen, 2023 proposed a **double-loop** algorithm, while our algorithm is **single-loop**. Then, our proof is totally different from Shen and Chen, 2023.
> * Second, the most related to our single-loop algorithm is Algorithm 2 in Kwon et al.,2023. Compared to that algorithm, the key novelty lies in our new proof techniques. Specifically, Kwon et al., 2023 introduces a regularization term to make the loss function with respect to the variable $y$ and $z$ strongly convex.  This allows them to apply techniques developed for **strongly convex** optimization in their convergence  analysis. For example, when analyzing the optimization error for the  subproblems involving $y$ and $z$, Kwon et al., 2023 bounds the optimization errors $||y\_{t} -y^{\*}\_{\rho}(x)||^2$ and $||z\_{t} - y^{\*}(x)||^2$  using methods tailored for strongly convex objectives (see Lemma D.3 in Kwon et al., 2023). Instead, we proposed directly bounding the metrics in Eq. (8) by **only using the PL condition**.
>
> **We sincerely hope the reviewer could take a look at our Section 4.3, which clearly states the key novelty of our proof techniques.**
>
> <To 2>: For bilevel optimization problems, there are multiple variables. If appropriate learning rates are not used for these variables, this can lead to divergence. Existing methods, such as the closely related work by Kwon et al. (2023), use two-timescale learning rates. However, this type of learning rate is difficult to tune in practice, and if not properly calibrated, it can also cause divergence. In contrast, our algorithm uses a single-timescale learning rate, which is much easier to tune than two-timescale approaches.
>
> Kwon et al. (2023) uses a time-dependent learning rate, e.g., $1/t^{2/5}$. Thus, a newly joined worker can easily infer the current number of iterations $t$ according to the learning rate $1/t^{2/5}$.
>
> <To 3>: Thank you for your suggestion. We will do that in the new version.
>
> <To 4>: STORM is not the contribution of our paper. In Line 215, we have clearly cited the original paper that proposed STORM. Maybe the words, _the key idea_,  lead to this misunderstanding. We will revise them to _the key step_ to avoid this misunderstanding in the next version.
>
> <To 5>:  Yes, the proof techniques in Section 4.3 is new. It is totally different from the existing method. In particular, we have pointed out the key difference between our proof technique and that of Kwon et al., 2023 in the first paragraph in Section 4.3. We sincerely hope the reviewer could take a look at their Lemma D.3 to see the clear difference between our paper and theirs.
>
> <To 6 and 7>: We will revise our writing as the reviewer suggested in the next version.

---

> > ### Author Rebuttal · Reviewer_ksL7 · 2026-04-02
> >
> > I understand you have developed new proof techniques. weakness <1> was referring to your assumptions mostly. I read your 4.3 again and it is indeed on the techniques to prove under your assumptions. Compared to (Shen and Chen, 2023), is the biggest difference in problem setup not the stochasticity? Also, in your paper you said " In the single-agent setting, (Shen & Chen, 2023) developed a single-loop first-order algorithm that uses a deterministic gradient. " Are they single loop or double loop in your opinion? I am confused.

---

> > > ### Author Response · Authors · 2026-04-02
> > >
> > > Regarding the difference from Shen and Chen, 2023, the **problem setup**, **algorithm design**, and **proof techniques** are very different.
> > >
> > > * As for the problem setup, Shen and Chen, 2023 focuses on the deterministic setting, while our algorithm focuses on the stochastic setting.
> > >
> > > * As for the algorithm design, Shen and Chen, 2023 developed a double-loop algorithm (Lines 102-103 is a typo, it should be double-loop), while our algorithm is single-loop.
> > >
> > > * As for the convergence analysis, please see our prior responses and Section 4.3.
> > >
> > > We believe these issues are trivial to address; they mainly involve clarifying some details. We sincerely hope the reviewer can reevaluate our paper based on our contributions to the proof techniques. Thank you!

---

### Decision · Program_Chairs · 2026-04-30

**Decision:**

Accept (regular)

**Comment:**

This paper makes a strong theoretical contribution to decentralized stochastic bilevel optimization by replacing the standard strong convexity assumption with the weaker PL condition, using constant single‑timescale learning rates, and achieving an \(\tilde O(\epsilon^{-5})\) convergence rate via novel proof techniques. The algorithm is Hessian‑/Jacobian‑free and validated on hyperparameter optimization and model pruning. Some reviewers raised concerns about the practical verifiability of the PL condition, the high \(\kappa^9\) dependence, and the lack of direct comparison with other decentralized bilevel methods. Nonetheless, the technical novelty and rigorous analysis are substantial. The paper has clear merit and deserves serious consideration.